# RETRIEVAL-AUGMENTED GENERATION AS NOISY IN-CONTEXT LEARNING: A UNIFIED THEORY AND RISK BOUNDS

## ABSTRACT

Retrieval-augmented generation (RAG) has seen many empirical successes in recent years by aiding the LLM with external knowledge. However, its theoretical aspect has remained mostly unexplored. In this paper, we propose the first finite-sample generalization bound for RAG in in-context linear regression and derive an exact bias-variance tradeoff. Our framework views the retrieved texts as query-dependent noisy in-context examples and recovers the classical in-context learning (ICL) and standard RAG as the limit cases. Within this simplified model, our analysis suggests that an intrinsic ceiling on generalization error can exist on RAG as opposed to the ICL. Furthermore, our framework is able to model retrieval both from the training data and from external corpora by introducing uniform and non-uniform RAG noise. Finally, our theoretical insights of RAG is consistent with preliminary experimental results common QA benchmarks, such as Natural Questions and TriviaQA.

## 1 INTRODUCTION

Retrieval-Augmented Generation (RAG) enhances language models by appending retrieved data to the input, enabling access to information beyond pretraining. It is widely used in open-domain QA, fact-checking, and knowledge-intensive tasks (Huang et al., 2023; Lewis et al., 2020a; Ramos et al., 2022; Sarto et al., 2022; Zhao et al., 2024a). Retrieval sources typically fall into two categories: (1) *labeled dataset*, such as the training dataset itself (Liu et al., 2021; Izacard et al., 2022; Huang et al., 2024), and (2) *generic corpora without labels*, such as Wikipedia (Chen et al., 2017). Despite its promise, empirical studies show that increasing the number of retrieved passages can degrade performance, especially when irrelevant or redundant texts are included (Levy et al., 2025; 2024). However, the theoretical aspects for understanding how retrieval affects generalization remain underexplored.

To study its behavior, we frame RAG as noisy in-context learning (ICL). ICL refers to the ability of language models to adapt given the contextual information without updating model weights (Dong et al., 2024). Under this view, retrieved RAG examples can act as noisy context and their quality depends on the retrieval. This view has motivated the development of many works in in-context retrieval (Luo et al., 2024; Shi et al., 2022), where the goal is to retrieve high-quality demonstration pairs, which reduce the noise of the retrieval.

Our framework viewing RAG as noisy ICL needs to address novel challenges. Prior ICL work has analyzed structured in-context learning where the context consists of fixed format demonstration examples, assuming the examples are clean and i.i.d. (Ahn et al., 2023; Zhang et al., 2024). These assumptions do not hold in RAG, since the retrieved examples are inherently query-dependent and noisy, and their noise level tends to be inversely correlated to their relevance. Furthermore, although retrieved examples close to the query should, in principle, improve the predictive accuracy, their quantitative contribution remains unknown because RAG introduces these examples only at the test time (absent during pretraining), thus imposing a distribution shift. In this work, we take an initial step towards bridging the gap by modeling RAG as a variant of ICL that keeps the two core ingredients, *query dependence* and *noisy examples*, while abstracting away other system details for tractability. To model the noises, we propose and examine two noise regimes: uniform (noise level identical across examples) and non-uniform (noise inversely correlated with relevance), which capture key phenomena seen in practice. Indeed, our experiments on common QA benchmarks show that the two categories of RAG (using training datasets or generic corpora for retrieval) lead to experimental results that align well with our two noise regimes, respectively. This framework thus allows us to quantify the impact of retrieval noise and derive generalization bounds that depend on the number of in-context and RAG examples, and the retrieval distance from queries.

Our contributions are summarized as follows:

- We propose a simplified theoretical framework for analyzing RAG and provide the first finite sample bounds for in-context linear regression with RAG. Our bounds show that the improvement from RAG shrinks as you add more retrieved examples, and can even flip to hurt performance.

- Our framework recovers ICL and standard RAG as limit cases, and also models retrieved data under different noise regimes: uniform and non-uniform retrieval noises.

- We develop new tools for analyzing the query-dependent RAG data, e.g., a derivation of the expectation for high order Gaussian monomial (Lemma 3), which can be useful for future research on RAG.

- We conduct experiments for representative models on common QA datasets and demonstrate that the results align with our theory about the two noise regimes, providing positive support for our analysis.

## 2 RELATED WORK

**Retrieval Augmented Generation** Retrieval-augmented generation (RAG) has emerged as a widely adopted paradigm for enriching LLMs with external knowledge by prepending retrieved passages to the input context (Lewis et al., 2020a; Izacard & Grave, 2020; Borgeaud et al., 2021). From a functional perspective, RAG transforms the model's input distribution by conditioning generation on retrieved textual evidence, often drawn from large-scale corpora via learned or heuristic retrieval mechanisms (Li et al., 2023; Meng et al., 2024; Chen et al., 2024). While much of the literature focuses on improving retrieval quality, system performance (Asai et al., 2023; Li et al., 2024; Xu et al., 2024a), and answer reliability (Xiang et al., 2024; Xu et al., 2024a). There are some recent attempts in studying the theory of RAG (Xu et al., 2024b; Kang et al., 2024) focusing on the token-level risk, and the sequence-level risk still remains underexplored. **In-context Learning (ICL)** ICL obtains its popularity from the original GPT-3 paper (Brown et al., 2020), and becomes widely used in LLM applications (Dong et al., 2024; Min et al., 2021). The recent advance in ICL theory (Ahn et al., 2023; Zhang et al., 2024; Xie et al., 2021) provides a rigorous and versatile framework to study transformers and LLMs. People have used this ICL framework to study novel settings, like out-of-distribution tasks (Wang et al., 2024b) and test-time training (Gozeten et al., 2025). People have also studied the noisy in-context learning from robustness (Cheng et al., 2025) and calibration perspectives (Zhao et al., 2024b), which are different from our setup.

**In-context Retrieval** In-context retrieval (Luo et al., 2024) refers to retrieving a set of query-dependent demonstrations rather than using a fixed set of demonstrations. The label of the demonstration pairs can come from various sources, such as in-domain training set (Izacard et al., 2022; Huang et al., 2024; Ye et al., 2023), cross-domain data (Cheng et al., 2023; Shi et al., 2022), automatic LLM generation (Zhang et al., 2022; Li & Qiu, 2023), pseudo-labels from unstructured data (Lyu et al., 2022; Li et al., 2022). In our theoretical analysis and experiments, we focus on the simplest in-context retrieval, in-domain retrieval from the training set, as in (Izacard et al., 2022; Huang et al., 2024). Note that in-context retrieval is a term developed later and some earlier papers discuss ICL with retrieval as retrieving relevant documents without labels (Ram et al., 2023).

## 3 PROBLEM SETUP

Our problem setup is similar to (Zhang et al., 2024; Garg et al., 2022), but with RAG examples to form the additional in-context examples. It is worth noting that many works focus on ICL at test (inference) time, specifically without parameter updates (Dong et al., 2022). Our work adopts the framework of *ICL with warmup*, also known as, *supervised in-context training*. Specifically, we assume that the pretraining data is also formed by in-context examples. Then, during the test time, we formed prompts with in-context examples with additional RAG examples.

**Notations** We denote $[n] = \{1, \ldots, n\}$ for an integer $n \geq 1$. We denote the trace product of two matrices $A, B \in \mathbb{R}^{m \times n}$ as $\mathrm{tr}(AB^\top)$. We denote $f(m, n) = \mathcal{O}_{m,n}(g(m, n))$, iff there exists $c, m_0, n_0 > 0$ such that $0 \leq f(m, n) \leq cg(m, n)$, $\forall m \geq m_0, n \geq n_0$. We denote $f(m, n) = \Theta_{m,n}(g(m, n))$, iff there exists $c_1, c_2, m_0, n_0 > 0$ such that $c_1 g(m, n) \leq f(m, n) \leq c_2 g(m, n)$, $\forall m \geq m_0, n \geq n_0$.

**Pretraining Data** We consider learning over linear regression data. The training data is a set of prompts. Each prompt is of size $m$: $(\boldsymbol{x}_1, y_1, \ldots, \boldsymbol{x}_m, y_m, \boldsymbol{x}_q) \in \mathbb{R}^{d(m+1)+m}$ where $(\boldsymbol{x}_1, y_1), \ldots, (\boldsymbol{x}_m, y_m)$ form the $m$ demonstration pairs. The goal is to predict $\hat{y}_q$ for the query example $\boldsymbol{x}_q$ to match the true label $y_q$. The prompt is embedded in the following form:

$$\mathbf{P}_m^{\mathrm{pt}} := \begin{pmatrix} \boldsymbol{x}_1 & \boldsymbol{x}_2 & \ldots & \boldsymbol{x}_m & \boldsymbol{x}_q \\ y_1 & y_2 & \ldots & y_m & 0 \end{pmatrix} \in \mathbb{R}^{(d+1) \times (m+1)}, \tag{1}$$

where $(\boldsymbol{x}_1, y_1), \ldots, (\boldsymbol{x}_m, y_m), (\boldsymbol{x}_q, y_q) \overset{\text{i.i.d.}}{\sim} \mathcal{D}_{\text{pt}}$ (pt denoting Pretraining). The output follows the linear model:

$$y_i = \boldsymbol{x}_i^\top \beta_{\text{pt}} + \epsilon_i, \quad \epsilon_i \overset{\text{i.i.d.}}{\sim} \mathcal{N}(0, \sigma^2) \quad \text{under} \quad \mathcal{D}_{\text{pt}}, \tag{2}$$

where $i \in [m] \cup \{q\}$, $\beta_{\text{tt}}$ is the weight vector in pretraining, and $\epsilon_i$ is the noise for example $i$.

**Inference Data (with RAG)** During inference/test time, the test prompt $\mathbf{P}_{m,n}^{\text{tt+rag}}$ (tt denoting test-time) is formed by $m$ in-context pairs $(\boldsymbol{x}_1, y_1), \ldots, (\boldsymbol{x}_m, y_m)$, $n$ retrieval-augmented pairs $(\boldsymbol{x}_1^{\text{rag}}, y_1^{\text{rag}}), \ldots, (\boldsymbol{x}_n^{\text{rag}}, y^{\text{rag}})$, and the query pair $\boldsymbol{x}_q, y_q$. The test prompt is embedded in the following form:

$$\mathbf{P}_{m,n}^{\text{tt+rag}} := \begin{pmatrix} \boldsymbol{x}_1 & \cdots & \boldsymbol{x}_m & \boldsymbol{x}_1^{\text{rag}} & \cdots & \boldsymbol{x}_n^{\text{rag}} & \boldsymbol{x}_q \\ y_1 & \cdots & y_m & y_1^{\text{rag}} & \cdots & y_n^{\text{rag}} & 0 \end{pmatrix} \in \mathbb{R}^{(d+1)\times(m+n+1)}. \tag{3}$$

The input $\boldsymbol{x}$ in each in-context or query pair follows the test-time distribution $\mathcal{D}_{\text{tt}}$, and the label is:

$$y_i = \boldsymbol{x}_i^\top \beta_{\text{tt}} + \epsilon_i, \quad \epsilon_i \overset{\text{i.i.d.}}{\sim} \mathcal{N}(0, \sigma^2) \quad \text{under} \quad \mathcal{D}_{\text{tt}}, \tag{4}$$

where $i \in [m] \cup \{q\}$, $\epsilon_i$ is the noise of example $i$, and $\beta_{\text{tt}}$ is the weight vector during test time. The input $\boldsymbol{x}$ in each RAG pair follows the corresponding RAG distribution $\mathcal{D}_{\text{rag}}(\boldsymbol{x}_q)$: assume the RAG query $\boldsymbol{x}_i^{\text{rag}} = \boldsymbol{x}_q + \boldsymbol{r}_i$ is generated around the query example $\boldsymbol{x}_q$, where $\boldsymbol{r}_i$ is the offset. The label in the RAG example is given by:

$$y_i^{\text{rag}} = (\boldsymbol{x}_i^{\text{rag}})^\top \beta_{\text{tt}} + \epsilon_i^{\text{rag}}, \quad \epsilon_i^{\text{rag}} \overset{\text{i.i.d.}}{\sim} \mathcal{N}(0, \sigma_{\text{rag},i}^2) \quad \text{under} \quad \mathcal{D}_{\text{rag}}(\boldsymbol{x}_q), \tag{5}$$

where $i \in [n]$, $\epsilon_i^{\text{rag}}$ is the noise of the $i$-th RAG example.

For the compactness of writing, we define the following matrices and vectors:

$$\boldsymbol{X}_{\text{icl}} := [\boldsymbol{x}_1^\top; \ldots; \boldsymbol{x}_m^\top], \ \boldsymbol{X}_{\text{rag}} := [(\boldsymbol{x}_1^{\text{rag}})^\top; \ldots; (\boldsymbol{x}_n^{\text{rag}})^\top], \ \boldsymbol{y}_{\text{icl}} := [y_1; \ldots; y_m], \ \boldsymbol{y}_{\text{rag}} := [y_1^{\text{rag}}; \ldots; y_n^{\text{rag}}],$$

$$\boldsymbol{\epsilon}_{\text{icl}} := [\epsilon_1; \ldots; \epsilon_m], \ \boldsymbol{\epsilon}_{\text{rag}} := [\epsilon_1^{\text{rag}}; \ldots; \epsilon_n^{\text{rag}}], \ \boldsymbol{r} = [\boldsymbol{r}_1^\top; \ldots; \boldsymbol{r}_n^\top]$$

$$\boldsymbol{X} = \begin{bmatrix} \boldsymbol{X}_{\text{icl}} \\ \boldsymbol{X}_{\text{rag}} \end{bmatrix} \in \mathbb{R}^{(m+n)\times d}, \ \boldsymbol{X}_{\text{rag}} = \begin{bmatrix} \boldsymbol{x}_q + \boldsymbol{r}_1 \\ \vdots \\ \boldsymbol{x}_q + \boldsymbol{r}_n \end{bmatrix} \in \mathbb{R}^{n\times d}, \ \boldsymbol{y} = \begin{bmatrix} \boldsymbol{y}_{\text{icl}} \\ \boldsymbol{y}_{\text{rag}} \end{bmatrix} \in \mathbb{R}^{m+n}, \ \boldsymbol{\epsilon} = \begin{bmatrix} \boldsymbol{\epsilon}_{\text{icl}} \\ \boldsymbol{\epsilon}_{\text{rag}} \end{bmatrix} \in \mathbb{R}^{m+n}.$$

**Training and Testing** We let $\boldsymbol{W}$ be the model parameters, and $F$ be the model. Given an input prompt $\mathbf{P}_m^{\text{pt}}$ with demonstration pairs, the model predicts $\hat{y}_q := F(\mathbf{P}_m^{\text{pt}}; \boldsymbol{W})$. As a common practice in theoretical studies of LLM for feasible analysis, we use the MSE loss as the evaluation metrics (Zhang et al., 2024; Ahn et al., 2023; Xie et al., 2021). Then, the population loss on the pretraining data is:

$$\mathcal{L}_{\text{pt}}(\boldsymbol{W}) := \mathop{\mathbb{E}}_{(\boldsymbol{x}_1, y_1), \ldots, (\boldsymbol{x}_m, y_m)(\boldsymbol{x}_q, y_q) \sim \mathcal{D}_{\text{pt}}} \left[ \left( y_q - F\left(\mathbf{P}_m^{\text{pt}}; \boldsymbol{W}\right) \right)^2 \right]. \tag{6}$$

Its minimizer is denoted as:

$$\bar{\boldsymbol{W}}^* := \min_{\boldsymbol{W}} \mathcal{L}_{\text{pt}}(\boldsymbol{W}). \tag{7}$$

To apply the pretrained $\boldsymbol{W}^*$ from the pretraining context size of $m$ to the test-time context size of $m + n$, we will need to scale it properly (see Lemma 1) and use

$$\boldsymbol{W}^* = \frac{m}{m+n} \bar{\boldsymbol{W}}^*. \tag{8}$$

During the test time we evaluate the population loss over the test prompt with RAG examples $\mathbf{P}_{m,n}^{\text{tt+rag}}$:

$$\mathcal{L}_{\text{tt+rag}}(\boldsymbol{W}) := \mathop{\mathbb{E}}_{\substack{(\boldsymbol{x}_1, y_1), \ldots, (\boldsymbol{x}_m, y_m)(\boldsymbol{x}_q, y_q) \sim \mathcal{D}_{\text{tt}} \\ (\boldsymbol{x}_1^{\text{rag}}, y_1^{\text{rag}}), \ldots, (\boldsymbol{x}_n^{\text{rag}}, y_n^{\text{rag}}) \sim \mathcal{D}_{\text{rag}}(\boldsymbol{x}_q)}} \left[ \left( y_q - F\left(\mathbf{P}_{m,n}^{\text{tt+rag}}; \boldsymbol{W}\right) \right)^2 \right]. \tag{9}$$

**Model Architecture** We study the single-layer linear self-attention model (LSA) as the framework for theoretical analysis, similar to many existing studies (e.g., (Ahn et al., 2023; Zhang et al., 2024)). The prediction of the model $F$ on a prompt $\mathbf{P}$ with query $\boldsymbol{x}_q$ is:

$$\hat{y}_q := F(\mathbf{P}) = [\mathbf{P}\boldsymbol{W}_Q \boldsymbol{W}_K^\top \mathbf{P}^\top \mathbf{P} \boldsymbol{W}_V]_{m+n+1, d+1} = \boldsymbol{x}_q^\top \boldsymbol{W} \boldsymbol{X}^\top \boldsymbol{y}, \tag{10}$$

where the query, key, and value matrices $\boldsymbol{W}_Q, \boldsymbol{W}, \boldsymbol{W}_V \in \mathbb{R}^{(d+1)\times(d+1)}$ are parameterized by $\boldsymbol{W}$ in the following way:

$$\boldsymbol{W}_Q \boldsymbol{W}_K^\top = \begin{bmatrix} \boldsymbol{W} & \boldsymbol{0}_{d\times 1} \\ \boldsymbol{0}_{1\times d} & 0 \end{bmatrix}, \quad \boldsymbol{W}_V = \begin{bmatrix} \boldsymbol{0}_{d\times d} & \boldsymbol{0}_{d\times 1} \\ \boldsymbol{0}_{1\times d} & 1 \end{bmatrix}.$$

We note that this parameterization is commonly used in the previous works (Ahn et al., 2023; Zhang et al., 2024), and is shown to capture the key properties of in-context learning. Furthermore, (Ahn et al., 2023) shows that the formulation is the optimum converged from pretraining on Gaussian data.

## 4 THEORETICAL ANALYSIS: GENERALIZATION BOUND FOR RAG

To study test-time error and sample complexity in in-context linear regression with RAG examples, we consider two noise regimes: **uniform retrieval noise** and **non-uniform retrieval noise**. Uniform retrieval noise assumes the RAG noise $\epsilon_i^{\mathrm{rag}}$ for each example $i$ is i.i.d. Since its variance is distance-agnostic, it can model a scenario of retrieval where the noise is similar across data points. Non-uniform retrieval noise assumes either the variance or the label-corruption probability grows with the variance of retrieval vectors — e.g. $\sigma_{\mathrm{rag},i}^2$ increases with $\delta_i^2$ or the probability of making mistakes increases with $\delta_i^2$. This captures retrieval from datasets where near neighbors often supply the right signal while far ones are potentially noisy or even misleading. Because the noise spectrum is now heavy-tailed, adding more RAG examples past a threshold could yield diminishing benefits for RAG examples and even become counter-productive. Framing RAG through these two lenses provides intuition about when extra retrieved examples will pay off, and when they will hit the intrinsic ceiling and more retrieved examples do not help anymore. These are well corroborated by our experimental results on real data (see Section 5).

First, we introduce the key data assumptions.

**Assumption 1** (Gaussian Retrieval Offset). *We assume the retrieval offset $\boldsymbol{r}_i$, $\forall i \in [n]$ to follow a Gaussian distribution:* $\boldsymbol{r}_i \overset{i.i.d.}{\sim} \mathcal{N}\left(0, \delta_i^2 I_d\right).$

The key property that we want to control for RAG examples is its distance from the query points $\boldsymbol{x}_q$. However, modeling the queried example directly through the retrieval distance leads to complicated theoretical analysis. Here, we note that the retrieval distance $\|\boldsymbol{r}_i\|_2$ converges to a distribution concentrated in an $\mathcal{O}(\delta_i\sqrt{d})$ ball around the query with respect to $d$ (Cover & Hart, 1967). Thus, controlling the variance of the retrieval offset can alternatively control the retrieval distance. And we make the following additional data assumptions.

**Assumption 2** (Data Assumption). *We assume the data follows the following:*

1. PRETRAINING EXAMPLES ($\mathcal{D}_{\mathrm{pt}}$). *For a pretraining prompt of length $m + 1$ and for all $i \in [m] \cup \{q\}$, we assume $\boldsymbol{x}_i \overset{i.i.d.}{\sim} \mathcal{N}(0,\Sigma)$, $\epsilon_i \overset{i.i.d.}{\sim} \mathcal{N}(0,\sigma^2)$, $\beta_{\mathrm{pt}} \sim \mathcal{N}(0,I)$.*

2. TEST TIME EXAMPLES ($\mathcal{D}_{\mathrm{tt}}$). *For a test-time prompt of length $m + n + 1$ and for all $i \in [m] \cup \{q\}$, we assume $\boldsymbol{x}_i \overset{i.i.d.}{\sim} \mathcal{N}(0,\Sigma)$, $\epsilon_i \overset{i.i.d.}{\sim} \mathcal{N}(0,\sigma^2)$, $\beta_{\mathrm{tt}} \sim \mathcal{N}(0,I)$.*

3. TEST-TIME RAG EXAMPLES ($\mathcal{D}_{\mathrm{rag}}(\boldsymbol{x}_q)$). *For a test-time prompt of length $m + n + 1$ and for all $i \in [m+1,\ldots,m+n]$, we assume $\boldsymbol{x}_i^{\mathrm{rag}} \overset{i.i.d.}{\sim} \mathcal{N}(0,\Sigma)$, $\epsilon_i^{\mathrm{rag}} \sim \mathcal{N}(0,\sigma_{\mathrm{rag},i}^2)$, and the same $\beta_{\mathrm{tt}}$ as (2).*

Here, we assume the generic Gaussian property for the input, and isotropic Gaussian property for the noise and the weight vector, a common assumption made in ICL theory (Ahn et al., 2023; Gozeten et al., 2025) for simple yet meaningful analysis.

---

**Overview of the Key Results**

- (*Uniform Noise*) RAG examples are as effective as ICL examples in reducing the variance-induced err but ineffective at reducing the bias-induced err, causing a loss plateau for $n \to \infty$.
- (*Non-Uniform Noise*) RAG could improve the variance-induced error up to a finite $n$ at a cost of increasing bias-induced error.

---

**Roadmap** Under these assumptions and uniform retrieval noise, we will first derive the population loss of RAG, $\mathcal{L}_{\mathrm{tt+rag}}(\boldsymbol{W})$, for general $\boldsymbol{W}$ as in Theorem 1, analyze its finite sample complexity under the optimal pretrained weight $\boldsymbol{W}^*$ as in Proposition 1 and derive an optimal number of RAG examples of $n^*$ for a given number of ICL examples $m$

as in Proposition 2. These discussions leads to our first key result. Then, under the non-uniform retrieval noise, we will prove the sample complexity under the distance-proportional noise (Theorem 2) and distance-weighted mixture noise (Theorem 3), and obtain our second key results above.

## 4.1 UNIFORM RETRIEVAL NOISE

**Assumption 3** (Uniform Retrieval Noise). *The RAG noise $\epsilon_{\text{rag}}$ shares the same Gaussian distribution with variance $\sigma_{\text{rag}}^2$, i.e. $\forall i \in [m+1, \ldots, m+n]$, $\sigma_{\text{rag},i}^2 = \sigma_{\text{rag}}^2$.*

First, we present the assumption for uniform retrieval noise. In other words, all RAG examples are as helpful, and its improvement on the actual prediction is determined by the retrieval distance.

**Theorem 1** (Population Loss for ICL with RAG Examples). *Under Assumption 1, 2, 3, $\Sigma = I$, the population loss of the linear self-attention predictor $\hat{y}_q = \boldsymbol{x}_q^\top \boldsymbol{W} \boldsymbol{X}^\top \boldsymbol{y}$ satisfies*

$$\mathcal{L}_{\text{tt+rag}}(\boldsymbol{W}) = \underbrace{\mathbb{E}\left(\hat{y}_q - \mathbb{E}\left(\hat{y}_q\right)\right)^2}_{:=\text{err}_{\text{variance}}(\boldsymbol{W})} + \underbrace{\mathbb{E}\left(\mathbb{E}\left(\hat{y}_q\right) - \mathbb{E}\left(y_q\right)\right)^2}_{:=\text{err}_{\text{bias}}(\boldsymbol{W})} + \underbrace{\sigma^2}_{\textit{irreducible noise}} \quad , \textit{ and specifically,} \tag{11}$$

$$\text{err}_{\text{variance}}(\boldsymbol{W}) = \left[m\sigma^2 + \left(1+\delta^2\right)n\sigma_{\text{rag}}^2\right]\text{tr}(\boldsymbol{W}^\top\boldsymbol{W}) + n\sigma_{\text{rag}}^2\text{tr}(\boldsymbol{W}^2) + n\sigma_{\text{rag}}^2\text{tr}(\boldsymbol{W})^2$$

$$\text{err}_{\text{bias}}(\boldsymbol{W}) = \beta_{\text{tt}}^\top\left[I - (n\delta^2 + 2n + m)(\boldsymbol{W} + \boldsymbol{W}^\top) - 2n\,\text{tr}(\boldsymbol{W})I + M_4\right]\beta_{\text{tt}}$$

$$= \beta_{\text{tt}}^\top\left[I - (n\delta^2 + 2n + m)(\boldsymbol{W} + \boldsymbol{W}^\top) - 2n\,\text{tr}(\boldsymbol{W})I\right.$$

$$+ \left[n^2\left(2+\delta^2\right) + n\left(m+\delta^2\right)\right]\left(\boldsymbol{W}^2 + \left(\boldsymbol{W}^2\right)^\top\right) + 2n(n+\delta^2)\boldsymbol{W}\boldsymbol{W}^\top$$

$$+ \left[m^2 + m + mn\left(2+2\delta^2\right) + n^2\left(2+2\delta^2+\delta^4\right) + n\left(2\delta^2+\delta^4\right)\right]\boldsymbol{W}^\top\boldsymbol{W}$$

$$+ \left[n^2\left(2+\delta^2\right) + n\left(m+\delta^2\right)\right]\left(\text{tr}(\boldsymbol{W})\left(\boldsymbol{W} + \boldsymbol{W}^\top\right)\right)$$

$$+ \left[n^2 + n\delta^2\right]\left(\text{tr}(\boldsymbol{W})^2 + \text{tr}\left(\boldsymbol{W}^2\right)\right)I + \left[m + n^2 + n\left(2\delta^2+\delta^4\right)\right]\text{tr}\left(\boldsymbol{W}^\top\boldsymbol{W}\right)I\right]\beta_{\text{tt}}.$$

Here, we derive the exact bias-variance decomposition for ICL with RAG. The first line is the variance-induced error formed by a weighted sum of noise from ICL examples and RAG examples. Because of the implicit scaling of $\boldsymbol{W}$ as discussed in Lemma 1, the second order term in $\boldsymbol{W}$ will introduce an additional weight scaling of $\frac{m^2}{(m+n)^2}$ when adapting from the weight learned on $m$ size context to $m+n$ size context. Thus, larger $n$ will let $\text{err}_{\text{variance}}(\boldsymbol{W}) \to 0$, and the convergence rate is affected by $\delta^2$. Larger retrieval distance leads to a slower convergence. The bias-induced error is composed of all possible monomials of $\boldsymbol{W}$ up to the 2nd-order with $\text{tr}$ operation. The complex dependency on $m, n, \delta^2, d$ requires additional assumptions on $\boldsymbol{W}$ to further interpret. As a sanity check, when $n = 0$ (ICL-only), this decomposition can exactly recover loss as in Lemma B.2 in (Gozeten et al., 2025).

As a proof sketch, we first compute $\text{err}_{\text{variance}}(\boldsymbol{W}) = \mathbb{E}(\boldsymbol{x}_q^\top \boldsymbol{W} \boldsymbol{X}^\top \boldsymbol{\epsilon})^2$ by splitting the calculation for ICL and RAG examples based on $\boldsymbol{X}$. Then, we compute $\text{err}_{\text{bias}}(\boldsymbol{W}) = \mathbb{E}[(\boldsymbol{x}_q^\top(I - \boldsymbol{W}\boldsymbol{X}^\top\boldsymbol{X})\beta_{\text{tt}})^2]$. The main technical challenge lies in the dependency of $\boldsymbol{X}_{\text{rag}}$ on $\boldsymbol{x}_q$, and $\text{err}_{\text{bias}}$ has a 6th-order dependency on $\boldsymbol{x}_q$ (2 from $\boldsymbol{x}_q$ and 4 from $\boldsymbol{X}$). As shown in Lemma 3, $\mathbb{E}\left[\boldsymbol{x}_q\boldsymbol{x}_q^\top A\boldsymbol{x}_q\boldsymbol{x}_q^\top B\boldsymbol{x}_q\boldsymbol{x}_q^\top\right]$ gives 15 new terms that include all the second order monomials of $\boldsymbol{W}$ with $\text{tr}$. The calculation requires multiple careful applications of Isserlis' theorem (Isserlis, 1918), and the full proof can be seen in Section B. Also, see Theorem 4 in the appendix for the full theorem with generic $\Sigma$.

Here, we present the finite sample bound for pretrained $\boldsymbol{W}^*$ for better interpretation.

**Proposition 1** (Finite Sample Generalization Bound). *Under Assumption 1, 2, 3, if $\delta^2 \ll 1, \Sigma = I$,*

$$\mathcal{L}_{\text{tt+rag}}(\boldsymbol{W}^*) = \mathcal{O}_{m,n}\left(\sigma^2 + \underbrace{\frac{dm}{(m+n)^2}\sigma^2 + \frac{d^2n}{(m+n)^2}\sigma_{\text{rag}}^2}_{\text{err}_{\textit{variance}}(\boldsymbol{W}^*)} + \underbrace{\|\beta_{\text{tt}}\|_2^2\left[\frac{d}{m} + d^2\left(\frac{n}{m+n}\right)^2\right]}_{\text{err}_{\textit{bias}}(\boldsymbol{W}^*)}\right)$$

$$\text{err}_{\text{variance}}(\boldsymbol{W}^*) = \begin{cases} \mathcal{O}_m(\frac{d}{m}\sigma^2 + \frac{d^2}{m^2}\sigma_{\text{rag}}^2) = \mathcal{O}_m\left(\frac{1}{m}\right) & m \to \infty, n \textit{ fixed.} \\ \mathcal{O}_n(\frac{d}{n^2}\sigma^2 + \frac{d^2}{n}\sigma_{\text{rag}}^2) = \mathcal{O}_n\left(\frac{1}{n}\right) & n \to \infty, m \textit{ fixed} \\ \mathcal{O}_m(\frac{d}{m}\sigma^2 + \frac{d^2}{m}\sigma_{\text{rag}}^2) = \mathcal{O}_m\left(\frac{1}{m}\right) & m \to \infty, n = \Theta_m(m) \end{cases} \tag{12}$$

$$\text{err}_{\text{bias}}(\boldsymbol{W}^*) = \begin{cases} \mathcal{O}_m\left(\|\beta_{\text{tt}}\|_2^2 \frac{d}{m}\right) & \text{if } m \to \infty, \ n \text{ is fixed} \\ \mathcal{O}_n\left(\|\beta_{\text{tt}}\|_2^2 d^2\right) = C_1 & \text{if } n \to \infty, \ m \text{ is fixed} \\ \mathcal{O}_m\left(\|\beta_{\text{tt}}\|_2^2 \left(\frac{d}{m} + d^2\right)\right) = C_2 + \mathcal{O}_m(\|\beta_{\text{tt}}\|_2^2 \frac{d}{m}) & \text{if } m \to \infty, \ n = \Theta_m(m) \end{cases} \tag{13}$$

Here, we assume $\delta^2 \ll 1$ as the test time example $\boldsymbol{x}_i$ has only a variance of $I$, and it is unrealistic to assume a higher retrieval variance than the input variance. On the limit case where $m \to \infty$ and $n$ are fixed, we observe that both variance-induced and bias-induced error decay at a rate of $\mathcal{O}\left(1/m\right)$, matching the results from the existing paper (Ahn et al., 2023; Zhang et al., 2024). When $n \to \infty$, the variance-induced error decays as $\mathcal{O}\left(1/n\right)$ matching the $\mathcal{O}(1/m)$ rate. However, introducing the RAG is ineffective at reducing the bias-induced error. Even when $m \to \infty$, increasing $n$ will cause a loss plateau.

This effect can be explained by the underlying adaptive ability of transformers. In an online learning setup, we could always use the mean of the queried data as the prediction. However, in the LSA setup, the pretrained $\boldsymbol{W}^*$ serves as a proxy for $\mathbb{E}^{-1}(\boldsymbol{X}^\top \boldsymbol{X})$. In order to retain the adaptivity to the entire distribution of $\beta_{\text{tt}}$, we cannot use the optimal linear classifier $(\boldsymbol{X}^\top \boldsymbol{X})^{-1} \boldsymbol{X}^\top \boldsymbol{y}$ or use the mean of the retrieved examples ad hoc. At the test stage, $\boldsymbol{X}_{\text{rag}}$ only appears in $\boldsymbol{X}^\top \boldsymbol{y}$ and not in $\boldsymbol{W}^*$. The difference between $\mathcal{D}_{\text{rag}}(\boldsymbol{x}_q)$ and $\mathcal{D}_{\text{tt}}$ directly leads to the increase of variance worsened by the increase of $n$. See full proof in Section B. Now, a natural question is whether we can find a balance of variance and bias and obtain an optimal RAG example size $n^*$.

**Proposition 2.** *Under Assumption 1,2,3, $\delta^2 \ll 1, \Sigma = I$, and reasonable choice of $\sigma^2, \sigma_{\text{rag}}^2$ ($\sigma^2, \sigma_{\text{rag}}^2 \ll \|\beta_{\text{tt}}\|_2^2$), the optimal $n^*$ that minimizes the RAG loss follows:*

$$n^* = \mathcal{O}_m\left(\frac{m\left(d^2\|\beta_{\text{tt}}\|_2^2 + d\sigma^2 - d^2\sigma_{\text{rag}}^2\right)}{md^2\|\beta_{\text{tt}}\|_2^2 - d^2\sigma_{\text{rag}}^2}\right) = \mathcal{O}_m\left(\frac{d\|\beta_{\text{tt}}\|_2^2 + \sigma^2 - d\sigma_{\text{rag}}^2}{d\|\beta_{\text{tt}}\|_2^2}\right) \tag{14}$$

*and the improvement on loss from picking the optimal $n^*$ over $n = 0$ is given as:*

$$\mathcal{L}_{\text{tt+rag}}(\boldsymbol{W}^*)|_{n=0} - \mathcal{L}_{\text{tt+rag}}(\boldsymbol{W}^*)|_{n=n^*} = \mathcal{O}_m\left(\frac{1}{m^2}\right). \tag{15}$$

In fact, the optimal $n^*$ does not scale with $m$ omitting the lower-order terms. Note that for $\|\beta_{\text{tt}}\|_2^2 = \mathcal{O}_m(1)$, $\|\beta_{\text{tt}}\|_2^2$ will dominate the numerator for reasonable choices of $\sigma^2$ and $\sigma_{\text{rag}}^2$. A larger ICL noise $\sigma^2$ leads to a larger $n^*$, i.e. requiring more RAG examples to compensate for the loss. A larger RAG noise $\sigma_{\text{rag}}^2$ leads to a smaller $n^*$, i.e. less efficiency on RAG examples. And the improvement converges at $\mathcal{O}(\frac{1}{m^2})$, diminishing for large $m$. See the full proof in Section B. Several empirical works also observe a performance drop when increasing the number of retrieved examples (Wang et al., 2024a; Levy et al., 2025).

## 4.2 NON-UNIFORM RETRIEVAL NOISE

The uniform-noise setup in Section 4.1 relies on a retrieval pool of data with similar noise, so we could keep the variance $\sigma_{\text{rag}}^2$ fixed. In open-domain retrieval, this assumption could collapse: many retrieved examples could contain no answer or even a wrong answer. Empirically, people have observed that passages that are closer to the query vector $\boldsymbol{x}_q$ are more likely (Yang & Seo, 2020; Yoran et al., 2023; Lewis et al., 2020b) to contain the correct label. We want to theoretically investigate if the following hypothesis still holds:

> Closer to query $\boldsymbol{x}_q \implies$ *more likely* to contain *correct* answer.

### 4.2.1 DISTANCE-PROPORTIONAL NOISE (DPN)

We first investigate the scenario where the retrieval noise is proportional to the retrieval distance. Since the ICL analysis only applies to the mean-squared error loss, we study the effect of RAG under DPN on the correctness of the predictions.

**Assumption 4** (Distance-Proportional Noise). *There exists a constant $\gamma_1 > 0$ such that, for every retrieved sample $i$, $\sigma_{\text{rag},i}^2 = \gamma_1 \sigma^2 \delta_i^2$, i.e. the RAG noise variance grows linearly with the variance $\delta_i^2$ that governs the retrieval distance.*

Under the new data assumption, we denote the corresponding RAG loss, bias-induced error, and variance-induced error for $\boldsymbol{W}$ to be $\hat{\mathcal{L}}_{\text{tt+rag}}(\boldsymbol{W})$, $\hat{\text{err}}_{\text{bias}}(\boldsymbol{W})$, and $\hat{\text{err}}_{\text{variance}}(\boldsymbol{W})$.

**Theorem 2** (Finite Sample RAG Generalization Bound under DPN). *Under Assumption [1], [2], $\Sigma = I$, [4], the population loss is given as:*

$$\hat{\mathrm{err}}_{\mathrm{variance}}(\boldsymbol{W}) = m\sigma^2 \operatorname{tr}(\boldsymbol{W}^\top \boldsymbol{W}) + \sum_{i=1}^{n} \gamma_1 \delta_i^2 [(1 + \delta_i^2) \operatorname{tr}(\boldsymbol{W}^\top \boldsymbol{W}) + \operatorname{tr}(\boldsymbol{W}^2) + \operatorname{tr}(\boldsymbol{W})^2].$$

*If the variance of the retrieval distance follows power law, i.e. $\exists \gamma_2 > 0, q \geq 0$ s.t. $\delta_i^2 = \gamma_2 i^q$, then*

$$\hat{\mathrm{err}}_{\mathrm{bias}}(\boldsymbol{W}^*) = \mathcal{O}_{m,n}\left(\mathrm{err}_{bias}(\boldsymbol{W}^*) + \|\beta_{\mathrm{tt}}\|_2^2 \left[\frac{dn^{2q+1} + n^{2q+2}}{(m+n)^2}\right]\right) \tag{16}$$

*and*

$$\hat{\mathrm{err}}_{\mathrm{variance}}(\boldsymbol{W}^*) = \mathcal{O}_n\left(\frac{dm\sigma^2 + d(n^{2q+1})\sigma^2}{(m+n)^2}\right) = \begin{cases} \mathcal{O}_n\left(dn^{2q-1}\sigma^2\right) & \text{if } n \to \infty, q \leq 1/2 \\ \text{diverges} & \text{if } n \to \infty, q > 1/2 \end{cases}. \tag{17}$$

Here, we derive the sample complexity under DPN. A second order dependency on $\delta_i^2$ shows up in both the variance-induced and bias-induced error (exact form seen in Section [B]). Thus, the $\delta_i^2$-involved constant will dominate the other constants. Specifically, it even leads to divergence for $q > 1/2$ for the variance-induced error and $q > 0$ for the bias-induced error.

### 4.2.2 DISTANCE-WEIGHTED MIXTURE NOISE

In this section, we discuss the scenario where further RAG examples are less likely to contain the correct answers. We use a pair of large and small noises to model the correct/incorrect examples.

**Assumption 5** (Distance-Weighted Mixture Noise). *We assume that the RAG noise is formed by a mixture of small and large noise:*

$$y(\boldsymbol{x}_{\mathrm{rag}}) = \begin{cases} f(\boldsymbol{x}_{\mathrm{rag},i}) + \epsilon_s & \text{w.p. } p_i \\ f(\boldsymbol{x}_{\mathrm{rag},i}) + \epsilon_l & \text{w.p. } 1 - p_i \end{cases},$$

*where $\epsilon_s \sim \mathcal{N}(0, c_s\sigma^2)$ corresponds to the small noise and $\epsilon_l \sim \mathcal{N}(0, c_l\sigma^2)$ corresponds to the large noise, with $c_l \geq c_s \geq 0$. The probability of sampling small noise $p_i$ follows an inverse power law of the variance of the retrieval distance, i.e. $p_i = (1 + \delta_i^2)^{-\tilde{q}}, \tilde{q} \geq 0$.*

Here, we choose the sampling probability (of small noise) $p_i$ to follow a polynomial decay and the constant 1 here is to ensure $p_i = 0$ when $\delta_i^2 = 0$. Under the new data assumption, we denote the corresponding RAG loss, bias-induced error, and variance-induced error for $\boldsymbol{W}$ to be $\tilde{\mathcal{L}}_{\mathrm{tt+rag}}(\boldsymbol{W})$, $\hat{\mathrm{err}}_{\mathrm{bias}}(\boldsymbol{W})$, and $\hat{\mathrm{err}}_{\mathrm{variance}}(\boldsymbol{W})$.

**Theorem 3** (Finite Sample RAG Bound under Distance-Weighted Mixture Noise). *Under Assumption [1], [2], [5], $\Sigma = I$, then $\hat{\mathrm{err}}_{bias}(\boldsymbol{W}) = \hat{\mathrm{err}}_{bias}(\boldsymbol{W})$, and*

$$\hat{\mathrm{err}}_{\mathrm{variance}}(\boldsymbol{W}) = m\sigma^2 \operatorname{tr}(\boldsymbol{W}^\top \boldsymbol{W}) + \sum_{i=1}^{n} \left(p_i \sigma_s^2 + (1 - p_i)\sigma_l^2\right)[(1 + \delta_i^2) \operatorname{tr}(\boldsymbol{W}^\top \boldsymbol{W}) + \operatorname{tr}(\boldsymbol{W}^2) + \operatorname{tr}(\boldsymbol{W})^2].$$

*If the variance of the retrieval distance follows power law, i.e. $\exists \gamma_2 > 0, q \geq 0$ s.t. $\delta_i^2 = \gamma_2 i^q$, then:*

$$\tilde{\mathrm{err}}_{\mathrm{variance}}(\boldsymbol{W}^*) = \begin{cases} \mathcal{O}_n\left(c_l dn^{q-1}\sigma^2 - (c_l - c_s)\sigma^2 dn^{q-1-q\tilde{q}}\right) & \text{if } n \to \infty, q \leq 1 \\ \text{diverges} & \text{if } n \to \infty, q > 1 \end{cases}. \tag{18}$$

The bias-induced error here is the same as in DPN, since we assume a polynomial dependency for $\delta_i^2$ on $i$ in both settings and the bias-induced error is independent of the variance of noise. Even though the variance of small/large noise is bounded, the dependency on the retrieval distance leads to the divergence at large $q$ ($q > 1$). The large prediction noise will dominate the variance-induced error, but a larger gap between large and small noise ($c_l - c_s$) can mitigate the error by a ratio of $\mathcal{O}_n(n^{-q\tilde{q}})$. That is, the smaller $q$ and $\tilde{q}$ are, the lower the error.

We note that the uniform noise scenario can also admit the mixture noise model by taking a constant $p_i, \forall i$, resulting in a form similar to the standard uniform retrieval noise in Proposition [1].

## 5 EXPERIMENTS

We investigate the effect of RAG with three guiding questions: (**Q1**) Whether RAG data outperform randomly sampled in-context examples? (**Q2**) What are the impacts of the RAG examples from training data and RAG passages from external corpora? (**Q3**) With a fixed budget, what is the effect of varying the ratio between the two types of RAG data? Our experiments provide the following findings: (**A1**) RAG data lead to better performance than in-context ones under different data budgets. (**A2**) Interestingly, the first few RAG training examples significantly improve performance, but later ones are harmful, because the first few are highly relevant but later ones are noise rather than signal. In contrast, RAG passages from external corpora can slowly but monotonically improve the performance, because external corpora are large enough to provide noisy but still relevant data. These are captured by different noise models in our theory. (**A3**) The performance is not monotonic with the ratio, and the sweet spot depends on the data/model.

We use representative models **ATLAS** Izacard et al. (2022) and **RAVEN** Huang et al. (2024) on two standard open-domain question answering benchmarks **Natural Questions (NQ)** Kwiatkowski et al. (2019) and **TriviaQA** Joshi et al. (2017). For evaluation, the context consists of $m$ in-context examples, and $n$ RAG data points (including $n_1$ RAG examples from the training data and $n_2$ RAG passages from external corpora like Wikipedia, so $n = n_1 + n_2$). We choose different $m, n_1, n_2$'s for our study purpose and report the standard exact match (EM) accuracy on the test set. See Section C for further experiment setup details.

**RAG v.s. In-Context** For a budget $c$, we compare using RAG only ($m = 0, n_1 = n_2 = c/2$) and in-context examples only ($m = c, n_1 = n_2 = 0$). The results in Figure 1 show that RAG consistently outperforms in-context examples, as RAG provides query-relevant data with more signals to address the query, consistent with our analysis.

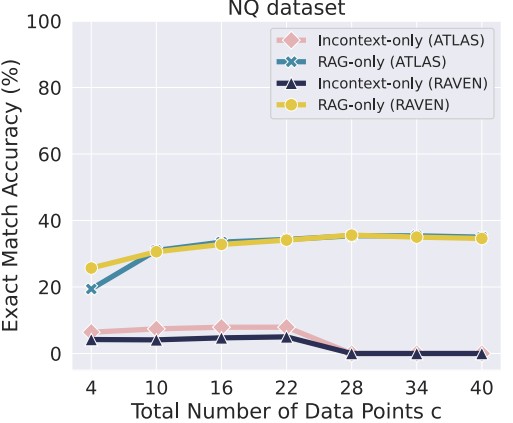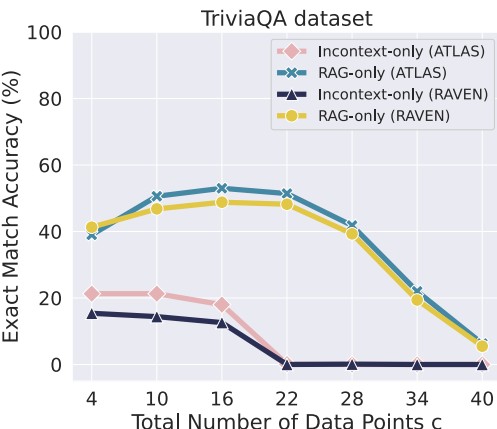

Figure 1: We compare performance between the RAG-only ($c = m$) versus in-context-only methods ($c = n_1 + n_2, n_1 = n_2$), where $c$ is the total number of data, $n_1$ refers to retrieved examples and $n_2$ to passages.

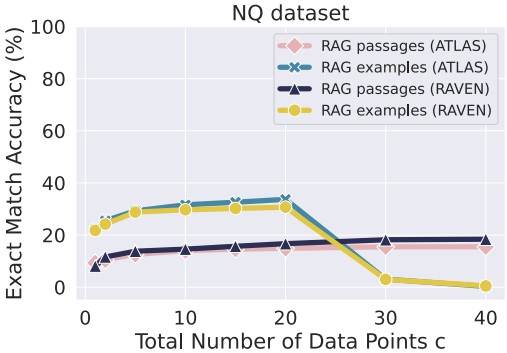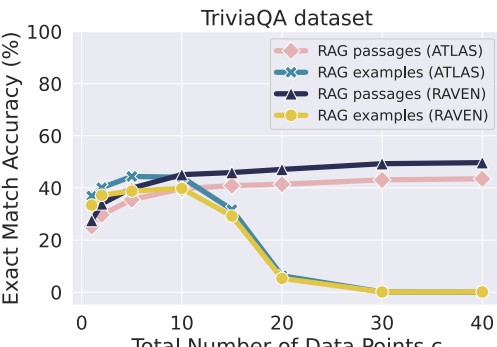

Figure 2: We compare the performance of RAG using examples ($c = n_1$) versus passages ($c = n_2$).

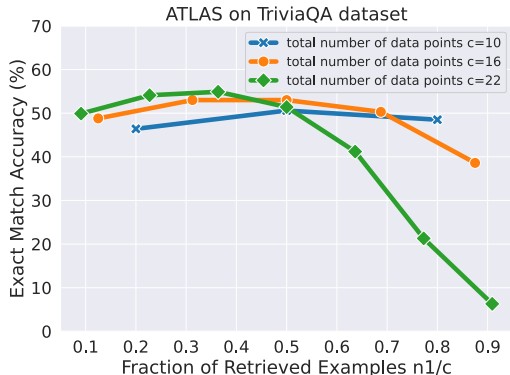

(a) ATLAS Performance as a function of $n_1/c$ under different data points $c$ on NQ.

(b) ATLAS Performance as a function of $n_1/c$ under different data points $c$ on TriviaQA.

Figure 3: Performance sensitivity to $n_1/c$ under total data points $c$. $n_1$ refers to retrieved examples and $n_2$ to passages.

**RAG Examples v.s. RAG External Passages** Next, we compare using RAG examples from training data only ($m = 0, n_1 = c, n_2 = 0$) and RAG passages from external corpora only ($m = 0, n_1 = 0, n_2 = c$). The results in Figure 2 show interesting patterns. For RAG examples-only, with more examples, the performance first significantly improves but later drops. This suggests that the first few examples are highly relevant but later ones contain more noise than signal. In contrast, for RAG passages only, the performance increases more slowly but steadily for larger budgets. This suggests the passages retrieved are noisy but still have relevant signals. This aligns with our noise modeling. When $n_1$ is small ($\leq 20$ for NQ and $\leq 10$ for TriviaQA), RAG examples resemble *uniform noise* due to the relevance of retrieved examples. As $n_1$ increases, $n_1$ introduces more irrelevant or conflicting examples (i.e., *non-uniform noise*). On the other hand, $n_2$ passages from external corpora resemble a *uniform noise* regime as the retrieval pool is broad with relevant data but also noisy.

When the retrieval budget is small, retrieval from training examples yields higher accuracy than from passages, even though both operate in the uniform-noise regime. This discrepancy follows from the mixture-noise effects: a passage judged relevant may still lack any answer-bearing text, raising its effective noise level relative to examples. Furthermore, the significant drop for the retrieval from examples as opposed to retrieval from passages can be explained by the size difference for the training data and passages pool (i.e. Wikipedia). Since the passages provide a denser coverage of the semantic space, more passages will remain relevant as opposed to examples. In all, our theory is consistent with both practical data types and matches the empirical results.

**Ratio between RAG Examples and Passages** The different noise properties of the two kinds of RAG data imply that we should find a proper ratio between them when the total budget $c$ is fixed. Figure 3 (Figure 4 for RAVEN in Section C) shows that as the ratio $n_1/c$ increases, the performance initially improves—benefiting from signal information—but eventually declines as low-quality examples dominate the context. The results demonstrate that performance initially improves as more signal (examples) is added, but eventually declines due to increasing noise from low-quality examples. This supports the theoretical perspective of balancing signal and noise in retrieval-augmented inputs.

## 6 CONCLUSION AND LIMITATIONS

We take an initial step by modeling RAG as query-dependent noisy in-context learning and deriving finite-sample error bounds for linear regression that help isolate the roles of retrieval signal and noise; we also explore how these bounds behave under different noise regimes and a simple test-time-training scenario. Experiments on Natural Questions and TriviaQA with RAVEN and ATLAS show trends consistent with our theoretical predictions.

Regarding limitations, our work considers the case without RAG finetuning. In practice, RAG information could also be used for finetuning, but our current work just focuses on the more common practice of using it as in-context information. Furthermore, our bounds focus on the linear setting, opening avenues for future studies on nonlinear methods like kernels and neural networks. While our framework accounts for common RAG noise models, new models may be needed for other types of RAG data. A further direction is to combine RAG with test-time training, studying how on-the-fly adaptation affects both theoretical guarantees and empirical performance. Our experiments feature representative models and datasets, but future research can explore newer retrievers and more RAG applications.

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
