## A  TECHNICAL PRELIMINARIES

**Additional Notations**   For two integer indices $i$ and $j$, we denote $\delta_{ij} = \begin{cases} 1 & \text{if } i = j \\ 0 & \text{if } i \neq j \end{cases}$ as Kronecker delta.

**Lemma 1** (Adapt $\boldsymbol{W}$ to Different Context Size). *Suppose $\bar{\boldsymbol{W}}$ is the weight with context length $m$, then the induced $\boldsymbol{W}$ when evaluating on context of length $m'$ is:*

$$\boldsymbol{W} = \frac{m}{m'}\bar{\boldsymbol{W}}$$

*Proof.*  We note that $\bar{\boldsymbol{W}}$ is the un-normalized weight, i.e. scaling with the inverse context size $1/m$. Only the normalized weight is preserved when applying to a sentence with a different context length.

Then, the prediction is given as:

$$\hat{y} := \boldsymbol{x}_q^\top \boldsymbol{W} \boldsymbol{X}^\top \boldsymbol{y}$$
$$= \boldsymbol{x}_q^\top \frac{1}{m'} \boldsymbol{W}_{\text{Normalized}} \boldsymbol{X}^\top \boldsymbol{y}$$
$$= \boldsymbol{x}_q^\top \underbrace{\frac{1}{m'} m\bar{\boldsymbol{W}}}_{=\boldsymbol{W}} \boldsymbol{X}^\top \boldsymbol{y}$$

Thus,

$$\boldsymbol{W} = \frac{m}{m'}\bar{\boldsymbol{W}}$$

□

**Lemma 2** (Expectation of Mixed 4th-Order Moment of Gaussian). *Suppose $\boldsymbol{x} \sim \mathcal{N}(0, I), \boldsymbol{r} \sim \mathcal{N}(0, \delta^2 I)$, then*

  *1.*
$$\mathbb{E}[\boldsymbol{r}\boldsymbol{r}^\top \boldsymbol{W}^\top \boldsymbol{x}\boldsymbol{x}^\top \boldsymbol{W}\boldsymbol{r}\boldsymbol{r}^\top] = 2\delta^4 \boldsymbol{W}^\top \Sigma \boldsymbol{W} + \delta^4 \operatorname{tr}\left(\boldsymbol{W}^\top \Sigma \boldsymbol{W}\right) I$$
$$= 2\delta^4 \boldsymbol{W}^\top \boldsymbol{W} + \delta^4 \operatorname{tr}(\boldsymbol{W}^\top \boldsymbol{W}) I \quad \textit{if } \Sigma = I \tag{19}$$

  *2.*
$$\mathbb{E}[\boldsymbol{r}\boldsymbol{x}^\top \boldsymbol{W}^\top \boldsymbol{x}\boldsymbol{x}^\top \boldsymbol{W}\boldsymbol{x}\boldsymbol{r}^\top] = \left(\operatorname{tr}\left(\boldsymbol{W}\Sigma\boldsymbol{W}^\top \Sigma\right) + \operatorname{tr}(\boldsymbol{W}\Sigma\boldsymbol{W}\Sigma) + \operatorname{tr}(\boldsymbol{W}\Sigma)\operatorname{tr}\left(\boldsymbol{W}^\top \Sigma\right)\right)\delta^2 I$$
$$= \left(\operatorname{tr}\left(\boldsymbol{W}^2\right) + \operatorname{tr}\left(\boldsymbol{W}^\top \boldsymbol{W}\right) + \operatorname{tr}\left(\boldsymbol{W}\right)^2\right)\delta^2 I \quad \textit{if } \Sigma = I \tag{20}$$

  *3.*
$$\mathbb{E}[\boldsymbol{x}\boldsymbol{r}^\top \boldsymbol{W}^\top \boldsymbol{x}\boldsymbol{x}^\top \boldsymbol{W}\boldsymbol{r}\boldsymbol{x}^\top] = 2\delta^2 \Sigma \boldsymbol{W}\boldsymbol{W}^\top \Sigma + \delta^2 \operatorname{tr}\left(\boldsymbol{W}\boldsymbol{W}^\top \Sigma\right)\Sigma$$
$$= 2\delta^2 \boldsymbol{W}\boldsymbol{W}^\top + \delta^2 \operatorname{tr}(\boldsymbol{W}^\top \boldsymbol{W}) I \quad \textit{if } \Sigma = I \tag{21}$$

  *4.*
$$\mathbb{E}[\boldsymbol{r}\boldsymbol{x}^\top \boldsymbol{W}^\top \boldsymbol{x}\boldsymbol{x}^\top \boldsymbol{W}\boldsymbol{r}\boldsymbol{x}^\top] = \delta^2 \left(\boldsymbol{W}^\top \Sigma \boldsymbol{W}\Sigma + \boldsymbol{W}^\top \Sigma \boldsymbol{W}^\top \Sigma + \boldsymbol{W}^\top \operatorname{tr}(\boldsymbol{W}\Sigma)\Sigma\right)$$
$$= \delta^2 \left(\boldsymbol{W}^\top \boldsymbol{W} + \boldsymbol{W}^\top \boldsymbol{W}^\top + \boldsymbol{W}^\top \operatorname{tr}(\boldsymbol{W})\right) \quad \textit{if } \Sigma = I \tag{22}$$

  *5.*
$$\mathbb{E}[\boldsymbol{r}\boldsymbol{r}^\top \boldsymbol{W}^\top \boldsymbol{x}\boldsymbol{x}^\top \boldsymbol{W}\boldsymbol{x}\boldsymbol{x}^\top] = \mathbb{E}[\boldsymbol{r}\boldsymbol{x}^\top \boldsymbol{W}^\top \boldsymbol{x}\boldsymbol{x}^\top \boldsymbol{W}\boldsymbol{r}\boldsymbol{x}^\top] \tag{23}$$

*Proof.*      1.  We have
$$\mathbb{E}_{\boldsymbol{x},\boldsymbol{r}}[\boldsymbol{r}\boldsymbol{r}^\top \boldsymbol{W}^\top \boldsymbol{x}\boldsymbol{x}^\top \boldsymbol{W}\boldsymbol{r}\boldsymbol{r}^\top] = \mathbb{E}_{\boldsymbol{r}}[\boldsymbol{r}\boldsymbol{r}^\top \boldsymbol{W}^\top \boldsymbol{W}\boldsymbol{r}\boldsymbol{r}^\top]$$
$$= 2\delta^2 I \boldsymbol{W}^\top \Sigma \boldsymbol{W}\delta^2 I + \operatorname{tr}(\boldsymbol{W}^\top \Sigma \boldsymbol{W}\delta^2 I)\delta^2 I$$
$$= 2\delta^4 \boldsymbol{W}^\top \Sigma \boldsymbol{W} + \delta^4 \operatorname{tr}(\boldsymbol{W}^\top \Sigma \boldsymbol{W}) I$$
$$= 2\delta^4 \boldsymbol{W}^\top \boldsymbol{W} + \delta^4 \operatorname{tr}(\boldsymbol{W}^\top \boldsymbol{W}) I \quad \text{if } \Sigma = I \tag{24}$$

where the first step follows from Equation (32).

2.

$$
\begin{aligned}
\mathop{\mathbb{E}}_{\boldsymbol{x},\boldsymbol{r}}\left[\boldsymbol{r}\boldsymbol{x}^\top\boldsymbol{W}^\top\boldsymbol{x}\boldsymbol{x}^\top\boldsymbol{W}\boldsymbol{x}\boldsymbol{r}^\top\right] &= \mathop{\mathbb{E}}_{\boldsymbol{r}}\left[\boldsymbol{r}\mathop{\mathbb{E}}_{\boldsymbol{x}}\left[\boldsymbol{x}^\top\boldsymbol{W}^\top\boldsymbol{x}\boldsymbol{x}^\top\boldsymbol{W}\boldsymbol{x}\right]\boldsymbol{r}^\top\right] \\
&= \left(\mathrm{tr}(\boldsymbol{W}^\top\Sigma\boldsymbol{W}\Sigma) + \mathrm{tr}(\boldsymbol{W}\Sigma\boldsymbol{W}\Sigma) + \mathrm{tr}(\boldsymbol{W}\Sigma)\,\mathrm{tr}(\boldsymbol{W}^\top\Sigma)\right)\delta^2 I \quad (25) \\
&= \left(\mathrm{tr}\left(\boldsymbol{W}^2\right) + \mathrm{tr}\left(\boldsymbol{W}^\top\boldsymbol{W}\right) + \mathrm{tr}\left(\boldsymbol{W}\right)^2\right)\delta^2 I \quad \text{if } \Sigma = I
\end{aligned}
$$

where the first step follows from Equation (34).

3.

$$
\begin{aligned}
\mathbb{E}[\boldsymbol{x}\boldsymbol{r}^\top\boldsymbol{W}^\top\boldsymbol{x}\boldsymbol{x}^\top\boldsymbol{W}\boldsymbol{r}\boldsymbol{x}^\top] &= \mathbb{E}\left[\boldsymbol{x}\mathbb{E}\left[\mathrm{tr}\left(\boldsymbol{r}^\top\boldsymbol{W}^\top\boldsymbol{x}\boldsymbol{x}^\top\boldsymbol{W}\boldsymbol{r}\right)\right]\boldsymbol{x}^\top\right] \\
&= \mathbb{E}\left[\boldsymbol{x}\mathbb{E}\left[\mathrm{tr}\left(\boldsymbol{r}\boldsymbol{r}^\top\boldsymbol{W}^\top\boldsymbol{x}\boldsymbol{x}^\top\boldsymbol{W}\right)\right]\boldsymbol{x}^\top\right] \\
&= \delta^2\,\mathbb{E}\,\boldsymbol{x}\,\mathrm{tr}(\boldsymbol{W}^\top\boldsymbol{x}\boldsymbol{x}^\top\boldsymbol{W})\boldsymbol{x}^\top \\
&= \delta^2\,\mathbb{E}\,\boldsymbol{x}\,\mathrm{tr}(\boldsymbol{x}^\top\boldsymbol{W}\boldsymbol{W}^\top\boldsymbol{x})\boldsymbol{x}^\top \quad (26) \\
&= \delta^2\,\mathbb{E}\,\boldsymbol{x}\boldsymbol{x}^\top\boldsymbol{W}\boldsymbol{W}^\top\boldsymbol{x}\boldsymbol{x}^\top \\
&= \delta^2\left(2\Sigma\boldsymbol{W}\boldsymbol{W}^\top\Sigma + \mathrm{tr}(\boldsymbol{W}\boldsymbol{W}^\top\Sigma)\Sigma\right) \\
&= 2\delta^2\boldsymbol{W}\boldsymbol{W}^\top + \delta^2\,\mathrm{tr}(\boldsymbol{W}^\top\boldsymbol{W})I \quad \text{if } \Sigma = I
\end{aligned}
$$

where the first three steps follow from the cyclic property of trace and the last step follows from Equation (32).

4.

$$
\begin{aligned}
\mathbb{E}[\boldsymbol{r}\boldsymbol{x}^\top\boldsymbol{W}\boldsymbol{x}\boldsymbol{x}^\top\boldsymbol{W}\boldsymbol{r}\boldsymbol{x}^\top] &= \mathbb{E}\left[\boldsymbol{r}(\boldsymbol{x}^\top\boldsymbol{W}\boldsymbol{r})^\top\boldsymbol{x}^\top\boldsymbol{W}\boldsymbol{x}\boldsymbol{x}^\top\right] \\
&= \mathbb{E}\left[\boldsymbol{r}\boldsymbol{r}^\top\boldsymbol{W}^\top\boldsymbol{x}\boldsymbol{x}^\top\boldsymbol{W}\boldsymbol{x}\boldsymbol{x}^\top\right] \\
&= \delta^2\,\mathbb{E}\left[\boldsymbol{W}^\top\boldsymbol{x}\boldsymbol{x}^\top\boldsymbol{W}\boldsymbol{x}\boldsymbol{x}^\top\right] \\
&= \delta^2\boldsymbol{W}^\top\left(\Sigma\left(\boldsymbol{W}+\boldsymbol{W}^T\right)\Sigma + \mathrm{tr}\left(\boldsymbol{W}\Sigma\right)\Sigma\right) \\
&= \delta^2\left(\boldsymbol{W}^\top\Sigma\boldsymbol{W}\Sigma + \boldsymbol{W}^\top\Sigma\boldsymbol{W}^\top\Sigma + \boldsymbol{W}^\top\,\mathrm{tr}(\boldsymbol{W}\Sigma)\Sigma\right) \\
&= \delta^2\left(\boldsymbol{W}^\top\boldsymbol{W} + \boldsymbol{W}^\top\boldsymbol{W}^\top + \boldsymbol{W}^\top\,\mathrm{tr}(\boldsymbol{W})\right) \quad \text{if } \Sigma = I
\end{aligned}
$$

where the first step follows from $\boldsymbol{x}^\top\boldsymbol{W}\boldsymbol{r}$ being scalar, and the third step follows from Equation (32).

5.

$$
\begin{aligned}
\mathbb{E}[\boldsymbol{r}\boldsymbol{r}^\top\boldsymbol{W}^\top\boldsymbol{x}\boldsymbol{x}^\top\boldsymbol{W}\boldsymbol{x}\boldsymbol{x}^\top] &= \mathbb{E}[\boldsymbol{r}\boldsymbol{x}^\top\boldsymbol{W}\boldsymbol{x}\boldsymbol{r}^\top\boldsymbol{W}^\top\boldsymbol{x}\boldsymbol{x}^\top] \\
&= \mathbb{E}[\boldsymbol{r}\boldsymbol{x}^\top\boldsymbol{W}\boldsymbol{x}\boldsymbol{x}^\top\boldsymbol{W}\boldsymbol{r}\boldsymbol{x}^\top] \quad (27)
\end{aligned}
$$

It follows from changing the order and transposing a scalar.

$\square$

**Lemma 3** (Expectation of 6th-Order Gaussian Monomial). *If $\boldsymbol{x} \sim \mathcal{N}(0, I)$, then*

$$
\begin{aligned}
\mathbb{E}[\boldsymbol{x}\boldsymbol{x}^\top A\boldsymbol{x}\boldsymbol{x}^\top B\boldsymbol{x}\boldsymbol{x}^\top] &= \Sigma A\Sigma B\Sigma + \Sigma A\Sigma B^\top\Sigma + \Sigma A^\top\Sigma B\Sigma + \Sigma A^\top\Sigma B^\top\Sigma \\
&\quad + \Sigma B^\top\Sigma A\Sigma + \Sigma B^\top\Sigma A^\top\Sigma + \Sigma B\Sigma A\Sigma + \Sigma B\Sigma A^\top\Sigma \\
&\quad + \mathrm{tr}(\Sigma B)\left(\Sigma A\Sigma + \Sigma A^\top\Sigma\right) + \mathrm{tr}(\Sigma A)\left(\Sigma B\Sigma + \Sigma B^\top\Sigma\right) \\
&\quad + \mathrm{tr}(\Sigma A)\,\mathrm{tr}(\Sigma B)\Sigma + \mathrm{tr}\left(\Sigma A\Sigma B^\top\right)\Sigma + \mathrm{tr}(\Sigma A\Sigma B)\Sigma \quad (28) \\
&= AB + AB^\top + A^\top B + A^\top B^\top + B^\top A + B^\top A^\top + BA + BA^\top \\
&\quad + \mathrm{tr}(B)A + \mathrm{tr}(B)A^\top + \mathrm{tr}(A)B + \mathrm{tr}(A)B^\top \\
&\quad + \mathrm{tr}(A)\,\mathrm{tr}(B)I + \mathrm{tr}\left(AB^\top\right)I + \mathrm{tr}(AB)I \quad \textit{if } \Sigma = I
\end{aligned}
$$

$$\mathbb{E}[\boldsymbol{x}\boldsymbol{x}^\top \boldsymbol{W}^\top \boldsymbol{x}\boldsymbol{x}^\top \boldsymbol{W}\boldsymbol{x}\boldsymbol{x}^\top] = \mathbb{E}[\boldsymbol{x}\boldsymbol{x}^\top \boldsymbol{W}\boldsymbol{x}\boldsymbol{x}^\top \boldsymbol{W}\boldsymbol{x}\boldsymbol{x}^\top]$$

$$= 2\left(\Sigma \boldsymbol{W}\Sigma \boldsymbol{W}\Sigma + \Sigma \boldsymbol{W}\Sigma \boldsymbol{W}^\top \Sigma + \Sigma \boldsymbol{W}^\top \Sigma \boldsymbol{W}\Sigma + \Sigma \boldsymbol{W}^\top \Sigma \boldsymbol{W}^\top \Sigma\right)$$

$$+ 2\operatorname{tr}(\Sigma \boldsymbol{W})\left(\Sigma \boldsymbol{W}\Sigma + \Sigma \boldsymbol{W}^\top \Sigma\right)$$

$$+ \left(\operatorname{tr}(\Sigma \boldsymbol{W})^2 + \operatorname{tr}(\Sigma \boldsymbol{W}\Sigma \boldsymbol{W}) + \operatorname{tr}\left(\Sigma \boldsymbol{W}\Sigma \boldsymbol{W}^\top\right)\right)\Sigma \qquad (29)$$

$$= 2\left(\boldsymbol{W}^2 + \boldsymbol{W}^\top \boldsymbol{W}^\top + \boldsymbol{W}^\top \boldsymbol{W} + \boldsymbol{W}\boldsymbol{W}^\top + \operatorname{tr}(\boldsymbol{W})\boldsymbol{W} + \operatorname{tr}(\boldsymbol{W})\boldsymbol{W}^\top\right)$$

$$+ \operatorname{tr}(\boldsymbol{W})^2 I + \operatorname{tr}\left(\boldsymbol{W}^2\right)I + \operatorname{tr}\left(\boldsymbol{W}^\top \boldsymbol{W}\right)I \quad \text{if } \Sigma = I$$

*Proof.* Let $T := \mathbb{E}[\boldsymbol{x}\boldsymbol{x}^\top A\boldsymbol{x}\boldsymbol{x}^\top B\boldsymbol{x}\boldsymbol{x}^\top]$. Then, let's consider one scalar entry:

$$T_{ij} = \mathbb{E}\left[\sum_{k,\ell,m,n} x_i x_k A_{k\ell} x_\ell x_m B_{mn} x_n x_j\right] = \sum_{k,\ell,m,n} A_{k\ell}B_{mn} \cdot \mathbb{E}\left[x_i x_k x_\ell x_m x_n x_j\right] \qquad (30)$$

We now need to compute the 6th-order central moment of standard normal variables. This can be computed using the Isserlis' Theorem (Isserlis, 1918):

$$\mathbb{E}[x_1 \cdots x_s] = \sum_{p \in P_s^2} \prod_{(i,j) \in p} \mathbb{E}[x_i x_j] \qquad (31)$$

where $P_s^2$ stands for all distinct ways of partitioning $\{1, \ldots, s\}$ into pairs $i, j$ (perfect matching), and the product is over the pairs contained in $p$.

We note that the number of perfect matching for $s$ examples is given as:

$$\#\text{perfect matching} = \frac{s!}{2^{s/2}(s/2)!}$$

where $2^{s/2}$ is for ignoring the ordering inside pairs and $(s/2)!$ is for ignoring the ordering between pairs.

We note that there are $\frac{6!}{2^3 \cdot 3!} = 15$ distinct partitions for the 6-th order product of Gaussian random variable. Suppose $(x_a, x_b), (x_c, x_d), (x_e, x_f)$ is a valid pairing, then:

$$\mathbb{E}[x_a x_b]\,\mathbb{E}[x_c x_d]\,\mathbb{E}[x_e x_f] = \Sigma_{ab} \cdot \Sigma_{cd} \cdot \Sigma_{ef}$$

Here, we will discuss the result for all 15 distinct pairings:

1. $(i,k)(\ell,m)(n,j)$

$$\sum_{k,\ell,m,n} A_{k\ell}B_{mn}\,\Sigma_{ik}\Sigma_{\ell m}\Sigma_{nj} = \left[\Sigma A \Sigma B \Sigma\right]_{ij}$$

2. $(i,k)(\ell,n)(m,j)$

$$\sum_{k,\ell,m,n} A_{k\ell}B_{mn}\,\Sigma_{ik}\Sigma_{\ell n}\Sigma_{mj} = \left[\Sigma A \Sigma B^\top \Sigma\right]_{ij}$$

3. $(i,k)(\ell,j)(m,n)$

$$\sum_{k,\ell,m,n} A_{k\ell}B_{mn}\,\Sigma_{ik}\Sigma_{\ell j}\Sigma_{mn} = \operatorname{tr}(\Sigma B)\left[\Sigma A \Sigma\right]_{ij}$$

4. $(i,\ell)(k,m)(n,j)$

$$\sum_{k,\ell,m,n} A_{k\ell}B_{mn}\,\Sigma_{i\ell}\Sigma_{km}\Sigma_{nj} = \left[\Sigma A^\top \Sigma B \Sigma\right]_{ij}$$

5. $(i,\ell)(k,n)(m,j)$

$$\sum_{k,\ell,m,n} A_{k\ell} B_{mn} \, \Sigma_{i\ell} \Sigma_{kn} \Sigma_{mj} = \left[\Sigma A^\top \Sigma B^\top \Sigma\right]_{ij}$$

6. $(i,\ell)(k,j)(m,n)$

$$\sum_{k,\ell,m,n} A_{k\ell} B_{mn} \, \Sigma_{i\ell} \Sigma_{kj} \Sigma_{mn} = \mathrm{tr}(\Sigma B) \left[\Sigma A^\top \Sigma\right]_{ij}$$

7. $(i,m)(k,\ell)(n,j)$

$$\sum_{k,\ell,m,n} A_{k\ell} B_{mn} \, \Sigma_{im} \Sigma_{k\ell} \Sigma_{nj} = \mathrm{tr}(\Sigma A) \left[\Sigma B^\top \Sigma\right]_{ij}$$

8. $(i,m)(k,n)(\ell,j)$

$$\sum_{k,\ell,m,n} A_{k\ell} B_{mn} \, \Sigma_{im} \Sigma_{kn} \Sigma_{\ell j} = \left[\Sigma B \Sigma A \Sigma\right]_{ij}$$

9. $(i,m)(k,j)(\ell,n)$

$$\sum_{k,\ell,m,n} A_{k\ell} B_{mn} \, \Sigma_{im} \Sigma_{kj} \Sigma_{\ell n} = \left[\Sigma B \Sigma A^\top \Sigma\right]_{ij}$$

10. $(i,n)(k,\ell)(m,j)$

$$\sum_{k,\ell,m,n} A_{k\ell} B_{mn} \, \Sigma_{in} \Sigma_{k\ell} \Sigma_{mj} = \mathrm{tr}(\Sigma A) \left[\Sigma B \Sigma\right]_{ij}$$

11. $(i,n)(k,m)(\ell,j)$

$$\sum_{k,\ell,m,n} A_{k\ell} B_{mn} \, \Sigma_{in} \Sigma_{km} \Sigma_{\ell j} = \left[\Sigma B^\top \Sigma A \Sigma\right]_{ij}$$

12. $(i,n)(k,j)(\ell,m)$

$$\sum_{k,\ell,m,n} A_{k\ell} B_{mn} \, \Sigma_{in} \Sigma_{kj} \Sigma_{\ell m} = \left[\Sigma B^\top \Sigma A^\top \Sigma\right]_{ij}$$

13. $(i,j)(k,\ell)(m,n)$

$$\sum_{k,\ell,m,n} A_{k\ell} B_{mn} \, \Sigma_{ij} \Sigma_{k\ell} \Sigma_{mn} = \mathrm{tr}(\Sigma A) \, \mathrm{tr}(\Sigma B) \, \Sigma_{ij}$$

14. $(i,j)(k,m)(\ell,n)$

$$\sum_{k,\ell,m,n} A_{k\ell} B_{mn} \, \Sigma_{ij} \Sigma_{km} \Sigma_{\ell n} = \mathrm{tr}(\Sigma A \Sigma B^\top) \, \Sigma_{ij}$$

15. $(i,j)(k,n)(\ell,m)$

$$\sum_{k,\ell,m,n} A_{k\ell} B_{mn} \, \Sigma_{ij} \Sigma_{kn} \Sigma_{\ell m} = \mathrm{tr}(\Sigma A \Sigma B) \, \Sigma_{ij}$$

Summing up all of these 15 terms together and recombining the element-wise terms into the matrix form, we obtain Eq. (28). Then, we plug in $A = \boldsymbol{W}, B = \boldsymbol{W}^\top$, we obtain Eq. (29). $\qquad \square$

**Lemma 4** (Expectation of 4th-Order Gaussian Monomial)**.** *Let* $\boldsymbol{x}, \boldsymbol{x}_1, \ldots, \boldsymbol{x}_m \sim \mathcal{N}(0, I)$ *and* $\boldsymbol{X} = [\boldsymbol{x}_1^\top; \ldots; \boldsymbol{x}_m^\top]$. *Then, we have*

$$
\begin{aligned}
\mathbb{E}\, \boldsymbol{x}\boldsymbol{x}^\top \boldsymbol{W} \boldsymbol{x}\boldsymbol{x}^\top &= \Sigma \left(\boldsymbol{W} + \boldsymbol{W}^T\right) \Sigma + \mathrm{tr}\left(\boldsymbol{W}\Sigma\right) \Sigma \\
&= \boldsymbol{W} + \boldsymbol{W}^\top + \mathrm{tr}(\boldsymbol{W})I \quad \textit{if } \Sigma = I \\
&= 2\boldsymbol{W} + \mathrm{tr}(\boldsymbol{W})I \qquad \textit{if } \boldsymbol{W} \textit{ is also symmetric}
\end{aligned}
\tag{32}
$$

*and*

$$
\begin{aligned}
\mathbb{E}\, \boldsymbol{X}^\top \boldsymbol{X} \boldsymbol{W} \boldsymbol{X}^\top \boldsymbol{X} &= m^2 \Sigma \boldsymbol{W} \Sigma + m \Sigma \boldsymbol{W}^\top \Sigma + m \operatorname{tr}(\boldsymbol{W}\Sigma)\Sigma \\
&= m^2 \boldsymbol{W} + m \boldsymbol{W}^\top + m \operatorname{tr}(\boldsymbol{W})I \quad \textit{if } \Sigma = I \\
&= m(m+1)\boldsymbol{W} + m \operatorname{tr}(\boldsymbol{W})I \quad \textit{if } \boldsymbol{W} \text{ is also symmetric}
\end{aligned}
\tag{33}
$$

$$
\mathbb{E}\, \boldsymbol{x}^\top A \boldsymbol{x} \boldsymbol{x}^\top B \boldsymbol{x} = \operatorname{tr}\left(A \Sigma \left(B + B^\top\right) \Sigma\right) + \operatorname{tr}(A\Sigma)\operatorname{tr}(B\Sigma)
\tag{34}
$$

*If* $A = \boldsymbol{W}^\top, B = \boldsymbol{W}$, *then*

$$
\begin{aligned}
\mathbb{E}\, \boldsymbol{x}^\top \boldsymbol{W}^\top \boldsymbol{x} \boldsymbol{x}^\top \boldsymbol{W} \boldsymbol{x} &= \mathbb{E}\, \boldsymbol{x}^\top \boldsymbol{W} \boldsymbol{x} \boldsymbol{x}^\top \boldsymbol{W} \boldsymbol{x} \\
&= \operatorname{tr}(\boldsymbol{W}^\top \Sigma \boldsymbol{W} \Sigma) + \operatorname{tr}(\boldsymbol{W}\Sigma\boldsymbol{W}\Sigma) + \operatorname{tr}(\boldsymbol{W}\Sigma)\operatorname{tr}(\boldsymbol{W}^\top\Sigma) \\
&= \operatorname{tr}(\boldsymbol{W}^\top \boldsymbol{W}) + \operatorname{tr}(\boldsymbol{W}^2) + \operatorname{tr}(\boldsymbol{W})^2 \quad \textit{if } \Sigma = I
\end{aligned}
\tag{35}
$$

*Proof.* Equation (32) follows from section 8.2.4 of (Petersen et al., 2008) by plugging in mean 0 and variance $\Sigma$.

$$
\begin{aligned}
\mathbb{E}\, \boldsymbol{X}^\top \boldsymbol{X} \boldsymbol{W} \boldsymbol{X}^\top \boldsymbol{X} &= \sum_i \boldsymbol{x}_i \boldsymbol{x}_i^\top \boldsymbol{W} \boldsymbol{x}_i \boldsymbol{x}_i^\top + \sum_{i \neq j} \boldsymbol{x}_i \boldsymbol{x}_i^\top \boldsymbol{W} \boldsymbol{x}_j \boldsymbol{x}_j^\top \\
&= m \left(\Sigma \left(\boldsymbol{W} + \boldsymbol{W}^T\right) \Sigma + \operatorname{tr}(\boldsymbol{W}\Sigma)\Sigma\right) + m(m-1)\Sigma \boldsymbol{W} \Sigma \\
&= m \left(\boldsymbol{W} + \boldsymbol{W}^\top + \operatorname{tr}(\boldsymbol{W})I\right) + m(m-1)\boldsymbol{W} \quad \textit{if } \Sigma = I \\
&= m^2 \boldsymbol{W} + m \boldsymbol{W}^\top + m \operatorname{tr}(\boldsymbol{W})I \quad \textit{if } \boldsymbol{W} \text{ is also symmetric} \\
&= m(m+1)\boldsymbol{W} + m \operatorname{tr}(\boldsymbol{W})I
\end{aligned}
\tag{36}
$$

where the second step follows from plugging in Equation (32).

Equation (34) follows from section 8.2.4 of (Petersen et al., 2008) by plugging in mean 0 and variance $\Sigma$.

$\square$

## B ADDITIONAL PROOF FOR RAG

Here, we provide an overview of the organization of the proof. First, we consider the uniform retrieval noise scenario, and compute the population loss for generic $\boldsymbol{W}$ in Theorem 1. Then, we plug in the special case $\boldsymbol{W}^*$ (isotropic pretrained weight), and provide a closed-form loss in Proposition 3. Then, we analyze its finite sample complexity in Proposition 1 and the optimal RAG examples in relation to ICL examples in Proposition 2.

Later on, we provide an finite sample complexity analysis for non-uniform retrieval noise, Theorem 2 for Distance Proportional Noise, and Theorem 3 for Distance-Weighted Mixture Noise.

### B.1 UNIFORM RETRIEVAL NOISE

**Theorem 4** (Generalization of Theorem 1). *Under Assumption 1, 2, 3, the population loss of the linear self-attention predictor* $\hat{y}_q = \boldsymbol{x}_q^\top \boldsymbol{W} \boldsymbol{X}^\top \boldsymbol{y}$ *satisfies*

$$
\mathcal{L}_{tt+rag}(\boldsymbol{W}) = \underbrace{\mathbb{E}\left(\mathbb{E}\left(\hat{y}_q\right) - \hat{y}_q\right)^2}_{:=\text{err}_{variance}(\boldsymbol{W})} + \underbrace{\mathbb{E}\left(\mathbb{E}\left(\hat{y}_q\right) - \mathbb{E}\left(y_q\right)\right)^2}_{:=\text{err}_{bias}(\boldsymbol{W})} + \underbrace{\sigma^2}_{\textit{irreducible noise}}
\tag{37}
$$

*For $\Sigma = I$, we have*

$$\text{err}_{variance}(\boldsymbol{W}) = \left[m\sigma^2 + \left(1+\delta^2\right)n\sigma^2_{rag}\right]\text{tr}(\boldsymbol{W}^\top\boldsymbol{W}) + n\sigma^2_{rag}\text{tr}(\boldsymbol{W}^2) + n\sigma^2_{rag}\text{tr}(\boldsymbol{W})^2$$

$$\text{err}_{bias}(\boldsymbol{W}) = \beta_{\text{tt}}^\top\left[I - (n\delta^2 + 2n + m)(\boldsymbol{W} + \boldsymbol{W}^\top) - 2n\,\text{tr}(\boldsymbol{W})I + M_4\right]\beta_{\text{tt}}$$

$$= \beta_{\text{tt}}^\top\Big[I - (n\delta^2 + 2n + m)(\boldsymbol{W} + \boldsymbol{W}^\top) - 2n\,\text{tr}(\boldsymbol{W})I$$

$$+ \underbrace{\left[n^2\left(2+\delta^2\right) + n\left(m+\delta^2\right)\right]}_{:=c_1}\left(\boldsymbol{W}^2 + \left(\boldsymbol{W}^2\right)^\top\right) + \underbrace{2n(n+\delta^2)}_{:=c_2}\boldsymbol{W}\boldsymbol{W}^\top$$

$$+ \underbrace{\left[m^2 + m + mn\left(2+2\delta^2\right) + n^2\left(2+2\delta^2+\delta^4\right) + n\left(2\delta^2+\delta^4\right)\right]}_{:=c_3}\boldsymbol{W}^\top\boldsymbol{W} \qquad (38)$$

$$+ \underbrace{\left[n^2\left(2+\delta^2\right) + n\left(m+\delta^2\right)\right]}_{:=c_4,\ c_4=c_1}\left(\text{tr}(\boldsymbol{W})\left(\boldsymbol{W} + \boldsymbol{W}^\top\right)\right)$$

$$+ \underbrace{\left[n^2 + n\delta^2\right]}_{:=c_5}\left(\text{tr}(\boldsymbol{W})^2 + \text{tr}\left(\boldsymbol{W}^2\right)\right)I + \underbrace{\left[m + n^2 + n\left(2\delta^2+\delta^4\right)\right]}_{:=c_6}\text{tr}\left(\boldsymbol{W}^\top\boldsymbol{W}\right)I\Big]\beta_{\text{tt}}$$

*For general $\Sigma$, we have*

$$\text{err}_{variance}(\boldsymbol{W}) = \left(n\sigma^2_{rag} + m\sigma^2\right)\text{tr}\left(\boldsymbol{W}^\top\Sigma\boldsymbol{W}\Sigma\right)$$

$$+ n\sigma^2_{rag}\left[\text{tr}(\boldsymbol{W}\Sigma\boldsymbol{W}\Sigma) + \text{tr}(\boldsymbol{W}\Sigma)\text{tr}\left(\boldsymbol{W}^\top\Sigma\right) + \delta^2\text{tr}\left(\boldsymbol{W}^\top\Sigma\boldsymbol{W}\right)\right]$$

$$\text{err}_{bias}(\boldsymbol{W}) = \beta_{\text{tt}}^\top\left[\Sigma - (m+2n)\Sigma\left(\boldsymbol{W}+\boldsymbol{W}^\top\right)\Sigma - n\delta^2\left(\Sigma\boldsymbol{W}+\boldsymbol{W}^\top\Sigma\right) - 2n\,\text{tr}(\boldsymbol{W}\Sigma)\Sigma + M_4\right]\beta_{\text{tt}}$$

$$M_4 = m(m+1)\Sigma\boldsymbol{W}^\top\boldsymbol{W}\Sigma + m\,\text{tr}\left(\boldsymbol{W}^\top\boldsymbol{W}\Sigma\right)\Sigma + (2n^2+2nm)\Sigma\boldsymbol{W}^\top\Sigma\boldsymbol{W}\Sigma$$

$$+ (2n^2+nm)\left(\Sigma\boldsymbol{W}^\top\Sigma\boldsymbol{W}^\top\Sigma + \Sigma\boldsymbol{W}\Sigma\boldsymbol{W}\Sigma\right) + 2n^2\Sigma\boldsymbol{W}\Sigma\boldsymbol{W}^\top\Sigma$$

$$+ (2n^2+nm)\,\text{tr}(\Sigma\boldsymbol{W})\left(\Sigma\boldsymbol{W}\Sigma + \Sigma\boldsymbol{W}^\top\Sigma\right) + n^2\left(\text{tr}(\Sigma\boldsymbol{W})^2 + \text{tr}(\Sigma\boldsymbol{W}\Sigma\boldsymbol{W}) + \text{tr}(\Sigma\boldsymbol{W}^\top\Sigma\boldsymbol{W})\right)\Sigma$$

$$+ \delta^2\left((n^2+n+nm)\left(\boldsymbol{W}^\top\Sigma\boldsymbol{W}\Sigma + \Sigma\boldsymbol{W}^\top\Sigma\boldsymbol{W}\right) + (n^2+n)\left(\boldsymbol{W}^\top\Sigma\boldsymbol{W}^\top\Sigma + \Sigma\boldsymbol{W}\Sigma\boldsymbol{W}\right)\right)$$

$$+ \delta^2\left((n^2+n)\,\text{tr}(\Sigma\boldsymbol{W})\left(\boldsymbol{W}^\top\Sigma + \Sigma\boldsymbol{W}\right) + 2n\,\Sigma\boldsymbol{W}\boldsymbol{W}^\top\Sigma + n\,\text{tr}(\boldsymbol{W}^\top\Sigma\boldsymbol{W})\Sigma\right)$$

$$+ \delta^2\left(n\left(\text{tr}(\Sigma\boldsymbol{W}^\top\Sigma\boldsymbol{W}) + \text{tr}(\Sigma\boldsymbol{W}\Sigma\boldsymbol{W}) + \text{tr}(\Sigma\boldsymbol{W})^2\right)I\right)$$

$$+ \delta^4\left((n^2+n)\,\boldsymbol{W}^\top\Sigma\boldsymbol{W} + n\,\text{tr}(\boldsymbol{W}^\top\Sigma\boldsymbol{W})I\right). \qquad (39)$$

*Proof.* For computational convenience, I will define the following quantities for Gram matrix: $\boldsymbol{G}_0 = \boldsymbol{X}_{\text{icl}}^\top\boldsymbol{X}_{\text{icl}}$, $\boldsymbol{G}_i := (\boldsymbol{x}_q + \boldsymbol{r}_i)(\boldsymbol{x}_q + \boldsymbol{r}_i)^\top$, and $\boldsymbol{G} := \boldsymbol{G}_0 + \sum_{i\in[n]}\boldsymbol{G}_i$.

We write down the error explicitly:

$$y_q - \boldsymbol{x}_q^\top\boldsymbol{W}\boldsymbol{X}^\top\boldsymbol{y} = \boldsymbol{x}_q^\top\beta_{\text{tt}} + \epsilon_q - \boldsymbol{x}_q^\top\boldsymbol{W}\boldsymbol{X}^\top\boldsymbol{X}\beta_{\text{tt}} - \boldsymbol{x}_q^\top\boldsymbol{W}\boldsymbol{X}^\top\boldsymbol{\epsilon}$$

$$= \boldsymbol{x}_q^\top\left(\beta_{\text{tt}} - \boldsymbol{W}\boldsymbol{G}\beta_{\text{tt}}\right) - \boldsymbol{x}_q^\top\boldsymbol{W}\boldsymbol{X}^\top\boldsymbol{\epsilon} + \epsilon_q \qquad (40)$$

$$= \boldsymbol{x}_q^\top\left(I - \boldsymbol{W}\boldsymbol{G}\right)\beta_{\text{tt}} - \boldsymbol{x}_q^\top\boldsymbol{W}\boldsymbol{X}^\top\boldsymbol{\epsilon} + \epsilon_q$$

Therefore, the population loss is equal to:

$$\mathcal{L}_{\text{tt+rag}}(\boldsymbol{W}) = \mathop{\mathbb{E}}_{(\boldsymbol{x}_q,y_q),(\boldsymbol{X},\boldsymbol{y}),\boldsymbol{\epsilon}}\left[\left(\boldsymbol{x}_q^\top\left(I - \boldsymbol{W}\boldsymbol{G}\right)\beta_{\text{tt}} - \boldsymbol{x}_q^\top\boldsymbol{W}\boldsymbol{X}^\top\boldsymbol{\epsilon}\right)^2\right] + \sigma^2$$

We note that both $\epsilon_{\mathrm{icl}}$ and $\epsilon_{\mathrm{rag}}$ are independent of $\boldsymbol{x}_q, \boldsymbol{X}$ (including $\boldsymbol{r}$), and $\mathbb{E}[\boldsymbol{\epsilon}] = 0$.

$$\mathbb{E}_{\boldsymbol{\epsilon}}\left[ -2\left( \boldsymbol{x}_q^\top (I - \boldsymbol{W}\boldsymbol{G}) \beta_{\mathrm{tt}} \right)\left( \boldsymbol{x}_q^\top \boldsymbol{W}\boldsymbol{X}^\top \boldsymbol{\epsilon} \right) \right] = 0$$

And therefore, we have the following loss decomposition:

$$\mathcal{L}_{\mathrm{tt+rag}}(\boldsymbol{W}) = \mathbb{E}_{\boldsymbol{x}_q, \boldsymbol{X}, \boldsymbol{\epsilon}}\left[ \left( \boldsymbol{x}_q^\top \boldsymbol{W}\boldsymbol{X}^\top \boldsymbol{\epsilon} \right)^2 \right] + \mathbb{E}_{\boldsymbol{x}_q, \boldsymbol{X}}\left[ \left( \boldsymbol{x}_q^\top (I - \boldsymbol{W}\boldsymbol{G}) \beta_{\mathrm{tt}} \right)^2 \right] + \sigma^2 \tag{41}$$

Then, we compute the mean of the prediction and the label:

$$\mathbb{E}_{\epsilon_q} y_q = \mathbb{E}_{\epsilon_q}\left( \boldsymbol{x}_q^\top \beta_{\mathrm{tt}} + \epsilon_q \right) = \boldsymbol{x}_q^\top \beta_{\mathrm{tt}}$$

$$\mathbb{E}_{\boldsymbol{\epsilon}} \hat{y}_q = \mathbb{E}\, \boldsymbol{x}_q^\top \boldsymbol{W}\boldsymbol{X}^\top \boldsymbol{y}$$

$$= \mathbb{E}\, \boldsymbol{x}_q^\top \boldsymbol{W}\boldsymbol{X}^\top (\boldsymbol{X}\beta_{\mathrm{tt}} + \boldsymbol{\epsilon})$$

$$= \mathbb{E}\, \boldsymbol{x}_q^\top \boldsymbol{W}\boldsymbol{G}\beta_{\mathrm{tt}}$$

And further, we have

$$\mathbb{E}_{\epsilon_q}\left( y_q - \mathbb{E}_{\epsilon_q} y_q \right)^2 = \mathbb{E}_{\epsilon_q}\left( \boldsymbol{x}_q^\top \beta_{\mathrm{tt}} + \epsilon_q - \boldsymbol{x}_q^\top \beta_{\mathrm{tt}} \right)^2 = \mathbb{E}_{\epsilon_q} \epsilon_q^2 = \sigma^2$$

$$\left( \hat{y}_q - \mathbb{E}_{\boldsymbol{\epsilon}} \hat{y}_q \right)^2 = \left( \boldsymbol{x}_q^\top \boldsymbol{W}\boldsymbol{X}^\top (\boldsymbol{X}\beta_{\mathrm{tt}} + \boldsymbol{\epsilon}) - \boldsymbol{x}_q^\top \boldsymbol{W}\boldsymbol{X}^\top \boldsymbol{X}\beta_{\mathrm{tt}} \right)^2 = \left( \boldsymbol{x}_q^\top \boldsymbol{W}\boldsymbol{X}^\top \boldsymbol{\epsilon} \right)^2 \tag{42}$$

$$\left( \mathbb{E}_{\epsilon_q}(y_q) - \mathbb{E}_{\boldsymbol{\epsilon}} \hat{y}_q \right)^2 = \left( \boldsymbol{x}_q^\top \beta_{\mathrm{tt}} - \boldsymbol{x}_q^\top \boldsymbol{W}\boldsymbol{G}\beta_{\mathrm{tt}} \right)^2 = \left( \boldsymbol{x}_q^\top (I - \boldsymbol{W}\boldsymbol{G})\beta_{\mathrm{tt}} \right)^2$$

If we plug Equation (42) into the loss decomposition Equation (41), we have

$$\mathcal{L}_{\mathrm{tt+rag}}(\boldsymbol{W}) = \mathbb{E}_{\boldsymbol{x}_q, \boldsymbol{X}, \boldsymbol{\epsilon}}\left[ \left( \boldsymbol{x}_q^\top \boldsymbol{W}\boldsymbol{X}^\top \boldsymbol{\epsilon} \right)^2 \right] + \mathbb{E}_{\boldsymbol{x}_q, \boldsymbol{X}}\left[ \left( \boldsymbol{x}_q^\top (I - \boldsymbol{W}\boldsymbol{G}) \beta_{\mathrm{tt}} \right)^2 \right] + \sigma^2$$

$$= \underbrace{\mathbb{E}\left( \mathbb{E}_{\boldsymbol{\epsilon}}(\hat{y}_q) - \hat{y}_q \right)^2}_{:=\mathrm{err}_{\mathrm{variance}}(\boldsymbol{W})} + \underbrace{\mathbb{E}\left( \mathbb{E}_{\epsilon_q}(\hat{y}_q) - \mathbb{E}_{\boldsymbol{\epsilon}}(y_q) \right)^2}_{:=\mathrm{err}_{\mathrm{bias}}(\boldsymbol{W})} + \underbrace{\mathbb{E}\left( y_q - \mathbb{E}_{\epsilon_q} y_q \right)^2}_{\substack{=\sigma^2 \\ \text{(irreducible noise)}}} \tag{43}$$

and we can obtain the bias-variance tradeoff as given in Equation (37).

**Compute** $\mathbb{E}_{\boldsymbol{x}_q, \boldsymbol{X}, \boldsymbol{r}, \boldsymbol{\epsilon}}\left[ \left( \boldsymbol{x}_q^\top \boldsymbol{W}\boldsymbol{X}^\top \boldsymbol{\epsilon} \right)^2 \right]$  First, we let

$$z := \boldsymbol{x}_q^\top \boldsymbol{W}\boldsymbol{X}^\top \boldsymbol{\epsilon} = \sum_{i=1}^{m+n} \boldsymbol{x}_q^\top \boldsymbol{W}\boldsymbol{x}_i \cdot \epsilon_i$$

Then,

$$z^2 = \sum_{i,j=1}^{m+n} (\boldsymbol{x}_q^\top \boldsymbol{W}\boldsymbol{x}_i)(\boldsymbol{x}_q^\top \boldsymbol{W}\boldsymbol{x}_j)\epsilon_i\epsilon_j = \sum_{i,j=1}^{m+n} (\boldsymbol{x}_i^\top \boldsymbol{W}^\top \boldsymbol{x}_q)(\boldsymbol{x}_q^\top \boldsymbol{W}\boldsymbol{x}_j)\epsilon_i\epsilon_j$$

Taking expectation:

$$\mathbb{E}_{\boldsymbol{\epsilon}}[z^2] = \sum_{i,j=1}^{m+n} (\boldsymbol{x}_i^\top \boldsymbol{W}^\top \boldsymbol{x}_q)(\boldsymbol{x}_q^\top \boldsymbol{W}\boldsymbol{x}_j) \cdot \mathbb{E}[\epsilon_i\epsilon_j]$$

Because the noise terms are independent and zero-mean, we have:

$$\mathbb{E}[\epsilon_i\epsilon_j] = \begin{cases} \sigma^2, & i = j \le m \\ \sigma_{\mathrm{rag}}^2, & i = j > m \\ 0, & i \ne j \end{cases}$$

So only the diagonal terms survive:

$$\mathbb{E}[z^2] = \sum_{i=1}^{m} \sigma^2 \cdot \mathbb{E}\left[(\boldsymbol{x}_q^\top \boldsymbol{W} \boldsymbol{x}_i)^2\right] + \sum_{i=m+1}^{m+n} \sigma_{\text{rag}}^2 \cdot \mathbb{E}\left[(\boldsymbol{x}_q^\top \boldsymbol{W} (\boldsymbol{x}_q + \boldsymbol{r}_{i-m}))^2\right]$$

- **ICL Term:** Since $\boldsymbol{x}_q, \boldsymbol{x}_i \sim \mathcal{N}(0, I)$ and are independent,

$$
\begin{aligned}
\mathbb{E}[(\boldsymbol{x}_q^\top \boldsymbol{W} \boldsymbol{x}_i)^2] &= \mathbb{E}\left[\boldsymbol{x}_i^\top \boldsymbol{W}^\top \boldsymbol{x}_q \boldsymbol{x}_q^\top \boldsymbol{W} \boldsymbol{x}_i\right] \\
&= \mathbb{E}\left[\text{tr}\left(\boldsymbol{W}^\top \boldsymbol{x}_q \boldsymbol{x}_q^\top \boldsymbol{W} \boldsymbol{x}_i \boldsymbol{x}_i^\top\right)\right] \\
&= \text{tr}(\boldsymbol{W}^\top \Sigma \boldsymbol{W} \Sigma) \\
&= \text{tr}(\boldsymbol{W}^\top \boldsymbol{W}) \quad \text{if } \Sigma = I
\end{aligned}
\tag{44}
$$

where the first step follows from the cyclic property of trace, the last step follows from the symmetry of $\boldsymbol{W}$.

$$\Rightarrow \quad \text{ICL contribution} = m \cdot \sigma^2 \cdot \text{tr}(\boldsymbol{W}^\top \Sigma \boldsymbol{W} \Sigma) = m \cdot \sigma^2 \cdot \text{tr}(\boldsymbol{W}^\top \boldsymbol{W}) \quad \text{if } \Sigma = I \tag{45}$$

- **RAG Term:**

Each row in RAG has the form $\boldsymbol{x}_q + \boldsymbol{r}_i$, so:

$$\boldsymbol{x}_q^\top \boldsymbol{W} (\boldsymbol{x}_q + \boldsymbol{r}_i) = \boldsymbol{x}_q^\top \boldsymbol{W} \boldsymbol{x}_q + \boldsymbol{x}_q^\top \boldsymbol{W} \boldsymbol{r}_i$$

Then, we plug in Equation (34) into the RAG term:

$$
\begin{aligned}
\mathbb{E}\left[\left(\boldsymbol{x}_q^\top \boldsymbol{W} (\boldsymbol{x}_q + \boldsymbol{r}_i)\right)^2\right] &= \mathbb{E}[(\boldsymbol{x}_q^\top \boldsymbol{W} \boldsymbol{x}_q)^2] + \mathbb{E}[(\boldsymbol{x}_q^\top \boldsymbol{W} \boldsymbol{r}_i)^2] + 2\,\mathbb{E}[\boldsymbol{x}_q^\top \boldsymbol{W} \boldsymbol{x}_q \cdot \boldsymbol{x}_q^\top \boldsymbol{W} \boldsymbol{r}_i] \\
&= \mathbb{E}[(\boldsymbol{x}_q^\top \boldsymbol{W} \boldsymbol{x}_q)^2] + \mathbb{E}[(\boldsymbol{x}_q^\top \boldsymbol{W} \boldsymbol{r}_i)^2] \\
&= \text{tr}(\boldsymbol{W} \Sigma \boldsymbol{W}^\top \Sigma) + \text{tr}(\boldsymbol{W} \Sigma \boldsymbol{W} \Sigma) + \text{tr}(\boldsymbol{W} \Sigma) \text{tr}(\boldsymbol{W}^\top \Sigma) + \delta^2 \cdot \text{tr}(\boldsymbol{W}^\top \Sigma \boldsymbol{W}) \\
&= \text{tr}(\boldsymbol{W}^\top \boldsymbol{W}) + \text{tr}(\boldsymbol{W}^2) + \delta^2 \text{tr}(\boldsymbol{W}^\top \boldsymbol{W}) + \text{tr}(\boldsymbol{W})^2 \quad \text{if } \Sigma = I
\end{aligned}
\tag{46}
$$

where the second step follows from $\mathbb{E}[\boldsymbol{r}_i] = 0$, the third step follows from the cyclic property of trace and Lemma 4.

$$
\begin{aligned}
\Rightarrow \quad \text{RAG contribution} &= n \cdot \sigma_{\text{rag}}^2 \cdot \left[\text{tr}(\boldsymbol{W} \Sigma \boldsymbol{W}^\top \Sigma) + \text{tr}(\boldsymbol{W} \Sigma \boldsymbol{W} \Sigma) + \text{tr}(\boldsymbol{W} \Sigma) \text{tr}(\boldsymbol{W}^\top \Sigma) + \delta^2 \cdot \text{tr}(\boldsymbol{W}^\top \Sigma \boldsymbol{W})\right] \\
&= n \cdot \sigma_{\text{rag}}^2 \cdot \left[(1 + \delta^2) \text{tr}(\boldsymbol{W}^\top \boldsymbol{W}) + \text{tr}(\boldsymbol{W}^2) + \text{tr}(\boldsymbol{W})^2\right] \quad \text{if } \Sigma = I
\end{aligned}
$$

Thus, we can combine the two terms above and obtain the following:

$$
\begin{aligned}
\mathbb{E}\left[\left(\boldsymbol{x}_q^\top \boldsymbol{W} \boldsymbol{X}^\top \boldsymbol{\epsilon}\right)^2\right] &= \left(n\sigma_{\text{rag}}^2 + m\sigma^2\right) \text{tr}\left(\boldsymbol{W}^\top \Sigma \boldsymbol{W} \Sigma\right) \\
&\quad + n\sigma_{\text{rag}}^2 \left[\text{tr}(\boldsymbol{W} \Sigma \boldsymbol{W} \Sigma) + \text{tr}(\boldsymbol{W} \Sigma) \text{tr}\left(\boldsymbol{W}^\top \Sigma\right) + \delta^2 \text{tr}\left(\boldsymbol{W}^\top \Sigma \boldsymbol{W}\right)\right] \\
&= \left[m\sigma^2 + \left(1 + \delta^2\right) n\sigma_{\text{rag}}^2\right] \text{tr}(\boldsymbol{W}^\top \boldsymbol{W}) + n\sigma_{\text{rag}}^2 \text{tr}(\boldsymbol{W}^2) + n\sigma_{\text{rag}}^2 \text{tr}(\boldsymbol{W})^2 \quad \text{if } \Sigma = I
\end{aligned}
\tag{47}
$$

**Compute** $\mathbb{E}_{\boldsymbol{x}_q, \boldsymbol{X}}\left[(\boldsymbol{x}_q^\top (I - \boldsymbol{W} \boldsymbol{G}) \beta_{\text{tt}})^2\right]$ First, we can expand the expectation and decompose the inner terms into 4 terms:

$$
\begin{aligned}
\mathbb{E}_{\boldsymbol{x}_q, \boldsymbol{X}}\left[(I - \boldsymbol{W} \boldsymbol{G})^\top \boldsymbol{x}_q \boldsymbol{x}_q^\top (I - \boldsymbol{W} \boldsymbol{G})\right] &= \mathbb{E}_{\boldsymbol{x}_q, \boldsymbol{X}} \left(I - \boldsymbol{G} \boldsymbol{W}^\top\right) \boldsymbol{x}_q \boldsymbol{x}_q^\top (I - \boldsymbol{W} \boldsymbol{G}) \\
&= \underbrace{\mathbb{E}\, \boldsymbol{x}_q \boldsymbol{x}_q^\top}_{:=M_1} - \underbrace{\mathbb{E}\, \boldsymbol{x}_q \boldsymbol{x}_q^\top \boldsymbol{W} \boldsymbol{G}}_{:=M_2} - \underbrace{\mathbb{E}\, \boldsymbol{G} \boldsymbol{W}^\top \boldsymbol{x}_q \boldsymbol{x}_q^\top}_{:=M_3} + \underbrace{\mathbb{E}\, \boldsymbol{G} \boldsymbol{W}^\top \boldsymbol{x}_q \boldsymbol{x}_q^\top \boldsymbol{W} \boldsymbol{G}}_{:=M_4}
\end{aligned}
\tag{48}
$$

We denote the four pieces $M_1, M_2, M_3, M_4$ in order. First, we note that:

$$M_1 = \mathbb{E}\left[\boldsymbol{x}_q \boldsymbol{x}_q^\top\right] = \Sigma$$

Then, we expand out the terms in $M_2$:

$$
\begin{aligned}
\mathop{\mathbb{E}}_{\boldsymbol{x}_q, \boldsymbol{r}} \boldsymbol{x}_q \boldsymbol{x}_q^\top \boldsymbol{W} \boldsymbol{G} &= \left(\mathop{\mathbb{E}}_{\boldsymbol{x}_q, \boldsymbol{r}} \boldsymbol{x}_q \boldsymbol{x}_q^\top\right) \boldsymbol{W} \boldsymbol{G}_0 + \mathop{\mathbb{E}}_{\boldsymbol{x}_q, \boldsymbol{r}} \boldsymbol{x}_q \boldsymbol{x}_q^\top \boldsymbol{W} \sum_{i=1}^n (\boldsymbol{x}_q + \boldsymbol{r}_i)(\boldsymbol{x}_q + \boldsymbol{r}_i)^\top \\
&= \Sigma \boldsymbol{W} \boldsymbol{G}_0 + \mathop{\mathbb{E}}_{\boldsymbol{x}_q, \boldsymbol{r}} \boldsymbol{x}_q \boldsymbol{x}_q^\top \boldsymbol{W} \sum_{i=1}^n (\boldsymbol{x}_q + \boldsymbol{r}_i)(\boldsymbol{x}_q + \boldsymbol{r}_i)^\top \\
&= \Sigma \boldsymbol{W} \boldsymbol{G}_0 + \mathop{\mathbb{E}}_{\boldsymbol{x}_q, \boldsymbol{r}} \boldsymbol{x}_q \boldsymbol{x}_q^\top \boldsymbol{W} \sum_{i=1}^n (\boldsymbol{x}_q \boldsymbol{x}_q^\top + \boldsymbol{r}_i \boldsymbol{r}_i^\top) \\
&= \Sigma \boldsymbol{W} \boldsymbol{G}_0 + \mathop{\mathbb{E}}_{\boldsymbol{x}_q, \boldsymbol{r}} \boldsymbol{x}_q \boldsymbol{x}_q^\top \boldsymbol{W} \sum_{i=1}^n (\boldsymbol{x}_q \boldsymbol{x}_q^\top + \delta^2 I) \\
&= \Sigma \boldsymbol{W} \boldsymbol{G}_0 + n\delta^2 \Sigma \boldsymbol{W} + n\left(\Sigma\left(\boldsymbol{W} + \boldsymbol{W}^T\right)\Sigma + \mathrm{tr}(\boldsymbol{W}\Sigma)\Sigma\right) \\
&= \Sigma \boldsymbol{W} \boldsymbol{G}_0 + n(1 + \delta^2)\boldsymbol{W} + n\boldsymbol{W}^\top + n\,\mathrm{tr}(\boldsymbol{W})I
\end{aligned}
\tag{49}
$$

where the first step follows from the independence between $\boldsymbol{X}$ and $\boldsymbol{x}_q$, the second step follows from $\mathbb{E}\,\boldsymbol{r}_i = 0, \; \forall i \in [n]$, the third step follows from the expectation of $\boldsymbol{r}_i \boldsymbol{r}_i^\top = \delta^2 I$, and the last step follows from Equation (32). Then,

$$
\begin{aligned}
M_2 &= m\Sigma \boldsymbol{W}\Sigma + n\delta^2 \Sigma \boldsymbol{W} + n\left(\Sigma\left(\boldsymbol{W} + \boldsymbol{W}^T\right)\Sigma + \mathrm{tr}(\boldsymbol{W}\Sigma)\Sigma\right) \\
&= \Sigma\left(\left((m+n)W + nW^\top\right)\Sigma + n\delta^2 W + n\,\mathrm{tr}(W\Sigma)I\right) \\
&= (n\delta^2 + n + m)\boldsymbol{W} + n\boldsymbol{W}^\top + n\,\mathrm{tr}(\boldsymbol{W})I \quad \text{if } \Sigma = I
\end{aligned}
\tag{50}
$$

Similarly, $M_3 = M_2^\top$, and we have

$$
\begin{aligned}
M_1 - M_2 - M_3 &= \Sigma - (m+2n)\Sigma\left(\boldsymbol{W} + \boldsymbol{W}^\top\right)\Sigma - n\delta^2\left(\Sigma \boldsymbol{W} + \boldsymbol{W}^\top \Sigma\right) - 2n\,\mathrm{tr}(\boldsymbol{W}\Sigma)\Sigma \\
&= I - \left(m + 2n + n\delta^2\right)\left(\boldsymbol{W} + \boldsymbol{W}^\top\right) - 2n\,\mathrm{tr}(\boldsymbol{W})I \quad \text{if } \Sigma = I
\end{aligned}
\tag{51}
$$

Now, we perform similar expansion for $M_4$:

$$
\begin{aligned}
M_4 &= \mathop{\mathbb{E}}_{\boldsymbol{x}_q, \boldsymbol{X}}[\boldsymbol{G} \boldsymbol{W}^\top \boldsymbol{x}_q \boldsymbol{x}_q^\top \boldsymbol{W} \boldsymbol{G}] \\
&= \mathop{\mathbb{E}}_{\boldsymbol{x}_q, \boldsymbol{X}}\left[\left(\boldsymbol{G}_0 + \sum_{i\in[n]} \boldsymbol{G}_i\right) \boldsymbol{W}^\top \boldsymbol{x}_q \boldsymbol{x}_q^\top \boldsymbol{W} \left(\boldsymbol{G}_0 + \sum_{i\in[n]} \boldsymbol{G}_i\right)\right] \\
&= \mathop{\mathbb{E}}_{\boldsymbol{x}_q, \boldsymbol{X}}\left[\boldsymbol{G}_0 \boldsymbol{W}^\top \boldsymbol{x}_q \boldsymbol{x}_q^\top \boldsymbol{W} \boldsymbol{G}_0 + \boldsymbol{G}_0 \boldsymbol{W}^\top \boldsymbol{x}_q \boldsymbol{x}_q^\top \boldsymbol{W} \sum_{i\in[n]} \boldsymbol{G}_i + \sum_{i\in[n]} \boldsymbol{G}_i \boldsymbol{W}^\top \boldsymbol{x}_q \boldsymbol{x}_q^\top \boldsymbol{W} \boldsymbol{G}_0 \right. \\
&\qquad \left. + \sum_{i\in n} \boldsymbol{G}_i \boldsymbol{W}^\top \boldsymbol{x}_q \boldsymbol{x}_q^\top \boldsymbol{W} \boldsymbol{G}_i + \sum_{i,j\in n, i\neq j} \boldsymbol{G}_i \boldsymbol{W}^\top \boldsymbol{x}_q \boldsymbol{x}_q^\top \boldsymbol{W} \boldsymbol{G}_j\right] \\
&= \mathop{\mathbb{E}}_{\boldsymbol{x}_q, \boldsymbol{X}}\left[\boldsymbol{G}_0 \boldsymbol{W}^\top \boldsymbol{x}_q \boldsymbol{x}_q^\top \boldsymbol{W} \boldsymbol{G}_0 + n\underbrace{\boldsymbol{G}_0 \boldsymbol{W}^\top \boldsymbol{x}_q \boldsymbol{x}_q^\top \boldsymbol{W} \boldsymbol{G}_i}_{i\in[n]} + n\underbrace{\boldsymbol{G}_i \boldsymbol{W}^\top \boldsymbol{x}_q \boldsymbol{x}_q^\top \boldsymbol{W} \boldsymbol{G}_0}_{i\in[n]}\right. \\
&\qquad \left. + n\underbrace{\boldsymbol{G}_i \boldsymbol{W}^\top \boldsymbol{x}_q \boldsymbol{x}_q^\top \boldsymbol{W} \boldsymbol{G}_i}_{i\in[n]} + n(n-1)\underbrace{\boldsymbol{G}_i \boldsymbol{W}^\top \boldsymbol{x}_q \boldsymbol{x}_q^\top \boldsymbol{W} \boldsymbol{G}_j}_{i,j\in[n], i\neq j}\right]
\end{aligned}
\tag{52}
$$

First, we can compute that:

$$M_{41} := \mathop{\mathbb{E}}_{\boldsymbol{x}_q, \boldsymbol{X}}[\boldsymbol{G}_0 \boldsymbol{W}^\top \boldsymbol{x}_q \boldsymbol{x}_q^\top \boldsymbol{W} \boldsymbol{G}_0] = \mathop{\mathbb{E}}_{\boldsymbol{X}}[\boldsymbol{G}_0 \boldsymbol{W}^\top \boldsymbol{W} \boldsymbol{G}_0]$$

$$= m(m+1)\Sigma \boldsymbol{W}^\top \boldsymbol{W} \Sigma + m \operatorname{tr}(\boldsymbol{W}^\top \boldsymbol{W} \Sigma)\Sigma \tag{53}$$

$$= m(m+1)\boldsymbol{W}^\top \boldsymbol{W} + m \operatorname{tr}(\boldsymbol{W}^\top \boldsymbol{W})I \quad \text{if } \Sigma = I$$

where the last line follows from Equation (33) and symmetry of $\boldsymbol{W}^\top \boldsymbol{W}$. Then, $\forall i \in [n]$ and $\Sigma$, we have:

$$M_{42} := \mathop{\mathbb{E}}_{\boldsymbol{x}_q, \boldsymbol{X}} \boldsymbol{G}_0 \boldsymbol{W}^\top \boldsymbol{x}_q \boldsymbol{x}_q^\top \boldsymbol{W} \boldsymbol{G}_i = \mathop{\mathbb{E}}_{\boldsymbol{x}_q, \boldsymbol{X}} \boldsymbol{G}_0 \boldsymbol{W}^\top \boldsymbol{x}_q \boldsymbol{x}_q^\top \boldsymbol{W} (\boldsymbol{x}_q + \boldsymbol{r}_i)(\boldsymbol{x}_q + \boldsymbol{r}_i)^\top$$

$$= \mathop{\mathbb{E}}_{\boldsymbol{x}_q, \boldsymbol{X}} \boldsymbol{G}_0 \boldsymbol{W}^\top \boldsymbol{x}_q \boldsymbol{x}_q^\top \boldsymbol{W} \left(\boldsymbol{x}_q \boldsymbol{x}_q^\top + \boldsymbol{r}_i \boldsymbol{r}_i^\top\right)$$

$$= \mathop{\mathbb{E}}_{\boldsymbol{x}_q, \boldsymbol{X}} \boldsymbol{G}_0 \boldsymbol{W}^\top \left(\boldsymbol{W} + \boldsymbol{W}^\top + \operatorname{tr}(\boldsymbol{W}) + \boldsymbol{W}\delta^2\right) \tag{54}$$

$$= m \left(\boldsymbol{W}^\top \boldsymbol{W} + \boldsymbol{W}^\top \boldsymbol{W}^\top + \operatorname{tr}(\boldsymbol{W})\boldsymbol{W}^\top + \delta^2 \boldsymbol{W}^\top \boldsymbol{W}\right)$$

$$= m \left((1+\delta^2)\boldsymbol{W}^\top \boldsymbol{W} + \boldsymbol{W}^\top \boldsymbol{W}^\top + \operatorname{tr}(\boldsymbol{W})\boldsymbol{W}^\top\right)$$

where the first steps follows from $\mathbb{E}[\boldsymbol{r}_i] = 0$, the second step follows from Equation (32).

For general $\Sigma$, we can use a similar derivation and obtain:

$$M_{42} = \mathop{\mathbb{E}}_{\boldsymbol{x}_q, \boldsymbol{X}} \boldsymbol{G}_0 \boldsymbol{W}^\top \boldsymbol{x}_q \boldsymbol{x}_q^\top \boldsymbol{W} \left(\boldsymbol{x}_q \boldsymbol{x}_q^\top + \boldsymbol{r}_i \boldsymbol{r}_i^\top\right)$$

$$= m\Sigma \boldsymbol{W}^\top \left(\Sigma \left(\boldsymbol{W} + \boldsymbol{W}^T\right)\Sigma + \operatorname{tr}(\boldsymbol{W}\Sigma)\Sigma\right) + m\Sigma \boldsymbol{W}^\top \Sigma \boldsymbol{W} \delta^2 I \tag{55}$$

Moreover, we note that $\forall i \in [n]$:

$$M_{43} := \mathop{\mathbb{E}}_{\boldsymbol{x}_q, \boldsymbol{X}, \boldsymbol{r}_i} \boldsymbol{G}_i \boldsymbol{W}^\top \boldsymbol{x}_q \boldsymbol{x}_q^\top \boldsymbol{W} \boldsymbol{G}_i = (\boldsymbol{x}_q + \boldsymbol{r}_i)(\boldsymbol{x}_q + \boldsymbol{r}_i)^\top \boldsymbol{W}^\top \boldsymbol{x}_q \boldsymbol{x}_q^\top \boldsymbol{W} (\boldsymbol{x}_q + \boldsymbol{r}_i)(\boldsymbol{x}_q + \boldsymbol{r}_i)^\top$$

$$= (\boldsymbol{x}_q \boldsymbol{x}_q^\top + \boldsymbol{r}_i \boldsymbol{x}_q^\top + \boldsymbol{x}_q \boldsymbol{r}_i^\top + \boldsymbol{r}_i \boldsymbol{r}_i^\top)\boldsymbol{W}^\top \boldsymbol{x}_q \boldsymbol{x}_q^\top \boldsymbol{W} (\boldsymbol{x}_q \boldsymbol{x}_q^\top + \boldsymbol{r}_i \boldsymbol{x}_q^\top + \boldsymbol{x}_q \boldsymbol{r}_i^\top + \boldsymbol{r}_i \boldsymbol{r}_i^\top)$$

$$= \underbrace{\boldsymbol{x}_q \boldsymbol{x}_q^\top \boldsymbol{W}^\top \boldsymbol{x}_q \boldsymbol{x}_q^\top \boldsymbol{W} \boldsymbol{x}_q \boldsymbol{x}_q^\top}_{\text{0 order in } \boldsymbol{r}_i} + \underbrace{\boldsymbol{r}_i \boldsymbol{r}_i^\top \boldsymbol{W}^\top \boldsymbol{x}_q \boldsymbol{x}_q^\top \boldsymbol{W} \boldsymbol{r}_i \boldsymbol{r}_i^\top}_{\text{4th-order in } \boldsymbol{r}_i}$$

$$+ \underbrace{(\boldsymbol{r}_i \boldsymbol{x}_q^\top + \boldsymbol{x}_q \boldsymbol{r}_i^\top)\boldsymbol{W}^\top \boldsymbol{x}_q \boldsymbol{x}_q^\top \boldsymbol{W} (\boldsymbol{r}_i \boldsymbol{x}_q^\top + \boldsymbol{x}_q \boldsymbol{r}_i^\top)}_{\text{2nd-order in } \boldsymbol{r}_i}$$

$$+ \underbrace{\boldsymbol{r}_i \boldsymbol{r}_i^\top \boldsymbol{W}^\top \boldsymbol{x}_q \boldsymbol{x}_q^\top \boldsymbol{W} \boldsymbol{x}_q \boldsymbol{x}_q^\top + \boldsymbol{x}_q \boldsymbol{x}_q^\top \boldsymbol{W}^\top \boldsymbol{x}_q \boldsymbol{x}_q^\top \boldsymbol{W} \boldsymbol{r}_i \boldsymbol{r}_i^\top}_{\text{2nd-order in } \boldsymbol{r}_i}$$

$$\tag{56}$$

It worth noting that given Gaussian vector $\boldsymbol{r}_i$, then its monomial of odd order has 0 expectation according to Isserlis' Theorem (Isserlis, 1918). And we can thus obtain the third line by keeping only the even order monomials of $\boldsymbol{r}_i$.

By adding up Lemma 3 and all the terms above, we obtain that for $\Sigma = I$:

$$
\begin{aligned}
\mathop{\mathbb{E}}_{\boldsymbol{x}_q, \boldsymbol{X}, \boldsymbol{r}_i} & \boldsymbol{G}_i \boldsymbol{W}^\top \boldsymbol{x}_q \boldsymbol{x}_q^\top \boldsymbol{W} \boldsymbol{G}_i \\
&= \left. \begin{array}{l} 2\left(\boldsymbol{W}^2 + (\boldsymbol{W}^2)^\top + \boldsymbol{W}^\top \boldsymbol{W} + \boldsymbol{W} \boldsymbol{W}^\top + \operatorname{tr}(\boldsymbol{W})(\boldsymbol{W} + \boldsymbol{W}^\top)\right) \\ + \operatorname{tr}(\boldsymbol{W})^2 I + \operatorname{tr}(\boldsymbol{W}^2) I + \operatorname{tr}(\boldsymbol{W}^\top \boldsymbol{W}) I \end{array} \right\} \text{0th-order in } \boldsymbol{r}_i, \text{ Lemma 3} \\
&\quad + \underbrace{2\delta^4 \boldsymbol{W}^\top \boldsymbol{W} + \delta^4 \operatorname{tr}(\boldsymbol{W}^\top \boldsymbol{W}) I}_{\text{4th-order in } \boldsymbol{r}_i, \text{ Equation (19)}} \\
&\quad + \delta^2 \underbrace{\left[\operatorname{tr}(\boldsymbol{W})\left(\boldsymbol{W}^\top + \boldsymbol{W}\right) + \boldsymbol{W}^2 + (\boldsymbol{W}^2)^\top + 2\boldsymbol{W}^\top \boldsymbol{W}\right]}_{\text{Equation (23) and its transpose}} \\
&\quad + \underbrace{\left(\operatorname{tr}(\boldsymbol{W}^2) + \operatorname{tr}(\boldsymbol{W}^\top \boldsymbol{W}) + \operatorname{tr}(\boldsymbol{W})^2\right) \delta^2 I}_{\text{Equation (20)}} \\
&\quad + \underbrace{2\delta^2 \boldsymbol{W} \boldsymbol{W}^\top + \delta^2 \operatorname{tr}(\boldsymbol{W}^\top \boldsymbol{W}) I}_{\text{Equation (21)}} \\
&\quad + \delta^2 \underbrace{\left[\operatorname{tr}(\boldsymbol{W})\left(\boldsymbol{W}^\top + \boldsymbol{W}\right) + \boldsymbol{W}^2 + (\boldsymbol{W}^2)^\top + 2\boldsymbol{W}^\top \boldsymbol{W}\right]}_{\text{Equation (22) and its transpose}} \\
&= (2 + 2\delta^2)\left[\operatorname{tr}(\boldsymbol{W})\left(\boldsymbol{W}^\top + \boldsymbol{W}\right) + \boldsymbol{W}^2 + (\boldsymbol{W}^2)^\top\right] \\
&\quad + (2 + 4\delta^2)\boldsymbol{W}^\top \boldsymbol{W} + 2\boldsymbol{W} \boldsymbol{W}^\top \\
&\quad + (1 + \delta^2)\left[\operatorname{tr}(\boldsymbol{W})^2 I + \operatorname{tr}(\boldsymbol{W}^2) I + \operatorname{tr}(\boldsymbol{W}^\top \boldsymbol{W}) I\right] \\
&\quad + 2\delta^4 \boldsymbol{W}^\top \boldsymbol{W} + \delta^4 \operatorname{tr}(\boldsymbol{W}^\top \boldsymbol{W}) I + 2\delta^2 \boldsymbol{W} \boldsymbol{W}^\top + \delta^2 \operatorname{tr}(\boldsymbol{W}^\top \boldsymbol{W}) I \\
&= (2 + 2\delta^2)\left[\operatorname{tr}(\boldsymbol{W})\left(\boldsymbol{W}^\top + \boldsymbol{W}\right) + \boldsymbol{W}^2 + (\boldsymbol{W}^2)^\top\right] \\
&\quad + (2 + 4\delta^2 + 2\delta^4)\boldsymbol{W}^\top \boldsymbol{W} + (2 + 2\delta^2)\boldsymbol{W} \boldsymbol{W}^\top \\
&\quad + (1 + \delta^2)\left(\operatorname{tr}(\boldsymbol{W})^2 + \operatorname{tr}(\boldsymbol{W}^2)\right) I + (1 + 2\delta^2 + \delta^4)\operatorname{tr}(\boldsymbol{W}^\top \boldsymbol{W}) I
\end{aligned}
\tag{57}
$$

For generic $\Sigma$, we can make a similar derivation:

$$\underset{\boldsymbol{x}_q,\boldsymbol{X},\boldsymbol{r}_i}{\mathbb{E}} \boldsymbol{G}_i \boldsymbol{W}^\top \boldsymbol{x}_q \boldsymbol{x}_q^\top \boldsymbol{W} \boldsymbol{G}_i$$

$$= 2\left(\Sigma \boldsymbol{W}^\top \Sigma \boldsymbol{W} \Sigma + \Sigma \boldsymbol{W}^\top \Sigma \boldsymbol{W}^\top \Sigma + \Sigma \boldsymbol{W} \Sigma \boldsymbol{W} \Sigma + \Sigma \boldsymbol{W} \Sigma \boldsymbol{W}^\top \Sigma\right)$$

$$+ 2\operatorname{tr}(\Sigma \boldsymbol{W})\left(\Sigma \boldsymbol{W} \Sigma + \Sigma \boldsymbol{W}^\top \Sigma\right)$$

$$+ \operatorname{tr}(\Sigma \boldsymbol{W})^2 \Sigma + \operatorname{tr}\left(\Sigma \boldsymbol{W}^\top \Sigma \boldsymbol{W}^\top\right)\Sigma + \operatorname{tr}\left(\Sigma \boldsymbol{W}^\top \Sigma \boldsymbol{W}\right)\Sigma$$

$$+ 2\delta^4 \boldsymbol{W}^\top \Sigma \boldsymbol{W} + \delta^4 \operatorname{tr}(\boldsymbol{W}^\top \Sigma \boldsymbol{W})I$$

$$+ \left(\operatorname{tr}(\boldsymbol{W} \Sigma \boldsymbol{W}^\top \Sigma) + \operatorname{tr}(\boldsymbol{W} \Sigma \boldsymbol{W} \Sigma) + \operatorname{tr}(\boldsymbol{W} \Sigma)\operatorname{tr}(\boldsymbol{W}^\top \Sigma)\right)\delta^2 I$$

$$+ 2\delta^2 \Sigma \boldsymbol{W} \boldsymbol{W}^\top \Sigma + \delta^2 \operatorname{tr}(\boldsymbol{W} \boldsymbol{W}^\top \Sigma)\Sigma$$

$$+ 2\delta^2 \left(\boldsymbol{W}^\top \Sigma \boldsymbol{W} \Sigma + \boldsymbol{W}^\top \Sigma \boldsymbol{W}^\top \Sigma + \boldsymbol{W}^\top \operatorname{tr}(\boldsymbol{W} \Sigma)\Sigma\right)$$

$$+ 2\delta^2 \left(\boldsymbol{W}^\top \Sigma \boldsymbol{W} \Sigma + \boldsymbol{W}^\top \Sigma \boldsymbol{W}^\top \Sigma + \boldsymbol{W}^\top \operatorname{tr}(\boldsymbol{W} \Sigma)\Sigma\right)^\top$$

$$= 2\left(\Sigma \boldsymbol{W}^\top \Sigma \boldsymbol{W} \Sigma + \Sigma \boldsymbol{W}^\top \Sigma \boldsymbol{W}^\top \Sigma + \Sigma \boldsymbol{W} \Sigma \boldsymbol{W} \Sigma + \Sigma \boldsymbol{W} \Sigma \boldsymbol{W}^\top \Sigma\right)$$

$$+ 2\operatorname{tr}(\Sigma \boldsymbol{W})\left(\Sigma \boldsymbol{W} \Sigma + \Sigma \boldsymbol{W}^\top \Sigma\right)$$

$$+ \left(\operatorname{tr}(\Sigma \boldsymbol{W})^2 + \operatorname{tr}\left(\Sigma \boldsymbol{W}^\top \Sigma \boldsymbol{W}\right) + \operatorname{tr}(\Sigma \boldsymbol{W} \Sigma \boldsymbol{W})\right)\Sigma$$

$$+ 2\delta^4 \boldsymbol{W}^\top \Sigma \boldsymbol{W} + \delta^4 \operatorname{tr}\left(\boldsymbol{W}^\top \Sigma \boldsymbol{W}\right)I$$

$$+ \delta^2 \left(\operatorname{tr}\left(\Sigma \boldsymbol{W}^\top \Sigma \boldsymbol{W}\right) + \operatorname{tr}(\Sigma \boldsymbol{W} \Sigma \boldsymbol{W}) + \operatorname{tr}(\Sigma \boldsymbol{W})^2\right)I$$

$$+ \delta^2 \left(2\Sigma \boldsymbol{W} \boldsymbol{W}^\top \Sigma + \operatorname{tr}\left(\boldsymbol{W}^\top \Sigma \boldsymbol{W}\right)\Sigma\right)$$

$$+ 2\delta^2 \left(\boldsymbol{W}^\top \Sigma \boldsymbol{W} \Sigma + \boldsymbol{W}^\top \Sigma \boldsymbol{W}^\top \Sigma + \operatorname{tr}(\Sigma \boldsymbol{W})\boldsymbol{W}^\top \Sigma\right)$$

$$+ 2\delta^2 \left(\Sigma \boldsymbol{W}^\top \Sigma \boldsymbol{W} + \Sigma \boldsymbol{W} \Sigma \boldsymbol{W} + \operatorname{tr}(\Sigma \boldsymbol{W})\Sigma \boldsymbol{W}\right)$$

$$\tag{58}$$

Also, we expand the cross-term out for $\forall i, j \in [n], i \neq j$ and $\Sigma = I$:

$$M_{44} := \mathbb{E}\, \boldsymbol{G}_i \boldsymbol{W}^\top \boldsymbol{x}_q \boldsymbol{x}_q^\top \boldsymbol{W} \boldsymbol{G}_j = \mathbb{E}(\boldsymbol{x}_q + \boldsymbol{r}_i)(\boldsymbol{x}_q + \boldsymbol{r}_i)^\top \boldsymbol{W}^\top \boldsymbol{x}_q \boldsymbol{x}_q^\top \boldsymbol{W}(\boldsymbol{x}_q + \boldsymbol{r}_j)(\boldsymbol{x}_q + \boldsymbol{r}_j)^\top$$

$$= \left(\boldsymbol{x}_q \boldsymbol{x}_q^\top + \boldsymbol{r}_i \boldsymbol{r}_i^\top\right) \boldsymbol{W}^\top \boldsymbol{x}_q \boldsymbol{x}_q^\top \boldsymbol{W}\left(\boldsymbol{x}_q \boldsymbol{x}_q^\top + \boldsymbol{r}_j \boldsymbol{r}_j^\top\right)$$

$$= \boldsymbol{x}_q \boldsymbol{x}_q^\top \boldsymbol{W}^\top \boldsymbol{x}_q \boldsymbol{x}_q^\top \boldsymbol{W} \boldsymbol{x}_q \boldsymbol{x}_q^\top + \boldsymbol{r}_i \boldsymbol{r}_i^\top \boldsymbol{W}^\top \boldsymbol{x}_q \boldsymbol{x}_q^\top \boldsymbol{W} \boldsymbol{x}_q \boldsymbol{x}_q^\top$$

$$+ \boldsymbol{x}_q \boldsymbol{x}_q^\top \boldsymbol{W}^\top \boldsymbol{x}_q \boldsymbol{x}_q^\top \boldsymbol{W} \boldsymbol{r}_j \boldsymbol{r}_j^\top + \boldsymbol{r}_i \boldsymbol{r}_i^\top \boldsymbol{W}^\top \boldsymbol{x}_q \boldsymbol{x}_q^\top \boldsymbol{W} \boldsymbol{r}_j \boldsymbol{r}_j^\top$$

$$= 2\left(\boldsymbol{W}^2 + \left(\boldsymbol{W}^2\right)^\top + \boldsymbol{W}^\top \boldsymbol{W} + \boldsymbol{W} \boldsymbol{W}^\top + \operatorname{tr}(\boldsymbol{W})\boldsymbol{W} + \operatorname{tr}(\boldsymbol{W})\boldsymbol{W}^\top\right)$$

$$+ \operatorname{tr}(\boldsymbol{W})^2 I + \operatorname{tr}\left(\boldsymbol{W}^2\right)I + \operatorname{tr}\left(\boldsymbol{W}^\top \boldsymbol{W}\right)I$$

$$+ \delta^2 \left(\boldsymbol{W}^2 + \left(\boldsymbol{W}^2\right)^\top + 2\boldsymbol{W}^\top \boldsymbol{W}\right) + \operatorname{tr}(\boldsymbol{W})(\boldsymbol{W} + \boldsymbol{W}^\top) + \delta^4 \boldsymbol{W}^\top \boldsymbol{W}$$

$$= (2 + \delta^2)\left(\boldsymbol{W}^2 + \left(\boldsymbol{W}^2\right)^\top + \operatorname{tr}(\boldsymbol{W})\boldsymbol{W} + \operatorname{tr}(\boldsymbol{W})\boldsymbol{W}^\top\right)$$

$$+ (2 + 2\delta^2)\boldsymbol{W}^\top \boldsymbol{W} + 2\boldsymbol{W} \boldsymbol{W}^\top$$

$$+ \operatorname{tr}(\boldsymbol{W})^2 I + \operatorname{tr}\left(\boldsymbol{W}^2\right)I + \operatorname{tr}\left(\boldsymbol{W}^\top \boldsymbol{W}\right)I + \delta^4 \boldsymbol{W}^\top \boldsymbol{W}$$

$$= (2 + \delta^2)\left(\boldsymbol{W}^2 + \left(\boldsymbol{W}^2\right)^\top + \operatorname{tr}(\boldsymbol{W})\boldsymbol{W} + \operatorname{tr}(\boldsymbol{W})\boldsymbol{W}^\top\right)$$

$$+ (2 + 2\delta^2 + \delta^4)\boldsymbol{W}^\top \boldsymbol{W} + 2\boldsymbol{W} \boldsymbol{W}^\top$$

$$+ \operatorname{tr}(\boldsymbol{W})^2 I + \operatorname{tr}\left(\boldsymbol{W}^2\right)I + \operatorname{tr}\left(\boldsymbol{W}^\top \boldsymbol{W}\right)I$$

$$\tag{59}$$

where the first step follows from the independence of $\boldsymbol{x}_q, \boldsymbol{r}_i, \boldsymbol{r}_j$, and the second step follows from applying Lemma 3 and Equation (32).

Similarly, we can compute $M_{44}$ for generic $\Sigma$,

$$
\begin{aligned}
M_{44} &= \boldsymbol{x}_q \boldsymbol{x}_q^\top \boldsymbol{W}^\top \boldsymbol{x}_q \boldsymbol{x}_q^\top \boldsymbol{W} \boldsymbol{x}_q \boldsymbol{x}_q^\top + \boldsymbol{r}_i \boldsymbol{r}_i^\top \boldsymbol{W}^\top \boldsymbol{x}_q \boldsymbol{x}_q^\top \boldsymbol{W} \boldsymbol{x}_q \boldsymbol{x}_q^\top \\
&\quad + \boldsymbol{x}_q \boldsymbol{x}_q^\top \boldsymbol{W}^\top \boldsymbol{x}_q \boldsymbol{x}_q^\top \boldsymbol{W} \boldsymbol{r}_j \boldsymbol{r}_j^\top + \boldsymbol{r}_i \boldsymbol{r}_i^\top \boldsymbol{W}^\top \boldsymbol{x}_q \boldsymbol{x}_q^\top \boldsymbol{W} \boldsymbol{r}_j \boldsymbol{r}_j^\top \\
&= 2 \left( \Sigma \boldsymbol{W}^\top \Sigma \boldsymbol{W} \Sigma + \Sigma \boldsymbol{W}^\top \Sigma \boldsymbol{W}^\top \Sigma + \Sigma \boldsymbol{W} \Sigma \boldsymbol{W} \Sigma + \Sigma \boldsymbol{W} \Sigma \boldsymbol{W}^\top \Sigma \right) \\
&\quad + 2 \operatorname{tr}(\Sigma \boldsymbol{W}) \left( \Sigma \boldsymbol{W} \Sigma + \Sigma \boldsymbol{W}^\top \Sigma \right) \\
&\quad + \operatorname{tr}(\Sigma \boldsymbol{W})^2 \Sigma + \operatorname{tr}\left( \Sigma \boldsymbol{W}^\top \Sigma \boldsymbol{W}^\top \right) \Sigma + \operatorname{tr}\left( \Sigma \boldsymbol{W}^\top \Sigma \boldsymbol{W} \right) \Sigma \\
&\quad + \left( \delta^2 \boldsymbol{W}^\top \left( \Sigma \left( \boldsymbol{W} + \boldsymbol{W}^T \right) \Sigma + \operatorname{tr}\left( \boldsymbol{W} \Sigma \right) \Sigma \right) \right) \\
&\quad + \left( \delta^2 \boldsymbol{W}^\top \left( \Sigma \left( \boldsymbol{W} + \boldsymbol{W}^T \right) \Sigma + \operatorname{tr}\left( \boldsymbol{W} \Sigma \right) \Sigma \right) \right)^\top \\
&\quad + \delta^4 \boldsymbol{W}^\top \Sigma \boldsymbol{W} \\
&= 2 \left( \Sigma \boldsymbol{W}^\top \Sigma \boldsymbol{W} \Sigma + \Sigma \boldsymbol{W}^\top \Sigma \boldsymbol{W}^\top \Sigma + \Sigma \boldsymbol{W} \Sigma \boldsymbol{W} \Sigma + \Sigma \boldsymbol{W} \Sigma \boldsymbol{W}^\top \Sigma \right) \\
&\quad + 2 \operatorname{tr}(\Sigma \boldsymbol{W}) \left( \Sigma \boldsymbol{W} \Sigma + \Sigma \boldsymbol{W}^\top \Sigma \right) \\
&\quad + \left( \operatorname{tr}(\Sigma \boldsymbol{W})^2 + \operatorname{tr}(\Sigma \boldsymbol{W} \Sigma \boldsymbol{W}) + \operatorname{tr}(\Sigma \boldsymbol{W}^\top \Sigma \boldsymbol{W}) \right) \Sigma \\
&\quad + \delta^2 \left( \boldsymbol{W}^\top \Sigma \boldsymbol{W} \Sigma + \boldsymbol{W}^\top \Sigma \boldsymbol{W}^\top \Sigma + \Sigma \boldsymbol{W} \Sigma \boldsymbol{W} + \Sigma \boldsymbol{W}^\top \Sigma \boldsymbol{W} + \operatorname{tr}(\Sigma \boldsymbol{W}) \left( \boldsymbol{W}^\top \Sigma + \Sigma \boldsymbol{W} \right) \right) \\
&\quad + \delta^4 \, \boldsymbol{W}^\top \Sigma \boldsymbol{W}
\end{aligned}
\tag{60}
$$

Combining the above terms together, we have for $\Sigma = I$:

$$
\begin{aligned}
M_4 &= M_{41} + n(M_{42} + M_{42}^\top) + n M_{43} + n(n-1) M_{44} \\
&= m(m+1) \boldsymbol{W}^\top \boldsymbol{W} + m \operatorname{tr}(\boldsymbol{W}^\top \boldsymbol{W}) I + mn \left( \left( 2 + 2\delta^2 \right) \boldsymbol{W}^\top \boldsymbol{W} + \boldsymbol{W}^2 + (\boldsymbol{W}^2)^\top + \operatorname{tr}\left( \boldsymbol{W} \right) \left( \boldsymbol{W} + \boldsymbol{W}^\top \right) \right) \\
&\quad + n M_{43} + n(n-1) M_{44} \\
&= 2n \left( 2n + \delta^2 \right) \boldsymbol{W}^2 + 2n \left( n + \delta^2 \right) \boldsymbol{W} \boldsymbol{W}^\top \\
&\quad + \left[ m^2 + m + (4 + 2\delta^2) mn + n^2 \left( 2 + 4\delta^2 + \delta^4 \right) + n \left( 2\delta^2 + \delta^4 \right) \right] \boldsymbol{W}^\top \boldsymbol{W} \\
&\quad + \left[ n^2 \left( 2 + \delta^2 \right) + n \left( m + \delta^2 \right) \right] \operatorname{tr}(\boldsymbol{W}) \left( \boldsymbol{W} + \boldsymbol{W}^\top \right) \\
&\quad + \left( n^2 + n\delta^2 \right) \left( \operatorname{tr}(\boldsymbol{W})^2 + \operatorname{tr}(\boldsymbol{W}^2) \right) I + \left[ m + n^2 + n \left( 2\delta^2 + \delta^4 \right) \right] \operatorname{tr}\left( \boldsymbol{W}^\top \boldsymbol{W} \right) I \\
&= \left[ n^2 \left( 2 + \delta^2 \right) + n \left( m + \delta^2 \right) \right] \left( \boldsymbol{W}^2 + (\boldsymbol{W}^2)^\top + \operatorname{tr}(\boldsymbol{W}) \left( \boldsymbol{W} + \boldsymbol{W}^\top \right) \right) \\
&\quad + \left[ 2n^2 + 2n\delta^2 \right] \boldsymbol{W} \boldsymbol{W}^\top \\
&\quad + \left[ m^2 + m + mn \left( 2 + 2\delta^2 \right) + n \left( 2\delta^2 + \delta^4 \right) + n^2 \left( 2 + 2\delta^2 + \delta^4 \right) \right] \boldsymbol{W}^\top \boldsymbol{W} \\
&\quad + \left[ n^2 + n\delta^2 \right] \left( \operatorname{tr}(\boldsymbol{W})^2 + \operatorname{tr}\left( \boldsymbol{W}^2 \right) \right) I \\
&\quad + \left[ m + n^2 + n \left( 2\delta^2 + \delta^4 \right) \right] \operatorname{tr}\left( \boldsymbol{W}^\top \boldsymbol{W} \right) I.
\end{aligned}
\tag{61}
$$

Similarly, for generic $\Sigma$, we have:

$$
\begin{aligned}
M_4 &= M_{41} + n\left(M_{42} + M_{42}^\top\right) + nM_{43} + n(n-1)M_{44} \\
&= m(m+1)\,\Sigma\boldsymbol{W}^\top\boldsymbol{W}\Sigma + m\,\mathrm{tr}\left(\boldsymbol{W}^\top\boldsymbol{W}\Sigma\right)\Sigma + (2n^2+2nm)\,\Sigma\boldsymbol{W}^\top\Sigma\boldsymbol{W}\Sigma \\
&\quad + (2n^2+nm)\left(\Sigma\boldsymbol{W}^\top\Sigma\boldsymbol{W}^\top\Sigma + \Sigma\boldsymbol{W}\Sigma\boldsymbol{W}\Sigma\right) + 2n^2\,\Sigma\boldsymbol{W}\Sigma\boldsymbol{W}^\top\Sigma \\
&\quad + (2n^2+nm)\,\mathrm{tr}(\Sigma\boldsymbol{W})\left(\Sigma\boldsymbol{W}\Sigma + \Sigma\boldsymbol{W}^\top\Sigma\right) + n^2\left(\mathrm{tr}(\Sigma\boldsymbol{W})^2 + \mathrm{tr}(\Sigma\boldsymbol{W}\Sigma\boldsymbol{W}) + \mathrm{tr}(\Sigma\boldsymbol{W}^\top\Sigma\boldsymbol{W})\right)\Sigma \\
&\quad + \delta^2\left((n^2+n+nm)\left(\boldsymbol{W}^\top\Sigma\boldsymbol{W}\Sigma + \Sigma\boldsymbol{W}^\top\Sigma\boldsymbol{W}\right) + (n^2+n)\left(\boldsymbol{W}^\top\Sigma\boldsymbol{W}^\top\Sigma + \Sigma\boldsymbol{W}\Sigma\boldsymbol{W}\right)\right) \\
&\quad + \delta^2\left((n^2+n)\,\mathrm{tr}(\Sigma\boldsymbol{W})\left(\boldsymbol{W}^\top\Sigma + \Sigma\boldsymbol{W}\right) + 2n\,\Sigma\boldsymbol{W}\boldsymbol{W}^\top\Sigma + n\,\mathrm{tr}(\boldsymbol{W}^\top\Sigma\boldsymbol{W})\Sigma\right) \\
&\quad + \delta^2\left(n\left(\mathrm{tr}(\Sigma\boldsymbol{W}^\top\Sigma\boldsymbol{W}) + \mathrm{tr}(\Sigma\boldsymbol{W}\Sigma\boldsymbol{W}) + \mathrm{tr}(\Sigma\boldsymbol{W})^2\right)I\right) \\
&\quad + \delta^4\left((n^2+n)\,\boldsymbol{W}^\top\Sigma\boldsymbol{W} + n\,\mathrm{tr}(\boldsymbol{W}^\top\Sigma\boldsymbol{W})I\right).
\end{aligned}
$$

In summary, combining all terms together, we have:

$$
\mathcal{L}(\boldsymbol{W}) := \mathrm{err}_{\text{variance}} + \mathrm{err}_{\text{bias}} + \sigma^2
$$

where the *irreducible variance* is $\sigma^2$, and the *reducible variance* (variance of ICL + RAG) is

$$
\text{Variance of ICL} + \text{Variance of RAG} = \left[m\sigma^2 + \left(1+\delta^2\right)n\sigma_{\text{rag}}^2\right]\mathrm{tr}(\boldsymbol{W}^\top\boldsymbol{W}) + n\sigma_{\text{rag}}^2\,\mathrm{tr}(\boldsymbol{W}^2) + n\sigma_{\text{rag}}^2\,\mathrm{tr}(\boldsymbol{W})^2
$$

And the err from the bias for $\Sigma = I$ term is given as:

$$
\begin{aligned}
\mathrm{err}_{\text{bias}} &= \beta_{\text{tt}}^\top[M_1 - M_2 - M_3 + M_4]\beta_{\text{tt}} \\
&= \beta_{\text{tt}}^\top\left[I - (n\delta^2+2n+m)(\boldsymbol{W}+\boldsymbol{W}^\top) - 2n\,\mathrm{tr}(\boldsymbol{W})I + M_4\right]\beta_{\text{tt}} \\
&= \beta_{\text{tt}}^\top\left[I - (n\delta^2+2n+m)(\boldsymbol{W}+\boldsymbol{W}^\top) - 2n\,\mathrm{tr}(\boldsymbol{W})I \right. \\
&\quad + \left[n^2\left(2+\delta^2\right) + n\left(m+\delta^2\right)\right]\left(\boldsymbol{W}^2 + \left(\boldsymbol{W}^2\right)^\top\right) + 2n(n+\delta^2)\boldsymbol{W}\boldsymbol{W}^\top \\
&\quad + \left[m^2 + m + mn\left(2+2\delta^2\right) + n^2\left(2+2\delta^2+\delta^4\right) + n\left(2\delta^2+\delta^4\right)\right]\boldsymbol{W}^\top\boldsymbol{W} \\
&\quad + \left[n^2\left(2+\delta^2\right) + n\left(m+\delta^2\right)\right]\left(\mathrm{tr}(\boldsymbol{W})\left(\boldsymbol{W}+\boldsymbol{W}^\top\right)\right) \\
&\quad \left. + \left[n^2+n\delta^2\right]\left(\mathrm{tr}(\boldsymbol{W})^2 + \mathrm{tr}\left(\boldsymbol{W}^2\right)\right)I + \left[m+n^2+n\left(2\delta^2+\delta^4\right)\right]\mathrm{tr}\left(\boldsymbol{W}^\top\boldsymbol{W}\right)I\right]\beta_{\text{tt}}
\end{aligned}
$$

And we can also derive the exact form of bias induced error for generic $\Sigma$. $\qquad\square$

The previous theorem gives the exact form the RAG population with general $\boldsymbol{W}$. In the following proposition, we will compute the population under special $\boldsymbol{W}$ in order to obtain a more fine-grained complexity analysis.

**Proposition 3** (RAG Population loss under isotropic setting)**.** *Assuming* $\boldsymbol{W}^* = \frac{m}{(m+d+1)(m+n)}I$. *Then, the population loss are given as:*

$$
\mathcal{L}_{tt+rag}(\boldsymbol{W}^*) = \mathrm{err}_{variance}(\boldsymbol{W}^*) + \mathrm{err}_{bias}(\boldsymbol{W}^*) + \sigma^2
$$

$$
\mathrm{err}_{variance}(\boldsymbol{W}^*) = \frac{m^3 d}{[(m+d+1)(m+n)]^2}\sigma^2 + \frac{dm^2n(2+\delta^2+d)}{[(m+d+1)(m+n)]^2}\sigma_{\text{rag}}^2
$$

$$
\mathrm{err}_{bias}(\boldsymbol{W}^*) = \|\beta_{\text{tt}}\|_2^2\left[1 - \frac{2m}{(m+d+1)(m+n)}\left(n\delta^2+2n+m+nd\right) + \frac{P(m,n,d,\delta)m^2}{(m+d+1)^2(m+n)^2}\right]
$$

*where*

$$
\begin{aligned}
P(m,n,d,\delta) =\ &6n^2 + 4n\delta^2 + m^2 + m + \left(4+2\delta^2\right)mn \\
&+ n^2\left(2+4\delta^2+\delta^4\right) + n\left(2\delta^2+\delta^4\right) + 2dn^2\left(2+\delta^2\right) + 2dn\left(m+\delta^2\right) \\
&+ d(d+1)\left(n^2+n\delta^2\right) + dm + dn^2 + dn\left(2\delta^2+\delta^4\right)
\end{aligned}
$$

*Proof.* Plugging in the value of $\boldsymbol{W}^*$, we first compute the error from input variance.

$$\text{tr}((\boldsymbol{W}^*)^2) = \frac{dm^2}{(m+d+1)^2(m+n)^2}$$

$$\text{tr}(\boldsymbol{W}^*) = \frac{dm}{(m+d+1)(m+n)}$$

$$\text{err}_{\text{variance}}(\boldsymbol{W}^*) = \left[m\sigma^2 + \left(1+\delta^2\right)n\sigma_{rag}^2\right]\text{tr}\left(\boldsymbol{W}^\top\boldsymbol{W}\right) + n\sigma_{rag}^2\,\text{tr}\left(\boldsymbol{W}^2\right) + n\sigma_{rag}^2\,\text{tr}(\boldsymbol{W})^2$$

$$= \frac{dm^2[m\sigma^2 + (1+\delta^2)n\sigma_{\text{rag}}^2]}{(m+d+1)^2(m+n)^2} + n\sigma_{\text{rag}}^2\frac{dm^2}{(m+d+1)^2(m+n)^2} + n\sigma_{\text{rag}}^2\frac{d^2m^2}{(m+d+1)^2(m+n)^2}$$

$$= \frac{m^3d}{[(m+d+1)(m+n)]^2}\sigma^2 + \frac{dm^2n(2+\delta^2+d)}{[(m+d+1)(m+n)]^2}\sigma_{\text{rag}}^2$$

Then, we proceed to plug in the value and compute the error from the estimation bias.

$$\text{err}_{\text{bias}}(\boldsymbol{W}^*) = \|\beta_{\text{tt}}\|_2^2\left[1 - \frac{2m(n\delta^2+2n+m)}{(m+n)(m+d+1)} - \frac{2ndm}{(m+n)(m+d+1)} + \frac{m^2}{(m+d+1)^2(m+n)^2}\underbrace{(\ldots)}_{P(m,n,d,\delta)}\right]$$

$$= \|\beta_{\text{tt}}\|_2^2\left[1 - \frac{2m}{(m+d+1)(m+n)}\left(n\delta^2+2n+m+nd\right) + \frac{P(m,n,d,\delta)m^2}{(m+d+1)^2(m+n)^2}\right]$$

where

$$P(m,n,d,\delta) = (2c_1+c_2+c_3)+2dc_4+(d^2+d)c_5+dc_6$$
$$= 2\left(n^2\left(2+\delta^2\right)+n\left(m+\delta^2\right)\right)+2n(n+\delta^2)$$
$$\quad + \left[m^2+m+mn\left(2+2\delta^2\right)+n^2\left(2+2\delta^2+\delta^4\right)+n\left(2\delta^2+\delta^4\right)\right]$$
$$\quad + 2d[n^2(2+\delta^2)+n(m+\delta^2)]+(d^2+d)(n^2+n\delta^2)+d[m+n^2+n(2\delta^2+\delta^4)]$$
$$= 6n^2+4n\delta^2+m^2+m+\left(4+2\delta^2\right)mn$$
$$\quad + n^2\left(2+4\delta^2+\delta^4\right)+n\left(2\delta^2+\delta^4\right)+2dn^2\left(2+\delta^2\right)+2dn\left(m+\delta^2\right)$$
$$\quad + d(d+1)\left(n^2+n\delta^2\right)+dm+dn^2+dn\left(2\delta^2+\delta^4\right)$$

$\square$

### B.1.1 FINITE SAMPLE COMPLEXITY OF RAG

**Proposition** (Restatement of Proposition 1). *Under Assumption 1, 2, 3, if $\delta^2 \ll 1$,*

$$\mathcal{L}_{\text{tt+rag}}(\boldsymbol{W}^*) = \mathcal{O}_{m,n}\left(\sigma^2 + \underbrace{\frac{dm}{(m+n)^2}\sigma^2 + \frac{d^2n}{(m+n)^2}\sigma_{\text{rag}}^2}_{\text{err}_{\textit{variance}}(\boldsymbol{W}^*)} + \underbrace{\|\beta_{\text{tt}}\|_2^2\left[\frac{d}{m}+d^2\left(\frac{n}{m+n}\right)^2\right]}_{\text{err}_{\textit{bias}}(\boldsymbol{W}^*)}\right)$$

$$\text{err}_{\text{variance}}(\boldsymbol{W}^*) = \begin{cases} \mathcal{O}_m(\frac{d}{m}\sigma^2 + \frac{d^2}{m^2}\sigma_{\text{rag}}^2) = \mathcal{O}_m\left(\frac{1}{m}\right) & m\to\infty, n\,\textit{fixed.} \\ \mathcal{O}_n(\frac{d}{n^2}\sigma^2 + \frac{d^2}{n}\sigma_{\text{rag}}^2) = \mathcal{O}_n\left(\frac{1}{n}\right) & n\to\infty, m\,\textit{fixed} \\ \mathcal{O}_m(\frac{d}{m}\sigma^2 + \frac{d^2}{m}\sigma_{\text{rag}}^2) = \mathcal{O}_m\left(\frac{1}{m}\right) & m\to\infty, n=\Theta_m(m) \end{cases} \quad (62)$$

$$\text{err}_{\text{bias}}(\boldsymbol{W}^*) = \begin{cases} \mathcal{O}_m\left(\|\beta_{\text{tt}}\|_2^2\frac{d}{m}\right) & \textit{if } m\to\infty, n \textit{ is fixed} \\ \mathcal{O}_n\left(\|\beta_{\text{tt}}\|_2^2d^2\right) = C_1 & \textit{if } n\to\infty, m \textit{ is fixed} \\ \mathcal{O}_m\left(\|\beta_{\text{tt}}\|_2^2\left(\frac{d}{m}+d^2\right)\right) = C_2 + \mathcal{O}_m(\|\beta_{\text{tt}}\|_2^2\frac{d}{m}) & \textit{if } m\to\infty, n=\Theta_m(m) \end{cases} \quad (63)$$

*Proof.* We will bound the variance-induced error and the bias-induced error separately.

**Variance-Induced Error**  First, we try to bound $\text{err}_{\text{variance}}(\boldsymbol{W}^*)$:

$$
\begin{aligned}
\text{err}_{\text{variance}}(\boldsymbol{W}^*) &= \frac{dm^3}{(m+d+1)^2(m+n)^2}\sigma^2 + \frac{dm^2n(2+\delta^2+d)}{(m+d+1)^2(m+n)^2}\sigma_{\text{rag}}^2 \\
&\leq \frac{dm^3}{m^2(m+n)^2}\sigma^2 + \frac{dm^2n(d+\delta^2+2)}{m^2(m+n)^2}\sigma_{\text{rag}}^2 \\
&= \frac{dm}{(m+n)^2}\sigma^2 + \frac{d(2+\delta^2+d)n}{(m+n)^2}\sigma_{\text{rag}}^2 \\
&= \mathcal{O}_{m,n}\left(\frac{dm}{(m+n)^2}\sigma^2 + \frac{d^2n}{(m+n)^2}\sigma_{\text{rag}}^2\right) \\
&= \begin{cases}
\mathcal{O}_m(\frac{d}{m}\sigma^2 + \frac{d^2}{m^2}\sigma_{\text{rag}}^2) = \mathcal{O}_m\left(\frac{1}{m}\right) & m \to \infty, n \text{ fixed.} \\
\mathcal{O}_n(\frac{d}{n^2}\sigma^2 + \frac{d^2}{n}\sigma_{\text{rag}}^2) = \mathcal{O}_n\left(\frac{1}{n}\right) & n \to \infty, m \text{ fixed} \\
\mathcal{O}_m(\frac{d}{m}\sigma^2 + \frac{d^2}{m}\sigma_{\text{rag}}^2) = \mathcal{O}_m\left(\frac{1}{m}\right) & m, n \to \infty, n = \Theta(m)
\end{cases}
\end{aligned}
\tag{64}
$$

where the second line follows from $(m+d+1) \geq m$ and the fourth line follows from the fact that $\delta^2$ is small relative to $m, n, d$.

**Bias-Induced Error**  We will expand out the term

$$
\text{err}_{\text{bias}}(\boldsymbol{W}^*) = \|\beta_{\text{tt}}\|_2^2 \frac{Q(m,n;d,\delta^2)}{(m+d+1)^2(m+n)^2}
\tag{65}
$$

where

$$
\begin{aligned}
Q(m,n;d,\delta^2) &:= (m+n)^2(m+d+1)^2 - 2m(m+n)(m+d+1)(n\delta^2 + 2n + m + nd) + m^2P(m,n,d,\delta^2) \\
&= (d+1)m^3 + \underbrace{(d^2 + 2d\delta^2 + 4d + \delta^4 + 2\delta^2 + 5)}_{:=\kappa_{22}}m^2n^2 \\
&\quad + \underbrace{(d^2\delta^2 - 2d^2 + d\delta^4 + 3d\delta^2 - 4d + \delta^4 + 4\delta^2 - 2)}_{:=\kappa_{21}}m^2n \\
&\quad - \underbrace{(2d^2 + 2d\delta^2 + 4d + 2\delta^2 + 2)}_{:=\kappa_{12}}mn^2 + (d^2 + 2d + 1)(m+n)^2 \\
&= (d+1)m^3 + \kappa_{22}m^2n^2 + |\kappa_{21}|m^2n + \text{lower-order terms} \\
&\leq (d+1)m^3 + \kappa_{22}m^2n^2 + |\kappa_{21}|m^2n + (d+1)^2(m+n)^2
\end{aligned}
\tag{66}
$$

where the last line follows from $\kappa_{12} < 0$.

Note that we assume $\delta^2 \ll 1$. Now, we can bound each of the term in $Q$ divided individually:

- Cubic term:
$$
\frac{(d+1)m^3}{m^2(m+n)^2} = \frac{d+1}{m}\left(\frac{m}{m+n}\right)^2 \leq \frac{d+1}{m}
\tag{67}
$$

- Skew-cubic term:
$$
\frac{|\kappa_{21}|\, m^2n}{m^2(m+n)^2} = |\kappa_{21}|\frac{n}{(m+n)^2} \leq |\kappa_{21}|\frac{n}{(m+n)^2}
\tag{68}
$$

- Quartic term:
$$
\frac{\kappa_{22}m^2n^2}{m^2(m+n)^2} = \kappa_{22}\left(\frac{n}{m+n}\right)^2
\tag{69}
$$

- last term:
$$
(d+1)^2(m+n)^2\frac{1}{m^2(m+n)^2} = \frac{d^2}{m^2}
$$

Combining Equation (66), Equation (67), Equation (68), Equation (69), we can obtain that

$$\text{err}_{\text{bias}}(\boldsymbol{W}^*) = \mathcal{O}_{m,n}\left(\|\beta_{\text{tt}}\|_2^2 \left[\frac{dm}{(m+n)^2} + d^2\frac{n^2}{(m+n)^2} + \frac{d^2}{m^2}\right]\right)$$

$$= \begin{cases} \mathcal{O}_m\left(\|\beta_{\text{tt}}\|_2^2\frac{d}{m}\right) & \text{if } m \to \infty, n \text{ is fixed} \\ \mathcal{O}_n\left(\|\beta_{\text{tt}}\|_2^2 d^2\right) = C_1 & \text{if } n \to \infty, m \text{ is fixed} \\ \mathcal{O}_m\left(\|\beta_{\text{tt}}\|_2^2\left(\frac{d}{m} + d^2\right)\right) = C_2 + \mathcal{O}_m(\|\beta_{\text{tt}}\|_2^2\frac{d}{m}) & \text{if } m \to \infty, n = \Theta_m(m) \end{cases} \quad (70)$$

where the third step follows from plugging in the highest order monomial of $d$ from $\kappa_{21}, \kappa_{22}$.

$\square$

### B.1.2 Optimality of Number of RAG Examples

**Proposition** (Restatement of Proposition 2). *Under Under Assumption 1,2,3, $\delta^2 \ll 1$, and reasonable choice of $\sigma^2, \sigma_{rag}^2$ ($\sigma^2, \sigma_{rag}^2 \ll \|\beta_{\text{tt}}\|_2^2$), the optimal $n^*$ that minimizes the RAG loss follows:*

$$n^* = \mathcal{O}_m\left(\frac{m\left(d^2\|\beta_{\text{tt}}\|_2^2 + d\sigma^2 - d^2\sigma_{rag}^2\right)}{md^2\|\beta_{\text{tt}}\|_2^2 - d^2\sigma_{rag}^2}\right) = \mathcal{O}_m\left(\frac{d\|\beta_{\text{tt}}\|_2^2 + \sigma^2 - d\sigma_{rag}^2}{d\|\beta_{\text{tt}}\|_2^2}\right) \quad (71)$$

*and the improvement on loss from picking the optimal $n^*$ over $n = 0$ is given as:*

$$\mathcal{L}_{tt+rag}(\boldsymbol{W}^*)|_{n=0} - \mathcal{L}_{tt+rag}(\boldsymbol{W}^*)|_{n=n^*} = \mathcal{O}_m\left(\frac{1}{m^2}\right) \quad (72)$$

*Proof.* First, we define several constants that can lead to a cleaner calculation. Let $\omega_1 := d$, $\omega_2 := d^2$. Then,

$$\text{err}_{\text{variance}}(\boldsymbol{W}^*) = \frac{dm^3}{(m+d+1)^2(m+n)^2}\sigma^2 + \frac{dm^2n(2+\delta^2+d)}{(m+d+1)^2(m+n)^2}\sigma_{\text{rag}}^2$$

$$\approx \frac{m^3}{(m+d+1)^2(m+n)^2}\omega_1\sigma^2 + \frac{m^2n}{(m+d+1)^2(m+n)^2}\omega_2\sigma_{\text{rag}}^2$$

where the last line follows from $\delta^2 \ll 1$. Let $Q(m,n,d,\delta^2) := \frac{\text{err}_{\text{bias}}(\boldsymbol{W}^*)(m+d+1)^2(m+n)^2}{\|\beta_{\text{tt}}\|_2^2}$ as in Equation (66). Then,

$$Q(m,n;d,\delta^2) = (m+n)^2(m+d+1)^2 - 2m(m+n)(m+d+1)(n\delta^2 + 2n + m + nd) + m^2P(m,n,d,\delta^2)$$

$$= (d+1)m^3 + (d^2 + 2d\delta^2 + 4d + \delta^4 + 2\delta^2 + 5)m^2n^2$$

$$+ (d^2\delta^2 - 2d^2 + d\delta^4 + 3d\delta^2 - 4d + \delta^4 + 4\delta^2 - 2)m^2n$$

$$- \left(2d^2 + 2d\delta^2 + 4d + 2\delta^2 + 2\right)mn^2 + (d^2 + 2d + 1)(m^2 + n^2)$$

$$\approx \underbrace{d}_{:=\tau_{30}}m^3 + \underbrace{d^2}_{\tau_{22}}m^2n^2 \underbrace{-2d^2}_{:=\tau_{21}}m^2n \underbrace{-2d^2}_{:=\tau_{12}}mn^2 + \underbrace{d^2}_{:=\tau_2}(m^2 + n^2)$$

$$= \tau_{30}m^3 + \tau_{22}m^2n^2 + \tau_{21}m^2n + \tau_{12}mn^2 + \tau_2(m^2 + n^2) \quad (73)$$

Now, we want to find the optimal $n^*$ w.r.t. $\mathcal{L}_{tt+rag}$. That is, we want to minimize

$$\left[m^3\omega_1\sigma^2 + m^2n\omega_2\sigma_{\text{rag}}^2 + \|\beta_{\text{tt}}\|_2^2\left(\tau_{30}m^3 + \tau_{22}m^2n^2 + \tau_{21}m^2n + \tau_{12}mn^2 + \tau_2\left(m^2 + n^2\right)\right)\right]\frac{1}{(m+n)^2(m+d+1)^2} \quad (74)$$

where all $\tau, \omega$ are positive except that $\tau_{12}$ is negative. First, we take out the terms that does not depend on $n$, and we equivalently minimize

$$L(n) := \left[m^3\omega_1\sigma^2 + m^2n\omega_2\sigma_{\text{rag}}^2 + \|\beta_{\text{tt}}\|_2^2\left(\tau_{30}m^3 + \tau_{22}m^2n^2 + \tau_{21}m^2n + \tau_{12}mn^2 + \tau_2\left(m^2 + n^2\right)\right)\right]\frac{1}{(m+n)^2}$$

Let

$$A = m^3\omega_1\sigma^2 + \|\beta_{\text{tt}}\|^2\tau_{30}m^3 + \|\beta_{\text{tt}}\|^2\tau_2m^2,$$

$$B = m^2\left(\omega_2\sigma_{\text{rag}}^2 + \|\beta_{\text{tt}}\|^2\tau_{21}\right), \quad (75)$$

$$C = \|\beta_{\text{tt}}\|^2\left(\tau_{22}m^2 + \tau_{12}m + \tau_2\right).$$

Then,
$$L(n) = (A + Bn + Cn^2)/(m + n)^2$$

Then, by the rule for derivative of quotient,
$$\frac{\partial (\mathcal{L}_{\text{tt+rag}}(\boldsymbol{W}^*))}{\partial n} = \frac{(B + 2Cn)(m + n)^2 - 2(m + n)\left(A + Bn + Cn^2\right)}{(m + n)^4}$$
$$= \frac{(B + 2Cn)(m + n) - 2\left(A + Bn + Cn^2\right)}{(m + n)^3}$$
$$= \frac{Bm + Bn + 2Cmn + 2Cn^2 - 2A - 2Bn - 2Cn^2}{(m + n)^3}$$
$$= \frac{Bm - Bn + 2Cmn - 2A}{(m + n)^3}$$

Set the derivative to be zero, we have

$$Bm - Bn + 2Cmn - 2A = 0$$

and
$$n^\star = \frac{Bm - 2A}{B - 2Cm}$$
$$= \frac{m(m^2\left(\omega_2\sigma_{\text{rag}}^2 + \|\beta_{\text{tt}}\|^2\tau_{21}\right)) - 2(m^3\omega_1\sigma^2 + \|\beta_{\text{tt}}\|^2\tau_{30}m^3 + \|\beta_{\text{tt}}\|^2\tau_2 m^2)}{(m^2\left(\omega_2\sigma_{\text{rag}}^2 + \|\beta_{\text{tt}}\|^2\tau_{21}\right)) - 2m(\|\beta_{\text{tt}}\|^2\left(\tau_{22}m^2 + \tau_{12}m + \tau_2\right))}$$
$$= \frac{m\left(2\|\beta_{\text{tt}}\|_2^2 dm + 2\|\beta_{\text{tt}}\|_2^2 d + 2\|\beta_{\text{tt}}\|_2^2 m - dm\sigma_{\text{rag}}^2 + 2m\sigma^2\right)}{d\left(2\|\beta_{\text{tt}}\|_2^2 m^2 - 2\|\beta_{\text{tt}}\|_2^2 m + 2\|\beta_{\text{tt}}\|_2^2 - m\sigma_{\text{rag}}^2\right)}$$
$$\leq \frac{md\left(2\|\beta_{\text{tt}}\|_2^2 dm - dm\sigma_{\text{rag}}^2 + 2m\sigma^2\right)}{d^2\left(2\|\beta_{\text{tt}}\|_2^2 m^2 - 2\|\beta_{\text{tt}}\|_2^2 m + 2\|\beta_{\text{tt}}\|_2^2 - m\sigma_{\text{rag}}^2\right)}$$
$$= \mathcal{O}_m\left(\frac{md\left(2\|\beta_{\text{tt}}\|_2^2 dm - dm\sigma_{\text{rag}}^2 + 2m\sigma^2\right)}{d^2\left(2\|\beta_{\text{tt}}\|_2^2 m^2 - m\sigma_{\text{rag}}^2\right)}\right)$$
$$= \mathcal{O}_m\left(\frac{m\left(d^2\|\beta_{\text{tt}}\|_2^2 + d\sigma^2 - d^2\sigma_{\text{rag}}^2\right)}{md^2\|\beta_{\text{tt}}\|_2^2 - d^2\sigma_{\text{rag}}^2}\right)$$
$$= \mathcal{O}_m\left(\frac{d\|\beta_{\text{tt}}\|_2^2 + \sigma^2 - d\sigma_{\text{rag}}^2}{d\|\beta_{\text{tt}}\|_2^2}\right)$$

where the third step follows from upper bounding the numerator, and the fourth step follows from lower bounding the denominator.

$n^*$ **as Global Minimizer**  Now, we will show that the stationary point is the global minimizer. The second order derivative is give as:
$$\frac{\partial (\mathcal{L}_{\text{tt+rag}}(\boldsymbol{W}^*))}{\partial n} = \frac{2\left(Cm^2 - 2Cmn - 2Bm + Bn + 3A\right)}{(m + n)^4} \tag{76}$$

Plug in $Bm - Bn^* + 2Cmn^* - 2A = 0$, we have
$$\frac{\partial (\mathcal{L}_{\text{tt+rag}}(\boldsymbol{W}^*))}{\partial n}\Big|_{n=n^*} = \frac{2\left(Cm^2 - A\right)}{(m - n)(m + n)^3} \geq 0 \tag{77}$$

Since $n^* = \mathcal{O}(1)$, we have $m > n^*$ for large $m$. Also, we have $Cm^2 > A$ for large $m$, thus we have $\frac{\partial(\mathcal{L}_{\text{tt+rag}}(\boldsymbol{W}^*))}{\partial n}|_{n=n^*} \geq 0$, and $n^*$ is the local minimum. Now, we check the first order derivative of $n \geq n^*$,
$$Bm - Bn + 2Cmn - 2A = Bm - Bn + 2Cmn - 2A - (Bm - Bn^* + 2Cmn^* - 2A)$$
$$= -B(n - n^*) + 2Cm(n - n^*) \geq 0$$

where it follows from $B \leq 0, C \geq 0$. Similarly, we can show that $Bm - Bn + 2Cmn - 2A \leq 0, \quad \forall n \leq n^*$. Thus, we have $n^*$ to be the global minimum of the loss.

**Improvement from** $n^*$    Here, we plug in $n = n^*$ and $n = 0$ into Equation (74). We have

$$\mathcal{L}_{\text{tt+rag}}|_{n=n^*}(\boldsymbol{W}^*) = \frac{A + Bn^* + C(n^*)^2}{(m + n^*)^2(m + d + 1)^2}$$

$$\mathcal{L}_{\text{tt+rag}}|_{n=0}(\boldsymbol{W}^*) = \frac{A}{m^2(m + d + 1)^2}$$

$$(78)$$

Then, the improvement is give as

$$\begin{aligned}
\mathcal{L}_{\text{tt+rag}}|_{n=0}(\boldsymbol{W}^*) - \mathcal{L}_{\text{tt+rag}}|_{n=n^*}(\boldsymbol{W}^*) &= \frac{A(m + n^*)^2 - m^2(A + Bn^* + C(n^*)^2)}{m^2(m + n^*)^2(m + d + 1)^2} \\
&= \frac{(n^*)^2(2Cm - B)}{2m^2\,(m + n^*)\,(m + d + 1)^2} \\
&= \mathcal{O}_m\left(\frac{Cm}{m^5}\right) \\
&= \mathcal{O}_m\left(\frac{m^2 d^2 \|\beta_{\text{tt}}\|_2^2 m}{m^5}\right) \\
&= \mathcal{O}_m\left(\frac{1}{m^2}\right)
\end{aligned}$$

$$(79)$$

where the second step step follows from $Bm - Bn^* + 2Cmn^* - 2A = 0$ and the third step follows from $n^* = \mathcal{O}(1)$, and the four step follows from $B \leq 0$ and $|B| = \mathcal{O}(C)$. It finishes the proof. $\qquad\square$

### B.2    NON-UNIFORM RETRIEVAL NOISE

Now, we proceed to the proof for non-uniform retrieval noise.

#### B.2.1    DISTANCE-PROPORTIONAL NOISE

**Theorem** (Restatement of Theorem 2). *Under Assumption 1, 2, 4, the population loss is given as:*

$$\hat{\text{err}}_{variance}(\boldsymbol{W}) = m\sigma^2 \operatorname{tr}(\boldsymbol{W}^\top \boldsymbol{W}) + \sum_{i=1}^{n} \gamma_1 \delta_i^2[(1 + \delta_i^2) \operatorname{tr}(\boldsymbol{W}^\top \boldsymbol{W}) + \operatorname{tr}(\boldsymbol{W}^2) + \operatorname{tr}(\boldsymbol{W})^2]$$

*If the variance of the retrieval distance follows power law, i.e. $\exists \gamma_2 > 0, q \geq 0$ s.t. $\delta_i^2 = \gamma_2 i^q$, then*

$$\hat{\text{err}}_{bias}(\boldsymbol{W}^*) = \mathcal{O}_{m,n}\left(\text{err}_{bias}(\boldsymbol{W}^*) + \|\beta_{\text{tt}}\|_2^2\left[\frac{dn^{2q+1} + n^{2q+2}}{(m + n)^2}\right]\right) \tag{80}$$

*and*

$$\hat{\text{err}}_{variance}(\boldsymbol{W}^*) = \mathcal{O}_n\left(\frac{dm\sigma^2 + d(n^{2q+1})\sigma^2}{(m + n)^2}\right) = \begin{cases} \mathcal{O}_n\left(dn^{2q-1}\sigma^2\right) & \text{if } n \to \infty, q \leq 1/2 \\ \text{diverges} & \text{if } n \to \infty, q > 1/2 \end{cases} \tag{81}$$

*Proof.* We first write down the error explicitly similar to Equation (40).

$$y_q - \boldsymbol{x}_q^\top \boldsymbol{W} \boldsymbol{X}^\top \boldsymbol{y} = \boldsymbol{x}_q^\top (I - \boldsymbol{W} \boldsymbol{G}) \beta_{\text{tt}} - \boldsymbol{x}_q^\top \boldsymbol{W} \boldsymbol{X}^\top \boldsymbol{\epsilon} + \epsilon_q$$

And we can break down the population loss as

$$\hat{\mathcal{L}}_{\text{tt+rag}}(\boldsymbol{W}) = \underset{(\boldsymbol{x}_q, y_q),(\boldsymbol{X}, \boldsymbol{y}), \boldsymbol{\epsilon}, \boldsymbol{r}}{\mathbb{E}} \left(\boldsymbol{x}_q^\top (I - \boldsymbol{W} \boldsymbol{G}) \beta_{\text{tt}}\right)^2 + \left(\boldsymbol{x}_q^\top \boldsymbol{W} \boldsymbol{X}^\top \boldsymbol{\epsilon}\right)^2 + \sigma^2 \tag{82}$$

**Variance-Induced Error**

$$\begin{aligned}
\hat{\text{err}}_{variance}(\boldsymbol{W}) &= \mathbb{E}(\boldsymbol{x}_q^\top \boldsymbol{W} \boldsymbol{X}^\top \boldsymbol{\epsilon})^2 \\
&= \sum_{i,j=1}^{m+n} (\boldsymbol{x}_i^\top \boldsymbol{W}^\top \boldsymbol{x}_q)(\boldsymbol{x}_q^\top \boldsymbol{W} \boldsymbol{x}_j) \mathbb{E}(\epsilon_i \epsilon_j)
\end{aligned} \tag{83}$$

Because the noise are independent and zero-mean, we have

$$\mathbb{E}[\epsilon_j \epsilon_j] = \begin{cases} \sigma^2, & i = j \le m \\ \sigma^2_{\text{rag},i}, & i = j > m \\ 0, & i \ne j \end{cases}$$

Then,

$$\text{LHS} = \sum_{i=1}^{m} \sigma^2 \, \mathbb{E}[(\boldsymbol{x}_q^\top \boldsymbol{W} \boldsymbol{x}_i)^2] + \sum_{i=m+1}^{m+n} \sigma^2_{\text{rag},i-m} \cdot \mathbb{E}[\boldsymbol{x}_q^\top \boldsymbol{W} (\boldsymbol{x}_q + \boldsymbol{r}_{i-m})^2]$$

Thus, the ICL contribution remains the same as Theorem 1, i.e.

$$\sum_{i=1}^{m} \sigma^2 \, \mathbb{E}[(\boldsymbol{x}_q^\top \boldsymbol{W} \boldsymbol{x}_i)^2] = m\sigma^2 \, \text{tr}(\boldsymbol{W}^\top \boldsymbol{W})$$

To compute the RAG contribution, we evaluate the formula similar to Equation (46).

$$\mathbb{E}\left[ \left( \boldsymbol{x}_q^\top \boldsymbol{W} (\boldsymbol{x}_q + \boldsymbol{r}_i) \right)^2 \right] = \mathbb{E}[(\boldsymbol{x}_q^\top \boldsymbol{W} \boldsymbol{x}_q)^2] + \mathbb{E}[(\boldsymbol{x}_q^\top \boldsymbol{W} \boldsymbol{r}_i)^2] + 2\,\mathbb{E}[\boldsymbol{x}_q^\top \boldsymbol{W} \boldsymbol{x}_q \cdot \boldsymbol{x}_q^\top \boldsymbol{W} \boldsymbol{r}_i]$$

$$= \text{tr}(\boldsymbol{W}^\top \boldsymbol{W}) + \text{tr}(\boldsymbol{W}^2) + \delta_i^2 \, \text{tr}(\boldsymbol{W}^\top \boldsymbol{W}) + \text{tr}(\boldsymbol{W})^2 \tag{84}$$

And thus, the RAG error contribution is

$$\sum_{i=m+1}^{m+n} \sigma^2_{\text{rag},i-m} \cdot \mathbb{E}[\boldsymbol{x}_q^\top \boldsymbol{W} (\boldsymbol{x}_q + \boldsymbol{r}_{i-m})^2] = \sum_{i=1}^{n} \sigma^2_{\text{rag},i}[(1 + \delta_i^2) \, \text{tr}(\boldsymbol{W}^\top \boldsymbol{W}) + \text{tr}(\boldsymbol{W}^2) + \text{tr}(\boldsymbol{W})^2]$$

Plug in $\sigma^2_{\text{rag},i} = \gamma_1 \delta_i^2$, and combining all terms together, we have

$$\hat{\text{err}}_{\text{variance}}(\boldsymbol{W}) = m\sigma^2 \, \text{tr}(\boldsymbol{W}^\top \boldsymbol{W}) + \sum_{i=1}^{n} \gamma_1 \delta_i^2 [(1 + \delta_i^2) \, \text{tr}(\boldsymbol{W}^\top \boldsymbol{W}) + \text{tr}(\boldsymbol{W}^2) + \text{tr}(\boldsymbol{W})^2]$$

Now, if we further assume $\delta_i^2 = \gamma_2 i^q$, and plug in the value of

$$\hat{\text{err}}_{\text{variance}}(\boldsymbol{W}^*) = m\sigma^2 \, \text{tr}\left((\boldsymbol{W}^*)^\top \boldsymbol{W}^*\right) + \sum_{i=1}^{n} \gamma_1 \gamma_2 i^q[(1 + \gamma_2 i^q) \, \text{tr}\left((\boldsymbol{W}^*)^\top \boldsymbol{W}^*\right) + \text{tr}\left((\boldsymbol{W}^*)^2\right) + \text{tr}(\boldsymbol{W}^*)^2]$$

$$= \frac{m^2}{(m+d+1)^2(m+n)^2} \left[ dm\sigma^2 + \gamma_1 \gamma_2 \left[ (2d + d^2) \sum_{i=1}^{n} i^q \sigma^2 + d\gamma_2 \sum_{i=1}^{n} i^{2q} \sigma^2 \right] \right]$$

$$= \frac{m^2}{(m+d+1)^2(m+n)^2} \left[ dm\sigma^2 + \gamma_1 \gamma_2 \sigma^2 \left[ (2d + d^2) \mathcal{O}_n \left( \frac{n^{q+1}}{q+1} + \frac{n^q}{2} \right) + d\gamma_2 \mathcal{O}_n \left( \frac{n^{2q+1}}{2q+1} + \frac{n^{2q}}{2} \right) \right] \right]$$

$$= \mathcal{O}_n \left( \frac{dm\sigma^2 + d(n^{2q+1})}{(m+n)^2} \right)$$

$$= \begin{cases} \mathcal{O}_n \left( dn^{2q-1}\sigma^2 \right) & \text{if } n \to \infty, q < 1/2 \\ \mathcal{O}_n \left( d\sigma^2 \right) & \text{if } n \to \infty, q = 1/2 \\ \text{diverges} & \text{if } n \to \infty, q > 1/2 \end{cases}$$

where the second step follows from the Euler–Maclaurin expansion of the power sum.

**Bias-Induced Error** From Equation (61), we note that

$$\text{err}_{\text{bias}}(\boldsymbol{W}) = \beta_{\text{tt}}^\top \left[ M_1 - M_2 - M_3 + M_{41} + \sum_{i=1}^{n}(M_{42} + M_{42}^\top) + \sum_{i=1}^{n} M_{43} + \sum_{i \ne j, i,j \in [n]} M_{44} \right] \beta_{\text{tt}}$$

Specifically,

$$
\begin{aligned}
\underset{\boldsymbol{x}_q,\boldsymbol{X}}{\mathbb{E}} \left[ (I - \boldsymbol{W}\boldsymbol{G})^\top \boldsymbol{x}_q \boldsymbol{x}_q^\top (I - \boldsymbol{W}\boldsymbol{G}) \right] &= \underset{\boldsymbol{x}_q,\boldsymbol{X}}{\mathbb{E}} \left( I - \boldsymbol{G}\boldsymbol{W}^\top \right) \boldsymbol{x}_q \boldsymbol{x}_q^\top (I - \boldsymbol{W}\boldsymbol{G}) \\
&= \underbrace{\mathbb{E}\,\boldsymbol{x}_q \boldsymbol{x}_q^\top}_{:=M_1} - \underbrace{\mathbb{E}\,\boldsymbol{x}_q \boldsymbol{x}_q^\top \boldsymbol{W}\boldsymbol{G}}_{:=M_2} - \underbrace{\mathbb{E}\,\boldsymbol{G}\boldsymbol{W}^\top \boldsymbol{x}_q \boldsymbol{x}_q^\top}_{:=M_3} + \underbrace{\mathbb{E}\,\boldsymbol{G}\boldsymbol{W}^\top \boldsymbol{x}_q \boldsymbol{x}_q^\top \boldsymbol{W}\boldsymbol{G}}_{:=M_4}
\end{aligned}
\tag{85}
$$

To avoid the repeated computation, we will highlight the calculation that involves $\delta_i$, omit some calculation steps given in the standard case and discuss its bound after allowing for non-uniform offset. We will only compute $\delta_i^2$-involving term and use $\ldots$ to denote the rest terms, since we assume $\delta^2 \ll 1$ in proving Theorem 1. The final bound will be given as

$$
\widehat{\mathrm{err}}_{\mathrm{bias}}(\boldsymbol{W}^*) = \mathrm{err}_{\mathrm{bias}}(\boldsymbol{W}^*) + \delta^2\text{-involved terms}
$$

$M_1 = \mathbb{E}\left[\boldsymbol{x}_q \boldsymbol{x}_q^\top\right] = I$ and remains the same. Let $s_\delta := \sum_i \delta_i^2$, $S_\delta := \sum_i (\delta_i^2)^2$.

Then, we expand out the terms in $M_2$:

$$
\begin{aligned}
M_2 = \underset{\boldsymbol{x}_q,\boldsymbol{r}}{\mathbb{E}}\,\boldsymbol{x}_q \boldsymbol{x}_q^\top \boldsymbol{W}\boldsymbol{G} &= \left( \underset{\boldsymbol{x}_q,\boldsymbol{r}}{\mathbb{E}}\,\boldsymbol{x}_q \boldsymbol{x}_q^\top \right) \boldsymbol{W}\boldsymbol{G}_0 + \underset{\boldsymbol{x}_q,\boldsymbol{r}}{\mathbb{E}}\,\boldsymbol{x}_q \boldsymbol{x}_q^\top \boldsymbol{W} \sum_{i=1}^n (\boldsymbol{x}_q + \boldsymbol{r}_i)(\boldsymbol{x}_q + \boldsymbol{r}_i)^\top \\
&= \boldsymbol{W}\boldsymbol{G}_0 + \underset{\boldsymbol{x}_q,\boldsymbol{r}}{\mathbb{E}}\,\boldsymbol{x}_q \boldsymbol{x}_q^\top \boldsymbol{W} \sum_{i=1}^n (\boldsymbol{x}_q \boldsymbol{x}_q^\top + \boldsymbol{r}_i \boldsymbol{r}_i^\top) \\
&= \boldsymbol{W}\boldsymbol{G}_0 + \underset{\boldsymbol{x}_q,\boldsymbol{r}}{\mathbb{E}}\,\boldsymbol{x}_q \boldsymbol{x}_q^\top \boldsymbol{W} \sum_{i=1}^n (\boldsymbol{x}_q \boldsymbol{x}_q^\top + \delta_i^2 I) \\
&= \cdots + s_\delta \boldsymbol{W}
\end{aligned}
\tag{86}
$$

Similarly, $M_3 = M_2^\top = \cdots + s_\delta \boldsymbol{W}^\top$. Now, we perform similar expansion for $M_4$.

First, we note that $M_{41} = \mathbb{E}_{\boldsymbol{x}_q,\boldsymbol{X}}[\boldsymbol{G}_0 \boldsymbol{W}^\top \boldsymbol{x}_q \boldsymbol{x}_q^\top \boldsymbol{W}\boldsymbol{G}_0]$ is independent of $\delta_i^2$.

$$
\begin{aligned}
\sum_{i \in [n]} M_{42} := \sum_{i \in [n]} \underset{\boldsymbol{x}_q,\boldsymbol{X}}{\mathbb{E}}\,\boldsymbol{G}_0 \boldsymbol{W}^\top \boldsymbol{x}_q \boldsymbol{x}_q^\top \boldsymbol{W}\boldsymbol{G}_i \\
= \sum_{i \in [n]} \underset{\boldsymbol{x}_q,\boldsymbol{X}}{\mathbb{E}}\,\boldsymbol{G}_0 \boldsymbol{W}^\top \boldsymbol{x}_q \boldsymbol{x}_q^\top \boldsymbol{W} \left(\boldsymbol{x}_q + \boldsymbol{r}_i\right) \left(\boldsymbol{x}_q + \boldsymbol{r}_i\right)^\top \\
= \sum_{i \in [n]} \underset{\boldsymbol{x}_q,\boldsymbol{X}}{\mathbb{E}}\,\boldsymbol{G}_0 \boldsymbol{W}^\top \boldsymbol{x}_q \boldsymbol{x}_q^\top \boldsymbol{W} \left(\boldsymbol{x}_q \boldsymbol{x}_q^\top + \boldsymbol{r}_i \boldsymbol{r}_i^\top\right) \\
= \sum_{i \in [n]} \underset{\boldsymbol{x}_q,\boldsymbol{X}}{\mathbb{E}}\,\boldsymbol{G}_0 \boldsymbol{W}^\top \left(\boldsymbol{W} + \boldsymbol{W}^\top + \mathrm{tr}(\boldsymbol{W}) + \boldsymbol{W}\delta^2\right) \\
= \sum_{i \in [n]} m \left(\boldsymbol{W}^\top \boldsymbol{W} + \boldsymbol{W}^\top \boldsymbol{W}^\top + \mathrm{tr}(\boldsymbol{W})\boldsymbol{W}^\top + \delta^2 \boldsymbol{W}^\top \boldsymbol{W}\right) \\
= \cdots + m s_\delta \boldsymbol{W}^\top \boldsymbol{W}
\end{aligned}
\tag{87}
$$

Following the derivation of the 6th-order and 4th-order moments as in Lemma 3 and Lemma 2, we have

$$
\sum_{i\in[n]} M_{43} := \sum_{i\in[n]} \mathop{\mathbb{E}}_{\boldsymbol{x}_q, \boldsymbol{X}, \boldsymbol{r}_i} \boldsymbol{G}_i \boldsymbol{W}^\top \boldsymbol{x}_q \boldsymbol{x}_q^\top \boldsymbol{W} \boldsymbol{G}_i
$$

$$
\left.\begin{aligned}
&= \begin{aligned}
&2\left(\boldsymbol{W}^2 + (\boldsymbol{W}^2)^\top + \boldsymbol{W}^\top \boldsymbol{W} + \boldsymbol{W}\boldsymbol{W}^\top + \mathrm{tr}\left(\boldsymbol{W}\right)\left(\boldsymbol{W} + \boldsymbol{W}^\top\right)\right) \\
&+ \mathrm{tr}\left(\boldsymbol{W}\right)^2 I + \mathrm{tr}\left(\boldsymbol{W}^2\right) I + \mathrm{tr}\left(\boldsymbol{W}^\top \boldsymbol{W}\right) I
\end{aligned}
\end{aligned}\right\} \text{0th-order in } \boldsymbol{r}_i, \text{ Lemma 3}
$$

$$
+ \underbrace{2\delta_i^4 \boldsymbol{W}^\top \boldsymbol{W} + \delta_i^4 \mathrm{tr}(\boldsymbol{W}^\top \boldsymbol{W}) I}_{\text{4th-order in } \boldsymbol{r}_i, \text{ Equation (19)}}
$$

$$
+ \underbrace{\delta_i^2 \left[\mathrm{tr}\left(\boldsymbol{W}\right)\left(\boldsymbol{W}^\top + \boldsymbol{W}\right) + \boldsymbol{W}^2 + (\boldsymbol{W}^2)^\top + 2\boldsymbol{W}^\top \boldsymbol{W}\right]}_{\text{Equation (23) and its transpose}}
$$

$$
+ \underbrace{\left(\mathrm{tr}\left(\boldsymbol{W}^2\right) + \mathrm{tr}\left(\boldsymbol{W}^\top \boldsymbol{W}\right) + \mathrm{tr}\left(\boldsymbol{W}\right)^2\right)\delta_i^2 I}_{\text{Equation (20)}}
$$

$$
+ \underbrace{2\delta_i^2 \boldsymbol{W}\boldsymbol{W}^\top + \delta_i^2 \mathrm{tr}(\boldsymbol{W}^\top \boldsymbol{W}) I}_{\text{Equation (21)}}
$$

$$
+ \underbrace{\delta_i^2 \left[\mathrm{tr}\left(\boldsymbol{W}\right)\left(\boldsymbol{W}^\top + \boldsymbol{W}\right) + \boldsymbol{W}^2 + (\boldsymbol{W}^2)^\top + 2\boldsymbol{W}^\top \boldsymbol{W}\right]}_{\text{Equation (22) and its transpose}}
$$

$$
= \sum_{i\in[n]} (2 + 2\delta_i^2)\left[\mathrm{tr}\left(\boldsymbol{W}\right)\left(\boldsymbol{W}^\top + \boldsymbol{W}\right) + \boldsymbol{W}^2 + (\boldsymbol{W}^2)^\top\right]
$$

$$
+ \sum_{i\in[n]} (2 + 4\delta_i^2)\boldsymbol{W}^\top \boldsymbol{W} + \sum_{i\in[n]} 2\boldsymbol{W}\boldsymbol{W}^\top
$$

$$
+ \sum_{i\in[n]} (1 + \delta_i^2)\left[\mathrm{tr}\left(\boldsymbol{W}\right)^2 I + \mathrm{tr}\left(\boldsymbol{W}^2\right) I + \mathrm{tr}\left(\boldsymbol{W}^\top \boldsymbol{W}\right) I\right]
$$

$$
+ \sum_{i\in[n]} \left(2\delta_i^4 \boldsymbol{W}^\top \boldsymbol{W} + \delta_i^4 \mathrm{tr}(\boldsymbol{W}^\top \boldsymbol{W}) I + 2\delta_i^2 \boldsymbol{W}\boldsymbol{W}^\top + \delta_i^2 \mathrm{tr}(\boldsymbol{W}^\top \boldsymbol{W}) I\right)
$$

$$
= \cdots + 2s_\delta \left[\mathrm{tr}\left(\boldsymbol{W}\right)\left(\boldsymbol{W}^\top + \boldsymbol{W}\right) + \boldsymbol{W}^2 + (\boldsymbol{W}^2)^\top\right]
$$

$$
+ (4s_\delta + 2S_\delta)\boldsymbol{W}^\top \boldsymbol{W} + 2s_\delta \boldsymbol{W}\boldsymbol{W}^\top
$$

$$
+ s_\delta \left(\mathrm{tr}\left(\boldsymbol{W}\right)^2 + \mathrm{tr}\left(\boldsymbol{W}^2\right)\right) I + (2s_\delta + S_\delta)\mathrm{tr}(\boldsymbol{W}^\top \boldsymbol{W}) I
$$

(88)

Also, we expand the cross-term out for $\forall i, j \in [n], i \neq j$:

$$
\sum_{i\neq j} M_{44} := \sum_{i\neq j} \mathbb{E}\, \boldsymbol{G}_i \boldsymbol{W}^\top \boldsymbol{x}_q \boldsymbol{x}_q^\top \boldsymbol{W} \boldsymbol{G}_j
$$

$$
= \sum_{i\neq j} \left(\boldsymbol{x}_q \boldsymbol{x}_q^\top \boldsymbol{W}^\top \boldsymbol{x}_q \boldsymbol{x}_q^\top \boldsymbol{W} \boldsymbol{x}_q \boldsymbol{x}_q^\top + \boldsymbol{r}_i \boldsymbol{r}_i^\top \boldsymbol{W}^\top \boldsymbol{x}_q \boldsymbol{x}_q^\top \boldsymbol{W} \boldsymbol{x}_q \boldsymbol{x}_q^\top\right)
$$

$$
+ \sum_{i\neq j} \left(\boldsymbol{x}_q \boldsymbol{x}_q^\top \boldsymbol{W}^\top \boldsymbol{x}_q \boldsymbol{x}_q^\top \boldsymbol{W} \boldsymbol{r}_j \boldsymbol{r}_j^\top + \boldsymbol{r}_i \boldsymbol{r}_i^\top \boldsymbol{W}^\top \boldsymbol{x}_q \boldsymbol{x}_q^\top \boldsymbol{W} \boldsymbol{r}_j \boldsymbol{r}_j^\top\right)
$$

$$
= \cdots + \sum_{i\neq j} \delta_i^2 \left(\boldsymbol{W}^2 + \boldsymbol{W}^\top \boldsymbol{W} + \mathrm{tr}\left(\boldsymbol{W}\right)\boldsymbol{W}\right)
$$

$$
+ \sum_{i\neq j} \delta_i^2 \left((\boldsymbol{W}^2)^\top + \boldsymbol{W}^\top \boldsymbol{W} + \mathrm{tr}\left(\boldsymbol{W}\right)\boldsymbol{W}^\top\right)
$$

$$
+ \sum_{i\neq j} \delta_i^2 \delta_j^2 \boldsymbol{W}^\top \boldsymbol{W}
$$

(89)

In the non-uniform noise scenario, 4th-order term in $\delta_i$ will dominate the 2nd-order term in $\delta_i$. Thus, we will plug $\delta_i^2 = \gamma_2 i^q$, $\boldsymbol{W}^* = \frac{m}{(m+d+1)(m+n)}$ into $\mathrm{err}_{\mathrm{bias}}$:

$$\mathrm{err}_{\mathrm{bias}}(\boldsymbol{W}^*) = \mathrm{err}_{\mathrm{bias}}(\boldsymbol{W}^*) + \mathcal{O}_{m,n}\left(\beta_{\mathrm{tt}}^\top \left[2\sum_i^n (\delta_i^2)^2 (\boldsymbol{W}^*)^\top \boldsymbol{W}^* + \sum_{i \neq j, i \in [n], j \in [n]} \delta_i^2 \delta_j^2 (\boldsymbol{W}^*)^\top \boldsymbol{W}^* + \sum_i^n (\delta_i^2)^2 \mathrm{tr}((\boldsymbol{W}^*)^\top \boldsymbol{W})I\right]\beta_{\mathrm{tt}}\right)$$

$$= \mathrm{err}_{\mathrm{bias}}(\boldsymbol{W}^*) + \mathcal{O}_{m,n}\left(\beta_{\mathrm{tt}}^\top \left[dn^{2q+1}(\boldsymbol{W}^*)^\top \boldsymbol{W}^* + n^{2q+2}(\boldsymbol{W}^*)^\top \boldsymbol{W}^*\right]\beta_{\mathrm{tt}}\right)$$

$$= \mathrm{err}_{\mathrm{bias}}(\boldsymbol{W}^*) + \mathcal{O}_{m,n}\left(\beta_{\mathrm{tt}}^\top \left[\frac{dn^{2q+1} + n^{2q+2}}{(m+n)^2}\right]\beta_{\mathrm{tt}}\right)$$

It finishes the proof. $\qquad\square$

### B.2.2 DISTANCE-WEIGHTED PROBABILISTIC NOISE

**Theorem** (Restatement of Theorem 3). *Under Assumption 1, 2, 5, then* $\tilde{\mathrm{err}}_{bias}(\boldsymbol{W}) = \hat{\mathrm{err}}_{bias}(\boldsymbol{W})$, *and*

$$\tilde{\mathrm{err}}_{variance}(\boldsymbol{W}) = m\sigma^2 \mathrm{tr}(\boldsymbol{W}^\top \boldsymbol{W}) + \sum_{i=1}^n \left(p_i \sigma_s^2 + (1 - p_i)\sigma_l^2\right)\left[(1 + \delta_i^2)\mathrm{tr}(\boldsymbol{W}^\top \boldsymbol{W}) + \mathrm{tr}(\boldsymbol{W}^2) + \mathrm{tr}(\boldsymbol{W})^2\right]$$

*If the variance of the retrieval distance follows power law, i.e.* $\exists \gamma_2 > 0, q \geq 0$ *s.t.* $\delta_i^2 = \gamma_2 i^q$, *then:*

$$\tilde{\mathrm{err}}_{variance}(\boldsymbol{W}^*) = \begin{cases} \mathcal{O}_n\left(c_l dn^{q-1}\sigma^2 - (c_l - c_s)\sigma^2 dn^{q-1-q\tilde{q}}\right) & \text{if } n \to \infty, q \leq 1 \\ \text{diverges} & \text{if } n \to \infty, q > 1 \end{cases} \tag{90}$$

*Proof.* First, we note that $\tilde{\mathrm{err}}_{\mathrm{bias}}(\boldsymbol{W}) = \hat{\mathrm{err}}_{\mathrm{bias}}(\boldsymbol{W})$, since both are independent of $\sigma_{\mathrm{rag}}^2$ and depend on the same set of $\forall i, \delta_i^2$.

We write down error explicitly similar to Equation (40) and break down the population loss as:

$$\tilde{\mathcal{L}}_{\mathrm{tt+rag}}(\boldsymbol{W}) = \mathop{\mathbb{E}}_{(\boldsymbol{x}_q, y_q), (\boldsymbol{X}, \boldsymbol{y}), \boldsymbol{\epsilon}, \boldsymbol{r}}\left(\boldsymbol{x}_q^\top (I - \boldsymbol{W}\boldsymbol{G})\beta_{\mathrm{tt}}\right)^2 + \left(\boldsymbol{x}_q^\top \boldsymbol{W}\boldsymbol{X}^\top \boldsymbol{\epsilon}\right)^2 + \sigma^2 \tag{91}$$

We note that $\tilde{\mathrm{err}}_{\mathrm{bias}}(\boldsymbol{W}) = \mathrm{err}_{\mathrm{bias}}(\boldsymbol{W})$, since the error from bias does not depend on the sample complexity.

$$\tilde{\mathrm{err}}_{\mathrm{variance}}(\boldsymbol{W}) = \mathbb{E}(\boldsymbol{x}_q^\top \boldsymbol{W}\boldsymbol{X}^\top \boldsymbol{\epsilon})^2$$
$$= \sum_{i,j=1}^{m+n}(\boldsymbol{x}_i^\top \boldsymbol{W}^\top \boldsymbol{x}_q)(\boldsymbol{x}_q^\top \boldsymbol{W}\boldsymbol{x}_j)\,\mathbb{E}(\epsilon_i \epsilon_j) \tag{92}$$

Because the noise are independent and zero-mean, we have

$$\mathbb{E}[\epsilon_j \epsilon_j] = \begin{cases} \sigma^2, & i = j \leq m \\ \sigma_s^2, & i = j > m, \text{ w.p. } p \\ \sigma_l^2, & i = j > m, \text{ w.p. } 1 - p \\ 0, & i \neq j \end{cases}$$

Thus, the ICL contribution remains the same as Theorem 1, i.e.

$$\sum_{i=1}^m \sigma^2\,\mathbb{E}[(\boldsymbol{x}_q^\top \boldsymbol{W}\boldsymbol{x}_i)^2] = m\sigma^2\,\mathrm{tr}(\boldsymbol{W}^\top \boldsymbol{W})$$

To compute the RAG contribution, we evaluate the formula similar to Equation (46).

$$\mathbb{E}\left[\left(\boldsymbol{x}_q^\top \boldsymbol{W}(\boldsymbol{x}_q + \boldsymbol{r}_i)\right)^2\right] = \mathbb{E}[(\boldsymbol{x}_q^\top \boldsymbol{W}\boldsymbol{x}_q)^2] + \mathbb{E}[(\boldsymbol{x}_q^\top \boldsymbol{W}\boldsymbol{r}_i)^2] + 2\,\mathbb{E}[\boldsymbol{x}_q^\top \boldsymbol{W}\boldsymbol{x}_q \cdot \boldsymbol{x}_q^\top \boldsymbol{W}\boldsymbol{r}_i]$$
$$= \mathrm{tr}(\boldsymbol{W}^\top \boldsymbol{W}) + \mathrm{tr}(\boldsymbol{W}^2) + \delta_i^2\,\mathrm{tr}(\boldsymbol{W}^\top \boldsymbol{W}) + \mathrm{tr}(\boldsymbol{W})^2 \tag{93}$$

And thus, the RAG error contribution is

$$\sum_{i=1}^n \left(p_i \sigma_s^2 + (1 - p_i)\sigma_l^2\right)\left[(1 + \delta_i^2)\mathrm{tr}(\boldsymbol{W}^\top \boldsymbol{W}) + \mathrm{tr}(\boldsymbol{W}^2) + \mathrm{tr}(\boldsymbol{W})^2\right]$$

Plug in $\sigma_{\text{rag},i}^2 = \gamma_1 \delta_i^2$, and combining all terms together, we have

$$\hat{\text{err}}_{\text{variance}}(\boldsymbol{W}) = m\sigma^2 \operatorname{tr}(\boldsymbol{W}^\top \boldsymbol{W}) + \sum_{i=1}^{n} \left(p_i \sigma_s^2 + (1-p_i)\sigma_l^2\right) \left[(1+\delta_i^2)\operatorname{tr}(\boldsymbol{W}^\top \boldsymbol{W}) + \operatorname{tr}(\boldsymbol{W}^2) + \operatorname{tr}(\boldsymbol{W})^2\right]$$

Now we further assume $p_i = (1+\delta_i^2)^{-\tilde{q}}$, $\tilde{q} \geq 0$, and plug in the value of $\boldsymbol{W}^*$. Let $B := \frac{m^2}{(m+d+1)^2(m+n)^2}$,

$$\tilde{\text{err}}_{\text{variance}}(\boldsymbol{W}^*) = m\sigma^2 \operatorname{tr}(\boldsymbol{W}^\top \boldsymbol{W}) + \sum_{i=1}^{n} \left(p_i \sigma_s^2 + (1-p_i)\sigma_l^2\right) \left[(1+\delta_i^2)\operatorname{tr}(\boldsymbol{W}^\top \boldsymbol{W}) + \operatorname{tr}(\boldsymbol{W}^2) + \operatorname{tr}(\boldsymbol{W})^2\right]$$

$$= B\left[dm\sigma^2 + \sum_{i=1}^{n} \left(c_l\sigma^2 - (1+\delta_i^2)^{-\tilde{q}}(c_l - c_s)\sigma^2\right)\left[(1+\delta_i^2)\cdot d + d + d^2\right]\right]$$

$$\approx B\left[dm\sigma^2 + c_l\sigma^2 \sum_{i=1}^{n} \left(d\delta_i^2 + d^2\right) - (c_l - c_s)\sigma^2 \sum_{i=1}^{n} \left(d(1+\delta_i^2)^{1-\tilde{q}} + d^2(1+\delta_i^2)^{-\tilde{q}}\right)\right]$$

$$\approx B\left[dm\sigma^2 + c_l\sigma^2 \sum_{i=1}^{n} d\delta_i^2 - (c_l - c_s)\sigma^2 \sum_{i=1}^{n} d(1+\delta_i^2)^{1-\tilde{q}}\right]$$

$$\approx \begin{cases} B\left[dm\sigma^2 + c_l\sigma^2 dn^{q+1} - (c_l - c_s)\sigma^2 d\log(n)\right] & \text{if } \tilde{q} = 1 + 1/q \\ B\left[dm\sigma^2 + c_l\sigma^2 dn^{q+1} - (c_l - c_s)\sigma^2 dn^{1+q-q\tilde{q}}\right] & \text{else} \end{cases}$$

where the second line follows from omitting the lower order term.

If $\tilde{q} = 1 + 1/q$, we note that the middle term will dominate the error. And combining all cases, we could obtain

$$\tilde{\text{err}}_{\text{variance}}(\boldsymbol{W}^*) = \begin{cases} \mathcal{O}_n\left(c_l dn^{q-1}\sigma^2 - (c_l - c_s)\sigma^2 dn^{q-1-q\tilde{q}}\right) & \text{if } n \to \infty, q \leq 1 \\ \text{diverges} & \text{if } n \to \infty, q > 1 \\ \mathcal{O}_n\left(c_l dn^{q-1}\sigma^2 + (c_l - c_s)d^2\frac{\log n}{n^2}\sigma^2\right) & \text{if } n \to \infty, \tilde{q} = 1 + 1/q \end{cases}$$

$\square$

## C   MORE DETAILS FOR THE EXPERIMENTS

For Natural Questions (NQ), the retrieval index is constructed from the December 2018 Wikipedia dump. For TriviaQA, we use the December 2021 version. To accommodate hardware limitations, we randomly subsample 10% of the full index for both datasets. This reduces retrieval cost and memory usage, allowing all experiments to be conducted on a single NVIDIA A100 or L40 GPU.

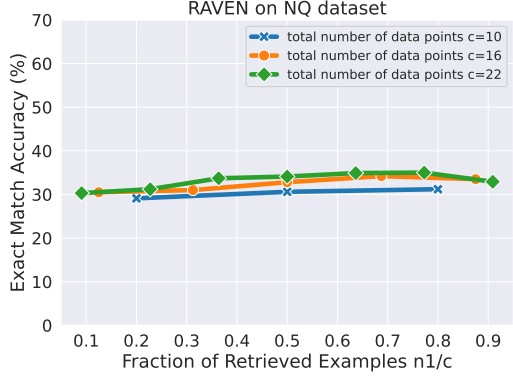 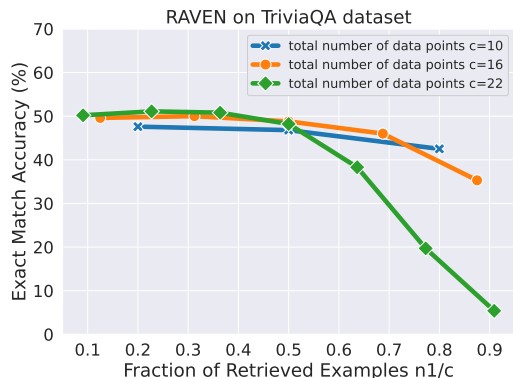

(a) RAVEN Performance as a function of $n_1/c$ under different data points $c$ on NQ.

(b) RAVEN Performance as a function of $n_1/c$ under different data points $c$ on TriviaQA.

Figure 4: Performance sensitivity to the ratio $n_1/n$ under different data points $c$, where $n_1$ refers to retrieved examples and $n_2$ to passages.