# OpenReview forum: "Retrieval-Augmented Generation as Noisy In-Context Learning: A Unified Theory and Risk Bounds"
_ICLR.cc/2026/Conference — Submitted to ICLR 2026_

### Official Review · Reviewer_rY2P · 2025-10-21

**Soundness:** 3
**Presentation:** 3
**Contribution:** 3
**Rating:** 6
**Confidence:** 4

**Summary:**

The paper systematically investigates generalization bound for retrieval-augmented generation (RAG) in in-context linear regression and derive an exact bias-variance tradeoff.
The proposed framework recovers classical in-context learning (ICL) and standard RAG as limiting cases by viewing the retrieved texts as query-dependent noisy in-context examples.
The paper presents an interesting finding: an intrinsic ceiling on generalization error exists in RAG, as opposed to ICL.
Experiments across two downstream datasets valiate the sample efficiency of ICL and RAG.

**Strengths:**

1. Important Problem: The paper identifies and systematically investigates an important problem.

2. Clear Organization: The paper is well-structured, with clear problem formulation, and visualizations.

3. Solid theoretical background: The proposed framework for RAG in in-context linear regression appears to be built upon a solid theoretical foundation.

4. Comprehensive Experimental Evaluation: Extensive and detailed experimental results provide clear evidence and detailed analysis.

**Weaknesses:**

1. The paper assumes that all RAG or ICL examples follow independent Gaussian distributions. However, some studies have shown that ICL or RAG use similarity-based retrievers to select demonstrations [1]. It would be useful for the authors to discuss the scenario where RAG or ICL examples are retrieved based on similarity-based retrieval methods.

2. The paper only considers a single-layer linear self-attention (LSA) model. It would be useful to extend the proposed framework to nonlinear or multi-layer self-attention architectures.

3. The paper validate the proposed framework only on two simple QA datasets. The authors should add more complex downstream tasks.

4. The author needs to correct the page margins of the whole paper.

5. Missing the analysis of different model sizes and architectures.

6. Missing the analysis of different prompt length.

7. Missing the details of used dataset, such as dataset sizes, prompts.

[1] What makes good in-context examples for GPT-3

**Questions:**

See Weaknesses

---

> ### Author Response · Authors · 2025-11-28
> **Rebuttal to Review rY2P**
>
> We really thank the reviewer rY2P for carefully reading our paper and for the insightful comments. Here, we address the questions and comments below.
>
>
> ## Response to Weakness
>
> 1. **Gaussian assumption vs. similarity-based retrieval.**
>
>     We agree that practical RAG/ICL uses similarity-based retrievers [1]. Our framework matches the “RAG to select ICL examples” setting in [1], which corresponds to our RAG-with-examples experiments. The key difference between that setting and general RAG is the noise structure of the retrieved text: examples come from a labeled dataset in one case, while passages can contain substantial irrelevant content in the other. Our analysis focuses on how performance depends on (i) query–example alignment and (ii) the noise distribution; similarity-based retrieval mainly changes this distribution. We will add a short discussion explaining how similarity-based retrievers fit into our model and how one could relax the Gaussian assumption.
>
>
> 2. **Single-layer linear self-attention.**
>     We use a single-layer linear self-attention model because it is the simplest architecture for which we can obtain a full, explicit analysis. Extending the framework to nonlinear and multi-layer models is an important direction, but technically nontrivial and beyond the scope of this paper. We will make this scope choice explicit in the revision.
>
>
> 3. **Only two QA datasets.**
>     Our goal is to test the qualitative predictions of the theory, not to benchmark RAG broadly. For this purpose, NaturalQuestions and TriviaQA are sufficient: they already show that the key trends from the analysis (e.g., performance drop with too many retrieved examples, effect of noise) appear in realistic setups. We are working on more diverse datasets, but we believe the current experiments are adequate for this theoretical paper and will clarify this positioning.
>
>
> 4. **Page margins.**
>     We will correct the page margins to fully comply with the ICLR style.
>
>
> 5. **Model sizes and architectures.**
>     Our main contribution is theoretical: we analyze a specific linear attention model and connect it to RAG behavior. A broad empirical sweep over model sizes and architectures is interesting but orthogonal to our main questions, and thus out of scope here. We will state this explicitly.
>
>
> 6. **Prompt length.**
>     In all experiments, we fix the prompt length by design to match the theoretical setup and avoid extra confounders. Varying prompt length is therefore not required to test our main claims. We will add a short note explaining this choice.
>
>
> 7. **Dataset and prompt details.**
>     We used NaturalQuestions dataset with train split size 80k and test example 3.6k. And we used open domain split TriviaQA with train/test size 80k/11k. Regarding the prompt format, we use prompts similar to RAVEN[1], of the form
> `q1 a1; q2 a2; …; q0 [answer placeholder] + passage`,
> where qi, ai are past QA pairs, q0 is the current question, and passage is the retrieved context.
>
> [1] Jie Huang, Wei Ping, Peng Xu, Mohammad Shoeybi, Kevin Chen-Chuan Chang, and Bryan Catanzaro. Raven:
> In-context learning with retrieval-augmented encoder-decoder language models, 2024.

---

### Official Review · Reviewer_CWz6 · 2025-10-30

**Soundness:** 2
**Presentation:** 3
**Contribution:** 2
**Rating:** 4
**Confidence:** 3

**Summary:**

The paper models RAG as noisy in-context learning and presents the first finite-sample generalization bounds with an exact bias–variance decomposition for in-context linear regression, revealing an intrinsic performance ceiling. The framework recovers vanilla ICL and standard RAG as limiting cases and formalizes query-dependent retrieval via Gaussian offsets around the query. Under uniform retrieval noise, variance error decreases with more retrieved examples while bias error does not, producing diminishing returns and a plateau.

**Strengths:**

* The paper offers a rigorous, unified view by casting RAG as noisy ICL and deriving finite-sample bounds plus an explicit bias–variance split.

* The query-dependent retrieval offset and the two noise regimes (uniform and distance-dependent) align well with practical retrieval behavior.

* The theory isolates why extra retrieval beyond a point stops helping by showing variance reduction without bias reduction under uniform noise.

**Weaknesses:**

* The theoretical results rely on a single-layer linear self-attention model with Gaussian inputs and isotropic noises, which may limit transfer to modern nonlinear transformer stacks.

* The evaluation metric in theory is MSE while the experiments report EM on QA, and the paper does not quantify how this metric mismatch affects conclusions.


* The Gaussian retrieval-offset assumption simplifies real retrieval distributions and ignores indexing heuristics and filtering used in practice

The experimental scope covers two QA datasets and two model families without sensitivity to retriever choice, retrieval-pool size, or fitted exponents q and \hat_q

**Questions:**

Can the authors provide an online diagnostic that separates variance-driven gains from bias-driven plateaus to indicate when to stop adding documents.

How robust are the conclusions when the retrieval-offset distribution deviates from Gaussian due to ANN index structures or filtered negative mining.

---

> ### Author Response · Authors · 2025-11-28
> **Rebuttal to Reviewer CWz6**
>
> We really thank the reviewer CWz6 for carefully reading our paper and for the insightful comments. Here, we address the questions and comments below.
>
> ## Response to Weakness
> 1. **Linear selfattention model.**
>     We agree this model is simpler than modern stacks; we chose it to make a full analysis possible. As noted in our global response, extending to nonlinear and multi-layer transformers is an important but orthogonal next step rather than something we can do in this paper.
>
> 2. **MSE vs EM.**
>     The theory works in a regression view (MSE on logits), while experiments report EM after mapping logits to tokens by argmax. Our claims are about relative trends (e.g., effect of more examples or more noise), which are governed by the underlying regression error and are preserved after discretization; we will state this mapping explicitly.
>
> 3. **Gaussian retrieval-offset assumption.**
>     The Gaussian assumption is a tractable proxy for real retrieval offsets. It abstracts the effect of retrieval into relevance and noise statistics; most practical indices and mining schemes still induce a concentrated, similarity-based offset distribution, so the main insights carry over even if the exact distribution is not Gaussian.
>
> 4. **Experimental scope.**
>     We only vary two QA datasets and two model families, without sweeping retrievers, pool sizes, or exponents. Since the paper is primarily theoretical, we see this as sufficient to check that the key qualitative predictions show up in realistic setups; a broader empirical sweep is left for future work.
>
> ## Response to Questions
>
> 1. **Diagnostic for when to stop adding documents.**
>     A precise variance–bias decomposition online is infeasible because the true distribution is unknown. Guided by our theory, a practical diagnostic is to monitor performance as the number of retrieved documents increases and stop once it plateaus or drops, indicating that added documents are becoming less relevant and bias dominates.
>
> 2. **Robustness beyond Gaussian offsets.**
>     When ANN indexing or hard-negative mining makes offsets non-Gaussian, we still expect the qualitative behavior to follow our analysis as long as relevance and noise behave similarly. Handling fully realistic, non-Gaussian offset distributions would require more technical machinery and is beyond the scope of the current work.

---

### Official Review · Reviewer_SkT8 · 2025-10-31

**Soundness:** 2
**Presentation:** 3
**Contribution:** 3
**Rating:** 4
**Confidence:** 3

**Summary:**

This paper (1) formalizes RAG as noisy in-context learning for in-context linear regression with a single-layer LSA predictor, (2) derives finite-sample generalization bounds and an exact bias–variance decomposition under query-dependent retrieval, and (3) analyzes both uniform and non-uniform noise regimes to explain when retrieved context helps, plateaus, or harms.

**Strengths:**

The most tangible contribution is a finite-sample, closed-form risk analysis that cleanly separates variance and bias effects of adding retrieved examples. Partition-depth-like roles are played by the number and distance of retrieved items, producing an optimal 𝑛* with diminishing returns O(1/m²). This provides decision-relevant guidance for when to stop retrieving and how to trade off close versus far items.

**Weaknesses:**

1. The theory rests on stylized assumptions, e.g., Gaussian linear data, LSA proxy, no RAG finetuning, and power-law retrieval distance distributions. These assumptions enable tractability, and the conclusions should be read as qualitative guidance.
2. The empirical section is still preliminary in scope, and does not benchmark against strong retrieval policies or compression strategies, such as error–time–memory Pareto. Consequently, claims about optimal budgets and mixing ratios remain suggestive rather than decisive for deployment.

**Questions:**

How sensitive are your bounds and the optimal 𝑛* to deviations from Gaussian linear assumptions, and to adding RAG fine-tuning or context compression?

---

> ### Author Response · Authors · 2025-11-28
> **Rebuttal to Reviewer SkT8**
>
> We thank the reviewer SkT8 for the careful reading and constructive feedback. We appreciate the positive assessment of our main contribution, and we address the concerns raised point by point in the following response.
>
> ## Response to Weakness
>
> 1. **Stylized setup is still valuable.** As we note in our global response, the LSA model is a standard stylized setting in statistical learning and allows us to derive closed-form bounds and an explicit optimality. The key of our analsyis is to break the down bound given different prediction noise and retrieval noise, rather than the specific Gaussianity or linearity per se.
>
>     Our goal is to provide quantitative guidance while still preserving intuition. Since a full theoretical treatment of modern LLMs is still very challenging, it is common practice to study key components in isolation—both experimentally and theoretically—to disentangle their effects in a controlled way.
>
>     RAG fine-tuning can be naturally interpreted in this framework. Retriever fine-tuning corresponds to an improved ability to retrieve relevant information, which in our model can be viewed as reducing the effective retrieval noise term. LLM fine-tuning can analogously be viewed as reducing the prediction noise conditional on the retrieved context.
>
>
>
> 2. **Strong retrieval policy is orthogonal to our theoretical analsyis.** Strong retrieval policies and context compression are interesting additional components to analyze, but they are not part of our current setup. Our notion of optimal budget allocation and mixing ratio is derived for a setting without compression, where the cost is driven by the number of retrieved items under a fixed interface to the model. A realistic deployment of a modern, highly engineered RAG system (with learned query reformulation, aggressive compression, routing, etc.) introduces many system-level design choices that go beyond the clean statistical setting we consider here. We view extending the analysis to such architectures as valuable future work, but it is orthogonal to the main theoretical question addressed in this paper.
>
>
> ## Response to Question
>
> 1. Our goal is to provide quantitative guidance in a simplified Gaussian linear setting while retaining clear insight. RAG fine-tuning and context compression can be viewed as interventions that change the effective noise and accuracy parameters of the retriever and predictor, thereby shifting the numerical values in our guarantees but not their qualitative structure. Because a full theory of modern LLMs is still very challenging, it is standard to study key components in to understand their effects, and our work follows this approach.

---

### Official Review · Reviewer_t2Gz · 2025-11-01

**Soundness:** 3
**Presentation:** 3
**Contribution:** 2
**Rating:** 6
**Confidence:** 2

**Summary:**

This paper presents a theoretical framework that views RAG as a form of noisy ICL. The authors derive the first finite-sample generalization bounds for RAG in a linear regression setting, analyzing how the number and quality of retrieved examples impact model performance. They introduce two noise models—uniform and non-uniform—to capture different retrieval scenarios (e.g., retrieving from a generic corpus vs. a labeled training set). Their theoretical findings suggest that while RAG can initially reduce variance-induced error, there's a ceiling to its effectiveness, and adding too many noisy examples can hurt performance. These theoretical insights are then backed up by experiments on common QA datasets like Natural Questions and TriviaQA, where the results align well with their proposed models.

**Strengths:**

1. It's great to see a paper that finally puts some solid theory behind RAG, which has been a mostly empirical field until now.

2. The idea of modeling RAG as noisy ICL is quite clever and provides a unified lens to understand its connection to standard in-context learning.

3. The experimental results do a good job of backing up the theoretical claims, especially in showing how performance can drop when you add too many retrieved examples.

**Weaknesses:**

1. The analysis is limited to a linear regression setting, which feels a bit disconnected from the complex, non-linear reality of modern language models.

2. The paper doesn't touch on how RAG fine-tuning might change the dynamics, which is a pretty common way people use RAG in practice.

3. While the noise models are a good start, they might be too simple to capture all the different ways retrieval can be "noisy" in the real world.

**Questions:**

1. I'm curious if you have any thoughts on how your theoretical framework might extend to more complex, non-linear models like Transformers. Do you think the fundamental trade-offs you identified would still hold?

---

> ### Author Response · Authors · 2025-11-28
> **Rebuttal to Reviewer t2Gz**
>
> We thank the reviewer t2Gz for the careful reading and constructive feedback. We appreciate the positive assessment of our main contribution, in particular the recognition that (i) the paper provides **solid theory for RAG**, (ii) modeling RAG as noisy in-context learning offers a **unified view of its relation to standard ICL**, and (iii) the **correspondence of experiments to the theory**. We address the remaining concerns point by point in the following response.
>
>
> ## Response to Weakness
>
> 1. **Linear self-attention setting as a stylized setting.** We fully acknowledge that modern LMs are highly non-linear. Our choice of a linear / linear self-attention setting is deliberate: it is a standard stylized model that allows finite-sample, closed-form risk analysis and a clean separation between retrieval noise and prediction noise. The phenomena we study depend primarily on these noise components rather than on a particular nonlinearity, and we expect the qualitative trade-offs to extend to richer models as theory for nonlinear transformers matures, even though a full extension is beyond the scope of this paper. See our global response for more detailed explanation.
>
> 2. **Interpretation of RAG finetuning RAG finetuning fits naturally into our framework.** Retriever fine-tuning corresponds to reducing the effective retrieval noise (better relevance and coverage), while LLM fine-tuning reduces prediction noise conditional on the retrieved context. In our analysis, this means that fine-tuning mainly shifts the values of the noise parameters and thus the resulting operating point, without changing the structure of the bounds or the basic retrieval–prediction trade-off we characterize. A detailed, method-specific treatment of concrete RAG fine-tuning pipelines is complementary but outside the scope of this theoretical work.
>
> 3. **Our noise models are intentionally simple.** In real RAG systems, retrieval error arises from many intertwined sources (distribution shift, partial relevance, conflicting or correlated documents, truncation, etc.), and capturing all of these in a single tractable model is difficult. Our goal here is to use a minimal abstraction that still exposes, in closed form, how risk depends on retrieval quality versus prediction quality. We view this as a clean baseline; extending the framework to more structured and realistic noise processes is an important direction for future work.
>
>
>
> ## Response to Questions
>
> 1. **Results under complex settings.** We view the linear self-attention setting as largely orthogonal to the core phenomenon we study. The key objects in our framework are the retrieval-side and prediction-side noise terms, not the specific choice of nonlinearity. For this reason, we expect the same qualitative trade-offs (diminishing returns from additional retrieved context) to persist for more complex models. What is currently missing, and beyond the scope of this work, is the technical machinery to prove analogous results in a fully realistic transformer model.

---

### Author Response · Authors · 2025-11-28
**Global Response to questions about the stylized LSA setup**

We really thank the reviewer for carefully reading our paper and for the insightful comments. Here, we address the recurring concerns and questions about our stylized LSA setup.


1. **LSA is simple yet useful for analyzing transformers.**  Our stylized linear self-attention (LSA) setup is **simple yet useful** for theoretical insights about RAG. To the best of our knowledge, this setting has *not been analyzed theoretically** before. Showing that we can fully analyze this regime is a basic but important step: it tells us what these methods do in a controlled case and clarifies which effects already appear in the simplest possible model. This simplification is also commonly used to analyze the incontext behavior of transformers in the previous literatures:
    - **Zhang et al. (JMLR ’24)** [1] proves that gradient flow on a one‑layer LSA transformer converges to a predictor that matches ordinary least squares.
    - **Mahankali et al. (ICLR ’24)** [2] shows a single LSA layer trained on noisy linear regression learns the provably optimal one‑step gradient‑descent algorithm for in‑context learning.
    - **Ahn et al. (NeurIPS ’24)** [3] analyzes multi‑layer linear transformers and demonstrate they learn preconditioned gradient descent, linking deeper LSA stacks to classical optimization.
These studies underline that LSA isn’t just a toy example—it’s the natural first step toward a principled, non‑heuristic understanding of full transformers, so our RAG analysis starts in the right place.


2. **Correpondence between simple theory and empirical observation.** The analysis is also relevant beyond this specific setup. The main trends we derive from the theory match what we observe in more realistic experiments, and we see the same qualitative behavior in practical scenarios. This supports the claim that the conclusions are not tied to the exact form of the linear model and do matter for practical systems.

3. **Simplification as factor isolation.** The simplification is useful because it lets us clearly isolate two key factors that drive performance: how the behavior depends on the query, and how different noise sources affect the outcome. These factors also appear in more complex practical settings, as we confirm empirically, but the simple model is what allows us to identify and study them cleanly. We then support the theoretical findings with experiments in more realistic setups, showing that the same patterns appear there as well.


[1] Zhang, Ruiqi, Spencer Frei, and Peter L. Bartlett. "Trained transformers learn linear models in-context." Journal of Machine Learning Research 25, no. 49 (2024): 1-55.

[2] Mahankali, Arvind, Tatsunori B. Hashimoto, and Tengyu Ma. "One step of gradient descent is provably the optimal in-context learner with one layer of linear self-attention." arXiv preprint arXiv:2307.03576 (2023).

[3] Ahn, Kwangjun, Xiang Cheng, Hadi Daneshmand, and Suvrit Sra. "Transformers learn to implement preconditioned gradient descent for in-context learning." Advances in Neural Information Processing Systems 36 (2023): 45614-45650.

---

### Meta-Review · Area_Chair_F8C6 · 2025-12-09

**Summary:**

This paper proposes a theoretical framework that models Retrieval-Augmented Generation (RAG) as noisy In-Context Learning (ICL), deriving finite-sample generalization bounds and bias-variance tradeoffs under a simplified linear self-attention (LSA) setting.

### Pros
* Modeling RAG as “ICL with Noise” offers an intriguing perspective that attempts to link retrieval quality (noise) with generative performance.

* As one of the few attempts to establish a learning theory foundation for RAG, deriving a closed-form solution is commendable.
### Cons

* Over-simplified Assumptions
* Limited Empirical Scope
* Formatting Violation: according to ICLR CFPs(https://iclr.cc/Conferences/2026/AuthorGuide), this should be desk-rejected

### AC's evaluation

1. from reviews and rebuttals

This paper receives 6644, this is actually a borderline case. And reviewers are ambivalent, with no support for acceptance. Reviewers t2Gz (6) and rY2P (6) appreciated the theoretical novelty but explicitly noted the limitations of the linear assumptions, offering only "marginal" support. Reviewers SkT8 (4) and CWz6 (4) were more critical, arguing that the highly stylized assumptions create a disconnect with reality, rendering the insights less actionable for real-world deployment.

2. from AC's reading

This is actually a borderline case in my batch. First, the formatting violation is a critical issue. Modifying margins violates ICLR policy.
Second, regarding technical merit, while the "RAG as Noisy ICL" concept is promising, the execution (LSA + Gaussian) is too reductive. The authors defended this as a "necessary simplification" in the rebuttal, but this does not bridge the gap to practical utility.
The core theory relies entirely on single-layer linear self-attention (LSA) and the assumption of Gaussian data distribution. As multiple reviewers (CWz6, SkT8, t2Gz) have pointed out, this represents a significant gap compared to the deep nonlinear Transformers used in modern RAG systems, making it difficult for theoretical conclusions to guide practical implementation.
The experiment only covered two datasets and lacked comparisons with strong retrieval strategies or actual RAG pipelines. Furthermore, the theoretical analysis was based on MSE while the experiment relied on EM, and the reason for this mismatch remains unclear.
Also, the author's rebuttal is insufficient and unconvincing enough.
Even disregarding formatting issues, this paper still fails to meet the bar for acceptance as a theoretical study on RAG.

**Reviewer Concerns:**

Resolved Concerns:

1. Interpretation of Assumptions (Reviewer t2Gz): The reviewer inquired about extending the theory to nonlinear models. The authors clarified that the current linear setting is a trade-off to obtain closed-form solutions and argued that qualitative trends may still apply. The reviewer appeared to accept this explanation but did not elevate the evaluation accordingly.

2. Clarification on Metrics (Reviewer CWz6): Regarding the mismatch between MSE and EM metrics, the authors explained that regression errors determine relative trends and committed to explicitly stating this in the final version.

Outstanding Concerns

1. Formatting Violation: This is a critical Outstanding issue. Authors modified page margins to reduce length, violating ICLR submission policies.

2. Gap between Theory and Practice (Reviewers CWz6, SkT8, rY2P): This is the core concern shared by all reviewers. The assumptions of a single-layer LSA and Gaussian retrieval noise are overly detached from reality (Reality gap). While the authors emphasize the value of simplifying the model in their Rebuttal, reviewers generally believe this is insufficient to justify the ambitious title “Unified Theory” and struggles to capture the complex retrieval noise distribution in the real world.

3. Limited Experimental Validation (Reviewers SkT8, CWz6): The experimental scope is narrow (only 2 datasets) and lacks analysis of practical strategies like RAG fine-tuning and context compression. Reviewer SkT8 explicitly states the current experiments are insufficient to provide deployment recommendations (“suggestive rather than decisive”).

**Reviewer Scores:**

1. Reviewer t2Gz (6) expected to maintain 6 points. He finds the theory valuable and assigns an encouraging score

2. Reviewer rY2P (6) expected to maintain 6 points. Although pointing out numerous issues (including formatting), his overall tone is relatively mild.

3. Reviewer SkT8 (4) definitely maintain 4 points. He highly values the theory's practical guidance but believes the current simplifying assumptions limit the conclusions' value.

4. Reviewer CWz6 (4) definitely maintain 4 points. He is very concerned about metric mismatches and assumptions, and the rebuttal did not address the physical limitations.

---

### Decision · Program_Chairs · 2026-01-26

Reject